# SNAP-TTA: Sparse Test-Time Adaptation for Latency-Sensitive Applications

## Abstract

Test-Time Adaptation (TTA) methods use unlabeled test data to dynamically adjust models in response to distribution changes. However, existing TTA methods are not tailored for practical use on edge devices with limited computational capacity, resulting in a latency-accuracy trade-off. To address this problem, we propose SNAP-TTA, a sparse TTA framework that significantly reduces adaptation frequency and data usage, delivering latency reductions proportional to adaptation rate. It achieves competitive accuracy even with an adaptation rate as low as 0.01, demonstrating its ability to adapt infrequently while utilizing only a small portion of the data compared to full adaptation. Our approach involves (i) Class and Domain Representative Memory (CnDRM), which identifies key samples that are both class-representative and domain-representative to facilitate adaptation with minimal data, and (ii) Inference-only Batch-aware Memory Normalization (IoBMN), which leverages representative samples to adjust normalization layers on-the-fly during inference, aligning the model effectively to changing domains. When combined with five state-of-the-art TTA algorithms, SNAP-TTA maintains the performances of these methods even with much-reduced adaptation rates from 0.01 to 0.5, making it suitable for edge devices serving latency-sensitive applications.

## 1 Introduction

Deep learning models often suffer from performance degradation under domain shifts caused by environmental changes or noise (Quiñonero-Candela et al., 2008). Test-Time Adaptation (TTA) offers a promising solution for domain shifts by utilizing only unlabeled test data without requiring source data. While TTA algorithms have advanced in complexity to improve accuracy in data streams (Wang et al., 2021; Niu et al., 2022; Wang et al., 2022; Yuan et al., 2023; Niu et al., 2023; Song et al., 2023), they are typically designed for resource-rich servers, overlooking the computational and memory limitations crucial for real-world deployment. Operations such as backpropagation, data augmentation, and model ensembling (Wang et al., 2022; Yuan et al., 2023; Zhang et al., 2022) result in substantial latency and memory consumption, making state-of-the-art (SOTA) TTA methods inefficient for practical use (Section 2).

For edge devices with limited computational power, such as mobile devices or IoT sensors, the adaptation latency from TTA methods becomes a critical bottleneck, particularly in latency-sensitive applications such as autonomous driving and real-time health monitoring. Moreover, the model must keep up with the data stream in those applications, but high computational overhead could cause it to miss critical samples, resulting in inference lags and reduced accuracy. This issue is exacerbated with fast data streams, such as high-frame-rate videos or high-performance sensors. For example, even a slight delay in processing sensor data can lead to dangerous situations in autonomous driving. A high adaptation latency that accumulates with each batch not only undermines real-time performance but also limits the potential of TTA algorithms in latency-sensitive applications.

In online TTA scenarios that require rapid response to incoming data streams on resource-constrained devices, *Sparse TTA (STTA)*, which adapts occasionally rather than at every batch, can offer a practical solution by reducing the adaption overhead. However, naïve STTA may result in performance degradation as it utilizes far less data (e.g., 0.1) for model adaptation (Figure 1). The

Figure 1: Comparison of average latency per batch and classification accuracy between the Original TTA and Sparse TTA approaches on edge devices processing an online data stream. With an adaptation rate of 0.33, adaptation occurs once every three batches, reducing latency relative to the adaptation rate but leading to a significant accuracy drop than fully adapting original TTA.

effectiveness of STTA hinges on selecting proper samples from a large pool, ensuring that the model maintains adequate performance with fewer updates (detailed analysis in Section 4).

Conventional TTA approaches that adopt sampling strategies are designed for non-i.i.d data (Gong et al., 2022; Niu et al., 2023; Yuan et al., 2023) or noisy data (Gong et al., 2023). They do not aim for data efficiency and thus yield high sample usage for updates. While EATA (Niu et al., 2022) excludes unreliable samples and utilizes fewer samples, it suffers from performance degradation when attempting more aggressive reductions. Data-efficient deep learning demonstrated that selecting easy, class-representative samples is effective when the sampling ratio is low (e.g., below 0.4) (Xia et al., 2022; Choi et al., 2024). However, these methods rely on ground-truth label information, which is typically unavailable in TTA scenarios.

We propose **SNAP-TTA**: **S**parse **N**etwork **A**daptation for **P**ractical **T**est-**T**ime **A**daptation, a low-latency TTA framework designed for resource-constrained devices. SNAP-TTA addresses the challenge of balancing adaptation accuracy with computational efficiency in STTA, where only a small subset of data is used for updates. To that end, SNAP-TTA has two key technical enablers: First, it introduces a sampling strategy that combines *class-representative* and *domain-representative* samples. This approach enables the model to adapt effectively to domain shifts even with minimal data. Class and Domain Representative Memory (CnDRM) selects these critical samples by using pseudo-label confidence in a prediction-balanced manner for class-representative samples, and by identifying the domain-representative samples closest to the center of the target domain's feature embedding (Section 3.1). Second, Inference-only Batch-aware Memory Normalization (IoBMN) refines the normalization process during inference by utilizing CnDRM's class-domain representative statistics, leveraging the representativeness of these selected samples to correct skewed feature distributions at each inference step. This ensures that the model effectively adapts to domain shifts without back-propagation, maintaining alignment with the evolving data distribution (Section 3.2). These two components are integrated to perform adaptation, minimizing accuracy drop and latency in real-world domain-shifted scenarios.

SNAP-TTA is designed to work together with existing TTA methods orthogonally; thus, we evaluated SNAP-TTA integrated with existing SOTA TTA algorithms under diverse adaptation rates. Specifically, we evaluated SNAP-TTA with five SOTA TTA algorithms (Tent(Wang et al., 2021), EATA(Niu et al., 2022), SAR(Niu et al., 2023),CoTTA(Wang et al., 2022), and RoTTA(Yuan et al., 2023)) on three common TTA benchmarks (CIFAR10-C, CIFAR100-C (Hendrycks & Dietterich, 2019a), and ImageNet-C (Hendrycks & Dietterich, 2019b)). SNAP-TTA effectively reduces latency while minimizing performance drops in existing TTA methods. For instance, on our implementation in Raspberry Pi 4(Raspberry Pi Foundation, 2019) testbed, SNAP-TTA achieved up to 87.5% latency reduction at an adaptation rate of 0.1. In CIFAR10-C, SNAP-TTA-integrated methods consistently outperformed their original counterparts, showing up to 13.38% accuracy gain for CoTTA at an adaptation rate of 0.01. In addition, SNAP-TTA integration performed comparable accuracy to the original TTA methods under full adaptation settings. For instance, it achieved 77.12%~81.74% accuracy for Tent at various adaptation rates, whereas the full adaptation accuracy was 80.43% in CIFAR10-C.

## 2 PRELIMINARIES

We focus on the Test-Time Adaptation (TTA) latency challenges specific to edge devices, highlighting the constraints of adapting models in real-time environments with limited resources. Detailed related works are in Appendix A.

**Test-Time Adaptation and Its Latency Challenge on Edge Devices.** In unsupervised domain adaptation, the source domain data $\mathcal{D}_\mathcal{S} = \mathcal{X}^\mathcal{S}, \mathcal{Y}$ is drawn from the distribution $P_\mathcal{S}(\mathbf{x}, y)$, while the target domain data $\mathcal{D}_\mathcal{T} = \mathcal{X}^\mathcal{T}, \mathcal{Y}$ follows $P_\mathcal{T}(\mathbf{x}, y)$, typically without known labels $y_j$. Given a pre-trained model $f(\cdot; \Theta)$ on the source domain $\mathcal{D}_\mathcal{S}$, test-time adaptation (TTA) (Wang et al., 2021) adjusts the model to the target distribution $P_\mathcal{T}$ using only target instances $\mathbf{x}_j$, updating the parameters $\Theta$ to reduce domain discrepancy.

When applied to resource-constrained devices, however, current TTA approaches face significant latency challenges. In real-time applications that require rapid inference, online TTA becomes impractical due to the need for adaptation at every batch (Figure 4, detailed latency tracking reported in Appendix E.3). Our experiment on Raspberry Pi 4 (Raspberry Pi Foundation, 2019) showed a minimum of 3.83 seconds latency per batch for existing TTA methods. This indicates existing methods could not handle real-time applications with fast data streams and strict latency requirements, such as autonomous driving (Tampuu et al., 2024; Liu et al., 2023). TTA methods such as CoTTA use computationally intensive operations such as data augmentations and ensemble models at the cost of increased latency. Relatively lightweight algorithms incur non-negligible latency from adaptation processes such as backpropagation, which becomes bottlenecks in resource-constrained devices without the parallel processing capabilities and memory bandwidth of GPUs.

A recent work (Alfarra et al., 2024), recognizing latency as a problem, proposed a TTA evaluation protocol that penalizes methods that are slower than the data stream rate. Instead of penalizing a model for being slow, we utilize Sparse TTA, where the model actively chooses to adapt at sparse intervals for the goal of maintaining a real-time inference rate. As real deployments involve devices with different computational capabilities and data streams of varying speeds, we believe a framework that effectively maintains various TTA methods' performance across different latency requirements is crucial.

**Sparse Test-Time Adaptation and Adaptation rates.** Sparse Test-Time Adaptation (STTA) aims to efficiently adapt models by reducing both the frequency of updates and the number of samples used per update, which is essential for minimizing latency in edge devices. The concept of adaptation rate plays a central role in STTA, as it controls both the update frequency and the number of data points used. Unlike Original Test-Time Adaptation (TTA), which uses full batches of data and can create significant computational overhead, STTA employs an adaptation rate to limit updates and data usage proportionally, thus introducing sparsity (Figure 1).

By adjusting the *adaptation rate*, STTA can minimize latency and computational costs while maintaining adaptation performance. This rate defines how sparsely updates occur and the proportion of samples used for updates compared to the Original TTA, enabling efficient model adjustments to distribution shifts. The balance between adaptation accuracy and computational efficiency makes STTA particularly suitable for environments that demand both quick responses and minimal resource usage.

## 3 METHODOLOGY

SNAP-TTA framework resolves the high latency and inefficiency issue of existing Test-Time Adaptation (TTA) methods. By introducing a Sparse TTA (STTA) strategy combined with a novel sampling method, SNAP-TTA minimizes adaptation delays while maintaining accuracy. The overall system, illustrated in Figure 2, consists of two primary components: (i) Class and Domain Representative Memory (CnDRM) for efficient sampling and (ii) Inference-only Batch-aware Memory Normalization (IoBMN) to correct feature distribution shifts during inference. Together, these components enable effective STTA with minimal computational overhead.

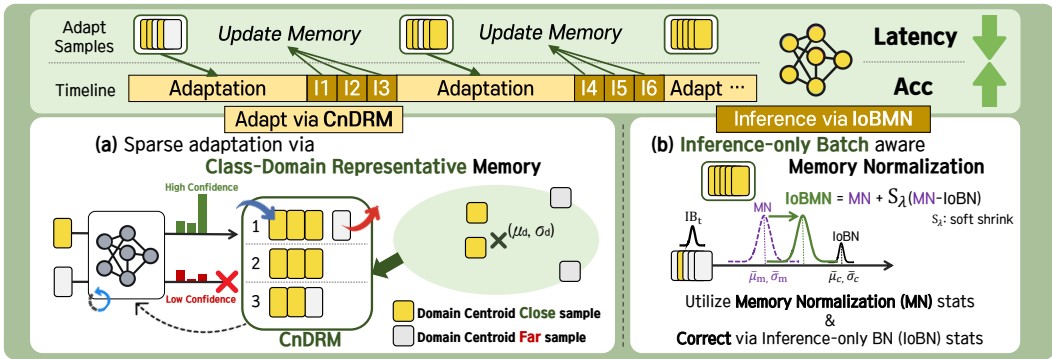

Figure 2: Design overview of SNAP-TTA. The framework consists of two primary components: (a) Class and Domain Representative Memory (CnDRM), which efficiently selects representative samples to minimize adaptation overhead, and (b) Inference-only Batch-aware Memory Normalization (IoBMN), which corrects feature distribution shifts during inference. Together, these components implement the Sparse TTA (STTA) strategy, reducing latency while maintaining model accuracy.

## 3.1 CLASS AND DOMAIN REPRESENTATIVE MEMORY (CNDRM)

CnDRM is a core component of SNAP-TTA that addresses the challenges of efficient data sampling for STTA. In STTA, the adaptation rate directly impacts the number of samples used, necessitating a careful sampling strategy to optimize performance with minimal data. Given this limited sampling ratio, CnDRM selects both class and domain-representative samples to maintain model performance while minimizing adaptation overhead.

**Motivation.** Data sampling is crucial in data-efficient deep learning, especially when working with a limited number of samples. In high data sampling ratio scenarios, score-based methods prioritize difficult or rare samples, often achieving performance comparable to full-dataset training. However, when the sampling ratio is low, selecting easy and class-representative samples becomes more effective (Choi et al., 2024). This method selects samples that minimize differences in loss gradients or curvature, ensuring that the generalizability is retained even with fewer samples. Similarly, the Moderate Coreset (Xia et al., 2022) paper demonstrates that at low sampling ratios of 0.2 to 0.4, the distance from the class center significantly impacts performance, with samples closer to the center being particularly effective in scenarios with high label noise. In the STTA setting, where ground truth labels are unavailable and the probability of incorrect predictions is high, selecting representative samples based on potentially incorrect predictions resembles a high label noise situation. Therefore, selecting class-representative easy samples could provide some benefit to STTA.

However, if the model must perform STTA at an even lower adaptation rate (e.g., 0.1) due to the latency limits, selecting class-representative samples alone would be insufficient (Table 4). Unlike traditional classification tasks, STTA is an unsupervised domain adaptation, which requires identifying target domain-representative samples that reflect the distributional shift between the source and target domains. In these cases, we argue that focusing on domain-representative instances is just as crucial, as selecting samples that best capture the domain shift can help the model retain generalizability with minimal data. Therefore, selecting both **class-representative and domain-representative samples** could enhance STTA performance in low-data environments, where each sample must contribute significantly to model adaptation.

**Critera 1: Class Representation.** CnDRM selects samples with higher confidence scores to avoid the issues caused by low-confidence samples. Low-confidence samples are typically located near decision boundaries and are more likely to carry incorrect pseudo-labels. This strategy ensures that the adaptation process is guided by stable learning signals, which is important in the absence of ground-truth labels. By focusing on high-confidence samples, CnDRM mitigates the risk of propagating errors resulting from incorrect pseudo-labels, thereby supporting more effective and stable adaptation (Details in Appendix E.2). The confidence score $C(\mathbf{x})$ for each sample $\mathbf{x}$ is calculated as: $C(\mathbf{x}) = \max_{y \in \mathcal{Y}} p(y|\mathbf{x}; \Theta)$ where $p(y|\mathbf{x}; \Theta)$ is the softmax probability for class $y$. Only samples with confidence above a predefined threshold $\tau_{conf}$ are retained. For a balanced representa-

tion across diverse classes, CnDRM selects these high-confidence samples in a prediction-balanced manner. This balance helps maintain the model's overall classification capability and prevents bias towards certain classes when only a low sample ratio is available for adaptation. By leveraging both high confidence and prediction balance, CnDRM effectively selects class-representative samples that are diverse and reliable, even without access to ground-truth labels.

**Critera 2: Domain Representation.**
In addition to class-representative sampling, CnDRM selects domain-representative samples to facilitate adaptation to new domain conditions. Building on the efficient class-representative sampling criteria, we argue that *selecting samples close to the domain centroid* would enhance performance in STTA. Our preliminary experiment results validate improved performance when selecting samples near the centroid (Figure 3).

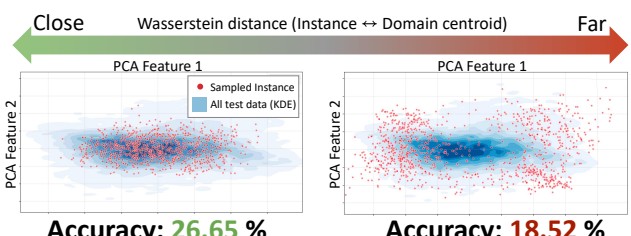

Figure 3: Samping visualization and accuracy comparison between the closest 20% and farthest 20% samples from the domain centroid (based on Wasserstein distance) on ImageNet-C Gaussian noise.

For ImageNet-C Gaussian noise, TTA with the closest 20% of samples achieved 26.65% accuracy, whereas the farthest 20% showed a lower accuracy of 18.52%.

As early layers in deep learning models tend to retain domain-specific features (Zeiler & Fergus, 2014; Lee et al., 2018; Segu et al., 2023), we utilize the hidden features of early layers to identify domain-representative samples (Appendix E.1). We use the feature statistics (mean and variance) of the first normalization layer to evaluate domain representation. This choice is made as domain discrepancies can be effectively reduced through normalization adjustments (Nado et al., 2020; Schneider et al., 2020). Domain discrepancies in hidden features are substantially reduced after passing through a single normalization layer, significantly minimizing domain shift differences (Li et al., 2016). While deeper layers provide detailed information, using the first layer balances capturing domain-specific information and maintaining computational efficiency.

The domain centroid $\mathbf{c}_{domain}$ is computed using a momentum-based update of batch statistics from the normalization layer: $\mu_{domain} \leftarrow (1 - \beta)\mu_{domain} + \beta\mu_t$ and $\sigma^2_{domain} \leftarrow (1 - \beta)\sigma^2_{domain} + \beta\sigma^2_t$, where $\mu_t$ and $\sigma^2_t$ are the mean and variance of the current batch $t$, and $\beta$ is the momentum parameter. In our preliminary study, we found that using only the mean and standard deviation values before the first normalization was sufficient to calculate the domain centroid. The sampled instances effectively represented the domain and were correctly positioned in the embedding space for each criterion (Figure 3).

To determine domain-representative samples, CnDRM calculates the Wasserstein distance between each sample's feature statistics and the domain centroid. The Wasserstein distance measures the similarity between two distributions by considering their mean and variance, evaluating how well a sample represents the domain. It is useful for capturing domain characteristics, leading to its wide use in domain generalization (Segu et al., 2023). For each sample $\mathbf{x_t}$, the feature statistics $(\mu_{\mathbf{x_t}}, \sigma_{\mathbf{x_t}})$ are taken from the input to the normalization layer, and the Wasserstein distance $W(\mathbf{x_t}, \mathbf{c}_{domain})$ is given by:

$$W(\mathbf{x_t}, \mathbf{c}_{domain}) = \sqrt{(\mu_{\mathbf{x_t}} - \mu_{domain})^2 + (\sigma_{\mathbf{x_t}} - \sigma_{domain})^2}. \tag{1}$$

**Memory Management Algorithm.** The memory management in CnDRM maintains efficiency without introducing additional overhead. To achieve this, the memory size is kept equal to the batch size for minimal resource usage. Within this fixed memory, samples are managed by balancing the number of samples per class based on predictions so that each class remains well-represented. For domain adaptation, samples in memory are periodically replaced with new samples that are closer to the domain centroid and meet the confidence threshold to retain only the most class-domain representative samples. Algorithm 1 has details.

---

**Algorithm 1** Class and Domain Representative Memory (CnDRM)

---

**Require:** test data stream $x_t$, memory $M$ with capacity $N$, confidence threshold $\tau_{conf}$, sample unit for memory $\mathbf{s}$, adaptation rate $1/k$
1: **for** batch $b \in \{1, \dots, B\}$ **do**
2:      $\hat{Y}_b \leftarrow f(b; \Theta)$
3:      **for** each sample $x_t$ in batch $b$ **do**
4:          $\hat{y}_t \leftarrow \hat{Y}_b[t]$
5:          confidence $\leftarrow C(x_t; \Theta)$
6:          $\mathbf{c}_t(\mu_{\mathbf{x_t}}, \sigma_{\mathbf{x_t}}) \leftarrow$ mean and variance of early hidden feature
7:          $w_{x_t} \leftarrow W(x_t, \mathbf{c}_{domain})$
8:          **if** confidence $> \tau_{conf}$ **then**                 ▷ Class-representative samples
9:             Add $\mathbf{s}_t(x_t, \hat{y}_t, c_t, w_{x_t})$ to $M$        ▷ Add samples in prediction-balanced manner
10:             **if** $|M| > N$ **then**
11:                 $L^* \leftarrow$ class with most samples in $M$
12:                 **if** $\hat{y}_t \notin L^*$ **then**          ▷ Removes domain-centroid farthest sample
13:                     $\mathbf{s}_{\text{max\_dist}} \leftarrow \arg\max_{\mathbf{s}_i \in M \wedge \hat{y}_i \in L^*} w_{x_i}$
14:                 **else**
15:                     $\mathbf{s}_{\text{max\_dist}} \leftarrow \arg\max_{\mathbf{s}_i \in M \wedge \hat{y}_i = \hat{y}_t} w_{x_i}$
16:                 Remove $\mathbf{s}_{\text{max\_dist}}$ from $M$
17:      $\mathbf{c}_{domain} \leftarrow (1 - \beta)\mathbf{c}_{domain} + \beta\mathbf{c}_t$               ▷ Update domain-centroid
18:      Recalculate $w_{s_i}$ for all $\mathbf{s}_i$ in $M$
19:      **if** $b \mod k == 0$ **then**             ▷ Adaptation occurs every $k$ batches
20:          Update model $\Theta$ using samples in $M$

---

## 3.2 Inference-only Batch-aware Memory Normalization (IoBMN)

**Motivation.** In Sparse Test-Time Adaptation (STTA) scenarios, models must adapt to domain shifts despite having limited opportunities for updates. In this setting, maintaining robust performance becomes challenging as the stored memory statistics, derived from representative adaptation batches, may not fully align with subsequent inference batches, especially when updates are skipped. This can lead to a potential mismatch between the stored statistics and the current data distribution. Traditional normalization methods, which solely rely on test batches' statistics, struggle to address these subtle shifts effectively. To tackle this issue, we introduce the Inference-only Batch-aware Memory Normalization (IoBMN) module, which leverages the robustness of class-domain representative statistics while dynamically adjusting for mismatches that arise in skipped batches. By primarily basing normalization on stable, representative memory statistics and selectively adapting with recent inference data, IoBMN efficiently corrects for distributional shifts, ensuring both robustness and adaptability in STTA conditions. This approach significantly enhances model stability in sparse adaptation scenarios, as shown in our ablation study in Section 4.

**Approach.** Given a feature map $f \in \mathbb{R}^{B \times C \times L}$, where $B$ is the batch size, $C$ is the number of channels, and $L$ is the number of spatial locations, the batch-wise statistics $\bar{\mu}_c$ and $\bar{\sigma}_c^2$ for the $c$-th channel are calculated as follows:

$$\bar{\mu}_c = \frac{1}{B \times L} \sum_{b=1}^{B} \sum_{l=1}^{L} f_{b,c,l}, \quad \bar{\sigma}_c^2 = \frac{1}{B \times L} \sum_{b=1}^{B} \sum_{l=1}^{L} (f_{b,c,l} - \mu_{b,c}), \quad (2)$$

where $\bar{\mu}_m$ and $\bar{\sigma}_m^2$ are calculated from the most recent adapted CnDRM samples in the same way with Equation 2, using the memory capacity $M$ with $m$ representing the memory. We assume that $\mu_m$ and $\sigma_m^2$ follow the *sampling distribution* of the feature map size $L$ and memory capacity $M$. The corresponding variances for the memory mean $\mu_m$ and variance $\sigma_m^2$ are calculated as:

$$s_{\mu_m}^2 := \frac{\bar{\sigma}_m^2}{C \times M}, \quad s_{\sigma_m^2}^2 := \frac{2\bar{\sigma}_m^4}{C \times M - 1}. \quad (3)$$

For the normalization process to adapt efficiently to the current inference batch statistics, IoBMN corrects $(\bar{\mu}_m, \bar{\sigma}_m^2)$ only when $\bar{\mu}_c$ (and $\bar{\sigma}_c^2$) significantly differ from $\bar{\mu}_m$ (and $\bar{\sigma}_m^2$) through soft shrinkage function:

$$\mu_m^{\text{IoBMN}} = \bar{\mu}_m + S_\lambda(\bar{\mu}_c - \bar{\mu}_m; \alpha s_{\mu_m}), \quad (\sigma_m^{\text{IoBMN}})^2 = \bar{\sigma}_m^2 + S_\lambda(\bar{\sigma}_c^2 - \bar{\sigma}_m^2; \alpha s_{\sigma_m^2}), \quad (4)$$

where $\alpha \geq 0$ in IoBMN controls the reliance on the normalization layer statistics. A larger $\alpha$ gives more weight to the last adapted memory normalization statistics, whereas a smaller $\alpha$ emphasizes the current inference batch normalization statistics. The soft shrinkage function $S_\lambda(x; \lambda)$ is defined as:

$$S_\lambda(x; \lambda) = \begin{cases} x - \lambda & \text{if } x > \lambda, \\ x + \lambda & \text{if } x < -\lambda, and \\ 0 & \text{otherwise,} \end{cases}$$

where $\lambda$ is the threshold, $s$ is a scaling factor, and $x$ is the input. The function allows for proportional adjustments based on the magnitude of the values, where smaller values are adjusted less and larger values more, preserving the critical information inherent in the adapted memory normalization statistics.

Finally, the output of the IoBMN for each feature $f_{b,c,l}$ is computed as:

$$\text{IoBMN}(f_{b,c,l}; \bar{\mu}_m, \bar{\sigma}_m^2, \mu_m^{\text{IoBMN}}, (\sigma_m^{\text{IoBMN}})^2) := \gamma \cdot \frac{f_{b,c,l} - \mu_m^{\text{IoBMN}}}{\sqrt{(\sigma_m^{\text{IoBMN}})^2 + \epsilon}} + \beta, \qquad (5)$$

where $\gamma$ and $\beta$ are learnable affine parameters of normalization layer, and $\epsilon$ is a small constant added for numerical stability. In our experiments, we chose $\alpha = 4$ to effectively handle various out-of-distribution scenarios. The parameter $s$ is a hyperparameter that determines the degree of adjustment desired and can be tuned based on specific requirements.

IoBMN utilizes CnDRM's class-domain representative statistics and adjusts them based on the current inferencing batch statistics. This dual-statistic approach allows IoBMN to correct the outdated and skewed distribution of the memory, ensuring alignment with the data distribution at each inference point. By leveraging the statistics of the data used during model update points, IoBMN adapts effectively without significant computational overhead. Additionally, this method mitigates the performance degradation caused by the prolonged intervals between adaptations so that the model remains well-aligned with the evolving data distribution.

## 4 EXPERIMENTS

This section outlines our experimental setup and presents the results obtained under various STTA settings. Refer to Appendix B for further details.

**Scenario.** We examined how different adaptation rates affect performance to simulate a scenario requiring a certain latency threshold for latency-sensitive applications. We varied the *adaptation rate* to observe its impact on both model accuracy and latency. The main evaluation was run with diverse adaptation rates (0.01, 0.03, 0.05, 0.1, 0.3, and 0.5). We report the average accuracy and standard deviation from three random seeds. Latency measurement was done on our Raspberry Pi 4 (Raspberry Pi Foundation, 2019) testbed.

**Dataset and Model.** We used three standard TTA benchmarks: **CIFAR10-C**, **CIFAR100-C** (Hendrycks & Dietterich, 2019a) and **ImageNet-C** (Hendrycks & Dietterich, 2019b). These datasets include 15 different types of corruption with five levels of severity, and we used the highest one. CIFAR10-C/CIFAR100-C has 10,000 test data with 10/100 classes, and ImageNet-C has 50,000 test data with 1,000 classes for each corruption. We employed **ResNet18** (He et al., 2016) as the backbone network, utilizing models pre-trained on CIFAR10 and CIFAR100 (Krizhevsky & Hinton, 2009). We also use **ResNet50** (He et al., 2016) and **ViT** (Dosovitskiy, 2020) pre-trained on ImageNet (Deng et al., 2009) from the TorchVision (maintainers & contributors, 2016) library.

**Baselines.** SNAP-TTA is designed to integrate with existing TTA algorithms. Therefore, testing existing *TTA algorithms under different adaptation rates* serves as our baseline (implementation details including hyperparameters are in Appendix B.1). We selected five SOTA TTA algorithms: (i) **Tent** (Wang et al., 2021) updates only BN affine parameters, (ii) **CoTTA** (Wang et al., 2022) updates the entire model parameters using a teacher-student framework, (iii) **EATA** (Niu et al., 2022), (iv) **SAR**(Niu et al., 2023), and (v) **RoTTA**(Yuan et al., 2023). For efficiency evaluation, we compared our method against **BN stats** (Nado et al., 2020; Schneider et al., 2020).

Table 1: STTA classification accuracy (%) and latency per batch (s) comparing with and without SNAP-TTA on ImageNet-C through Adaptation Rates (AR) (0.3, 0.1, and 0.05).AR is the ratio of the number of backpropagation occurrences to the total, and thus represents the reduction in adaptation latency compared to full adaptation (AR=1). More results on diverse AR (0.5, 0.03 and 0.01) are on Appendix C.1. **Bold** numbers are the highest accuracy.

| AR | Methods | Gau. | Shot | Imp. | Def. | Gla. | Mot. | Zoom | Snow | Fro. | Fog | Brit. | Cont. | Elas. | Pix. | JPEG | Avg. | Lat. |
|---|---|---|---|---|---|---|---|---|---|---|---|---|---|---|---|---|---|---|
| 1 | Source | 3.00 | 3.70 | 2.64 | 17.90 | 9.74 | 14.72 | 22.45 | 16.60 | 23.06 | 24.00 | 59.11 | 5.37 | 16.50 | 20.88 | 32.63 | 18.15 | 16.60 |
| | BN stats | 14.29 | 15.06 | 14.89 | 13.30 | 13.38 | 23.78 | 35.22 | 31.78 | 30.26 | 44.40 | 62.39 | 15.14 | 40.42 | 45.25 | 36.53 | 29.00 | 17.36 |
| | Tent | 27.03 | 28.98 | 28.64 | 24.66 | 23.63 | 38.70 | 45.77 | 44.82 | 38.06 | 54.59 | 64.61 | 16.84 | 51.64 | 55.54 | 49.38 | 39.53 | 38.33 |
| | CoTTA | 13.12 | 13.98 | 13.94 | 12.44 | 12.18 | 23.74 | 35.22 | 31.78 | 30.26 | 44.40 | 62.40 | 15.13 | 40.42 | 45.26 | 36.53 | 28.72 | 300.23 |
| | EATA | 29.62 | 31.79 | 31.17 | 26.89 | 26.30 | 40.65 | 47.44 | 46.29 | 40.78 | 55.57 | 64.97 | 38.02 | 52.66 | 56.03 | 50.26 | 42.56 | 31.98 |
| | SAR | 17.49 | 22.04 | 21.21 | 11.62 | 12.60 | 39.76 | 44.13 | 45.98 | 29.39 | 55.13 | 63.71 | 17.34 | 52.31 | 56.09 | 49.35 | 35.21 | 78.15 |
| | RoTTA | 20.60 | 22.83 | 19.81 | 10.46 | 10.10 | 21.31 | 31.83 | 39.66 | 32.09 | 46.08 | 62.22 | 20.27 | 42.54 | 47.47 | 40.67 | 31.20 | 87.00 |
| 0.3 | Tent | 23.63 | 25.18 | 24.80 | 21.81 | 20.97 | 34.11 | 43.60 | 41.44 | 36.98 | 52.66 | 64.21 | 22.74 | **48.96** | 53.46 | 46.80 | 37.42 | 27.34 |
| | + SNAP | 26.60 | 28.21 | 27.94 | 24.37 | 22.39 | 36.45 | 44.36 | 42.64 | 38.54 | 52.91 | 64.26 | 33.47 | 48.58 | **53.90** | 47.41 | **39.47** | 28.84 |
| | CoTTA | 11.74 | 12.74 | 12.68 | 11.77 | 11.62 | 22.64 | 34.97 | 31.05 | 29.81 | 44.24 | 62.12 | 13.73 | 40.31 | 45.19 | 36.71 | 28.09 | 205.22 |
| | + SNAP | 15.26 | 16.00 | 15.83 | 13.81 | 14.13 | 24.84 | 36.46 | 32.58 | 31.73 | 46.04 | 63.52 | 15.69 | 42.18 | 46.74 | 38.00 | 30.19 | 208.10 |
| | EATA | 27.35 | 29.03 | 28.62 | 23.94 | 23.45 | 37.21 | 46.18 | 44.05 | 39.19 | 54.52 | 64.54 | 32.20 | 51.22 | 55.00 | 49.27 | 40.38 | 20.27 |
| | + SNAP | 29.48 | 31.20 | 30.69 | 26.68 | 25.90 | 38.24 | 46.60 | 44.62 | 39.31 | 54.82 | 64.44 | 32.87 | 51.41 | 55.41 | 49.78 | 41.43 | 22.16 |
| | SAR | 28.12 | 29.30 | 29.63 | 22.37 | 23.88 | 39.34 | 45.36 | 45.69 | 36.73 | 54.11 | 64.11 | 10.96 | 52.22 | 55.76 | 49.60 | 39.20 | 36.44 |
| | + SNAP | 32.63 | 34.69 | 34.26 | 28.91 | 27.96 | 43.51 | 47.79 | 48.27 | 42.41 | 56.45 | 64.77 | 32.76 | 53.74 | 57.21 | 51.67 | 43.80 | 38.01 |
| | RoTTA | 16.90 | 17.88 | 17.25 | 12.89 | 12.51 | 23.96 | 35.26 | 36.26 | 32.32 | 47.25 | 63.98 | 17.46 | 42.77 | 48.21 | 39.35 | 30.95 | 59.32 |
| | + SNAP | 18.63 | 19.94 | 19.35 | 14.88 | 14.34 | 25.88 | 36.47 | 37.13 | 33.32 | 47.74 | 63.96 | 19.08 | 42.98 | 48.73 | 40.27 | 32.18 | 60.31 |
| 0.1 | Tent | 22.00 | 23.51 | 23.07 | 19.38 | 18.86 | 32.15 | 42.29 | 39.70 | 34.33 | 51.62 | 63.70 | 15.79 | 47.74 | 52.35 | 45.54 | 35.47 | 18.01 |
| | + SNAP | 26.21 | 27.85 | 27.50 | 23.62 | 22.73 | 36.01 | 44.11 | 42.19 | 38.15 | 52.95 | 64.57 | 30.23 | 48.56 | 53.71 | 47.09 | 39.03 | 18.76 |
| | CoTTA | 10.97 | 11.92 | 11.98 | 11.45 | 11.48 | 22.39 | 34.96 | 30.88 | 29.89 | 44.00 | 61.96 | 13.08 | 40.20 | 45.27 | 36.71 | 27.81 | 161.98 |
| | + SNAP | 15.13 | 16.03 | 15.91 | 13.86 | 14.02 | 24.90 | 36.51 | 32.56 | 31.81 | 46.02 | 63.60 | 15.69 | 41.94 | 46.78 | 38.03 | 30.19 | 163.24 |
| | EATA | 22.43 | 23.78 | 23.26 | 19.38 | 19.42 | 32.18 | 43.22 | 40.65 | 36.64 | 52.38 | 63.87 | 24.59 | 48.13 | 52.89 | 46.33 | 36.61 | 16.00 |
| | + SNAP | 26.10 | 27.29 | 27.13 | 22.38 | 22.15 | 33.45 | 43.92 | 40.96 | 36.68 | 52.71 | 63.77 | 27.93 | 48.47 | 53.23 | 47.46 | 38.24 | 17.45 |
| | SAR | 26.12 | 27.56 | 26.93 | 22.51 | 23.35 | 36.03 | 44.48 | 43.19 | 37.26 | 53.82 | 64.15 | 19.87 | 50.78 | 54.78 | 48.43 | 38.62 | 21.39 |
| | + SNAP | 30.28 | 31.97 | 31.30 | 26.67 | 26.31 | 39.66 | 46.08 | 45.43 | 40.26 | 54.76 | 64.42 | 36.12 | 51.26 | 55.42 | 49.63 | 41.99 | 23.99 |
| | RoTTA | 14.77 | 15.59 | 15.33 | 13.17 | 13.19 | 23.85 | 35.38 | 32.73 | 30.77 | 45.22 | 63.08 | 15.62 | 41.05 | 46.15 | 37.19 | 29.54 | 45.98 |
| | + SNAP | 15.35 | 16.20 | 16.01 | 13.67 | 13.66 | 24.27 | 35.62 | 33.04 | 31.02 | 45.38 | 62.95 | 15.96 | 41.06 | 46.17 | 37.44 | 29.85 | 47.47 |
| 0.05 | Tent | 23.77 | 24.65 | 24.44 | 20.54 | 20.27 | 32.73 | 43.82 | 39.70 | 34.94 | 51.91 | 63.82 | 15.95 | 40.83 | 53.46 | 47.19 | 36.62 | 16.93 |
| | + SNAP | 29.12 | 30.46 | 30.30 | 25.77 | 25.22 | 38.21 | 46.14 | 44.29 | 39.95 | 54.65 | 65.47 | 33.81 | 50.83 | 55.59 | 49.21 | 41.27 | 17.55 |
| | CoTTA | 11.03 | 11.91 | 11.75 | 11.03 | 11.20 | 22.30 | 34.98 | 30.87 | 29.78 | 43.99 | 61.87 | 12.92 | 40.26 | 45.23 | 36.63 | 27.72 | 152.94 |
| | + SNAP | 15.22 | 15.97 | 15.93 | 13.91 | 14.05 | 24.87 | 36.48 | 32.60 | 31.65 | 46.09 | 63.59 | 15.67 | 42.00 | 46.71 | 37.96 | 30.18 | 153.34 |
| | EATA | 19.53 | 20.65 | 20.72 | 16.74 | 16.96 | 29.11 | 41.22 | 37.96 | 34.84 | 50.75 | 63.29 | 19.86 | 45.92 | 51.15 | 44.13 | 34.19 | 15.82 |
| | + SNAP | 22.83 | 23.95 | 23.62 | 19.43 | 19.70 | 30.34 | 41.59 | 38.06 | 35.06 | 50.98 | 63.30 | 23.72 | 46.26 | 51.52 | 45.46 | 35.72 | 16.44 |
| | SAR | 23.25 | 24.23 | 23.66 | 19.98 | 20.38 | 33.05 | 43.04 | 40.73 | 36.06 | 52.61 | 64.09 | 20.17 | 49.00 | 53.35 | 46.73 | 36.69 | 19.98 |
| | + SNAP | 27.54 | 29.03 | 28.66 | 24.05 | 23.42 | 36.28 | 44.12 | 42.89 | 38.54 | 53.24 | 64.25 | 31.83 | 48.79 | 54.04 | 47.80 | 39.63 | 20.94 |
| | RoTTA | 14.42 | 15.22 | 15.02 | 13.25 | 13.31 | 23.79 | 35.27 | 32.09 | 30.43 | 44.71 | 62.64 | 15.24 | 40.63 | 45.55 | 36.75 | 29.22 | 43.32 |
| | + SNAP | 14.65 | 15.48 | 15.29 | 13.43 | 13.45 | 23.93 | 35.33 | 32.18 | 30.53 | 44.71 | 62.58 | 15.41 | 40.64 | 45.55 | 36.81 | 29.33 | 44.71 |

Table 2: STTA classification accuracy (%) and latency per batch (s) comparing with and without SNAP-TTA on CIFAR10/100-C at Adaptation Rate 0.1. Numbers in parentheses represent the performance difference of SNAP-TTA compared to full adaptation **Bold** numbers are the highest accuracy. More results on other adaptation rates are in Appendix C.2 and C.3.

| Methods | Gau. | Shot | Imp. | Def. | Gla. | Mot. | Zoom | Snow | Fro. | Fog | Brit. | Cont. | Elas. | Pix. | JPEG | Avg. | Lat. |
|---|---|---|---|---|---|---|---|---|---|---|---|---|---|---|---|---|---|
| | | | | | | | CIFAR10-C | | | | | | | | | | |
| Tent | 67.32 | 69.39 | 60.69 | 85.34 | 63.82 | 83.52 | 84.70 | 79.68 | 77.79 | 83.75 | 88.53 | 83.12 | 75.18 | 77.82 | 71.47 | 76.81 (-3.62) | 2.80 (-29.47%) |
| + SNAP | 70.22 | 71.48 | 63.08 | 87.35 | 65.74 | 85.89 | 86.38 | 81.93 | 80.00 | 85.62 | 90.34 | 87.47 | 76.44 | 79.63 | 72.72 | 78.95 (1.48) | 3.08 (-22.42%) |
| CoTTA | 59.11 | 60.26 | 56.07 | 72.23 | 56.77 | 73.55 | 72.20 | 68.05 | 66.68 | 72.88 | 77.66 | 65.95 | 65.67 | 64.12 | 65.16 | 66.42 (-11.58) | 4.92 (-93.14%) |
| + SNAP | 71.70 | 73.54 | 66.70 | 85.16 | 66.83 | 84.30 | 84.88 | 81.02 | 80.61 | 84.20 | 89.84 | 81.71 | 76.60 | 79.66 | 75.71 | 78.83 (+0.83) | 4.93 (-93.12%) |
| EATA | 66.65 | 68.96 | 59.73 | 84.93 | 63.26 | 83.10 | 84.53 | 79.28 | 77.46 | 83.48 | 88.12 | 82.46 | 74.49 | 77.48 | 70.43 | 76.29 (-5.27) | 2.52 (-35.86%) |
| + SNAP | 69.29 | 70.49 | 61.71 | 87.32 | 65.48 | 85.96 | 86.64 | 81.44 | 79.56 | 85.47 | 90.50 | 86.84 | 76.32 | 79.64 | 72.51 | 78.61 (-2.95) | 2.87 (-26.97%) |
| SAR | 66.11 | 68.18 | 59.15 | 84.91 | 62.87 | 82.33 | 84.27 | 79.23 | 77.58 | 83.21 | 88.29 | 82.60 | 74.65 | 75.92 | 70.79 | 76.01 (-3.04) | 2.85 (-50.43%) |
| + SNAP | 67.76 | 70.68 | 60.82 | 86.78 | 64.73 | 85.29 | 86.22 | 80.82 | 79.30 | 84.95 | 91.33 | 86.59 | 75.72 | 78.72 | 71.24 | 78.06(-0.99) | 2.98 (-48.17%) |
| RoTTA | 63.12 | 64.84 | 56.72 | 84.49 | 62.15 | 82.53 | 83.84 | 78.03 | 76.13 | 82.88 | 87.48 | 81.49 | 73.75 | 76.04 | 68.24 | 74.78 (-2.22) | 2.91 (-50.93%) |
| + SNAP | 65.35 | 66.99 | 58.09 | 86.77 | 63.63 | 85.47 | 86.01 | 80.54 | 78.38 | 84.99 | 90.00 | 85.99 | 75.67 | 78.14 | 70.09 | 77.07 (+0.07) | 2.94 (-50.42%) |
| | | | | | | | CIFAR100-C | | | | | | | | | | |
| Tent | 43.55 | 44.25 | 37.95 | 62.56 | 41.80 | 59.45 | 62.13 | 53.04 | 51.60 | 56.76 | 64.60 | 61.19 | 51.01 | 56.42 | 49.58 | 52.84 (-2.92) | 3.34 (-27.49%) |
| + SNAP | 46.51 | 47.68 | 39.92 | 65.39 | 44.14 | 63.29 | 64.53 | 55.20 | 55.55 | 59.71 | 68.05 | 64.90 | 53.91 | 59.28 | 49.58 | 55.84 (+0.08) | 3.67 (-19.17%) |
| CoTTA | 28.53 | 29.53 | 26.45 | 42.19 | 30.34 | 44.69 | 41.88 | 34.44 | 33.93 | 39.03 | 45.49 | 31.17 | 37.25 | 36.17 | 36.84 | 35.86 (-13.53) | 4.94 (-93.40%) |
| + SNAP | 41.72 | 42.62 | 37.46 | 58.43 | 41.24 | 57.33 | 57.96 | 50.34 | 51.17 | 52.29 | 63.59 | 51.32 | 49.68 | 54.78 | 47.89 | 50.52 (+1.13) | 4.95 (-93.38%) |
| EATA | 38.41 | 39.03 | 32.29 | 61.07 | 38.45 | 58.21 | 60.62 | 49.59 | 49.19 | 54.23 | 62.88 | 57.39 | 49.00 | 53.01 | 42.05 | 49.70 (-1.04) | 3.13 (-27.17%) |
| + SNAP | 40.62 | 41.53 | 34.31 | 64.08 | 40.29 | 61.32 | 63.04 | 52.00 | 51.77 | 56.65 | 65.98 | 61.96 | 51.05 | 55.67 | 44.80 | 52.35 (+1.61) | 3.51 (-17.50%) |
| SAR | 43.92 | 45.28 | 38.64 | 63.36 | 42.58 | 60.36 | 62.78 | 53.39 | 52.23 | 57.54 | 65.41 | 60.88 | 52.07 | 56.80 | 47.16 | 53.49 (-4.45) | 2.95 (-56.16%) |
| + SNAP | 46.29 | 47.60 | 39.95 | 65.26 | 44.00 | 63.09 | 64.97 | 55.08 | 55.17 | 59.73 | 68.13 | 64.72 | 53.84 | 58.98 | 49.54 | 55.76 (-2.18) | 3.09 (-53.73%) |
| RoTTA | 36.28 | 37.12 | 31.38 | 61.20 | 38.36 | 58.26 | 60.30 | 49.20 | 48.21 | 53.54 | 62.80 | 56.78 | 49.61 | 52.28 | 41.26 | 49.11 (-2.44) | 2.96 (-55.92%) |
| + SNAP | 37.83 | 38.42 | 32.38 | 63.73 | 39.72 | 61.32 | 62.58 | 51.38 | 51.18 | 55.61 | 65.70 | 61.39 | 51.36 | 54.51 | 42.85 | 51.33 (-0.22) | 2.99 (-55.41%) |

**Overall performance across various adaptation rates.** Table 1, 2 and Appendix C summarize the performance comparison of baseline state-of-the-art (SOTA) TTA methods and SNAP-TTA integration across various adaptation rates (0.01 to 0.5) on CIFAR10/100-C and ImageNet-C. These results reveal that while Sparse TTA achieves a substantial reduction in adaptation latency up to 87.5% conventional SOTA algorithms suffer significant accuracy degradation under sparse adaptation settings (Table 3, Figure 4). In contrast, SNAP-TTA demonstrates a robust ability to mitigate this performance drop. Leveraging minimal updates with only a few samples, SNAP-TTA consistently outperforms baseline methods and shows competitive accuracy even when compared to fully adapted models. Furthermore, in certain scenarios, SNAP-TTA achieves accuracy gains over the

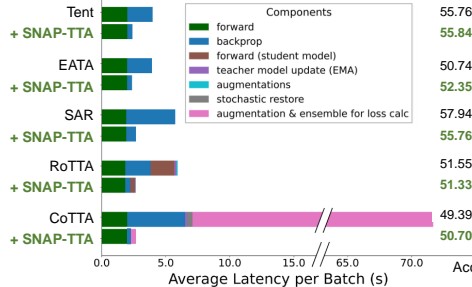

Figure 4: Latency and accuracy comparison of original TTA methods and their SNAP-TTA integration on CIFAR100-C. SNAP-TTA significantly enhances the efficiency.

Table 3: Latency reduction and accuracy gaps of SNAP-TTA (adaptation rate 0.1) compared by original TTA, tested on Raspberry Pi 4. Performance averaged over 15 CIFAR10-C corruptions. Numbers in parentheses represent the performance difference of SNAP-TTA compared to full adaptation.

| Methods | Latency per batch (s) | | Accuracy (%) | |
|---|---|---|---|---|
| | Original TTA | **SNAP-TTA** | naïve STTA | **SNAP-TTA** |
| Tent | 3.97 | **2.20 (-44.0%)** | 76.81 (-3.62) | **78.95 (-1.48)** |
| CoTTA | 71.68 | **8.96 (-87.5%)** | 66.42 (-11.58) | **78.83 (+0.83)** |
| EATA | 3.93 | **2.18 (-44.6%)** | 76.29 (-5.27) | **78.61 (-2.95)** |
| SAR | 5.75 | **2.30 (-60.1%)** | 76.01 (-3.04) | **78.06 (-0.99)** |
| RoTTA | 5.93 | **2.25 (-62.0%)** | 74.78 (-2.27) | **77.07 (+0.07)** |

original counterparts, highlighting its adaptability and effectiveness. These results underscore the capability of SNAP-TTA to balance efficiency and performance, providing a significant advantage in sparse adaptation scenarios while maintaining or even enhancing classification accuracy. This validates the effectiveness of utilizing class-domain representative samples in the STTA setting.

Furthermore, Figure 5 shows more computationally complex and latency-intensive methods such as CoTTA tend to have greater performance gain when integrated with SNAP-TTA. This is because methods that update the entire model parameters are more susceptible to the influence of specific adaptation samples, leading to significant performance drops under sparse update conditions, which SNAP-TTA's CnDRM and IoBMN effectively mitigate. In addition, adaptation rates of 0.5 or 0.3, which represent relatively high adaptation frequencies, sometimes can achieves even better performance with SNAP-TTA than the original TTA, despite in the STTA setting. This is likely because the sampling rate was not critically low but rather comparable to that of existing data-efficient methods such as EATA (Niu et al., 2022), allowing SNAP-TTA to achieve performance gains similar to various sampling-based TTA methods (Niu et al., 2022; 2023; Gong et al., 2022; 2023) using fewer yet effective samples. Overall, SNAP-TTA significantly reduced the average latency per batch while effectively maintaining accuracy, highlighting its benefits for resource-constrained environments. More details on all other adaptation rates are reported in Appendix C.

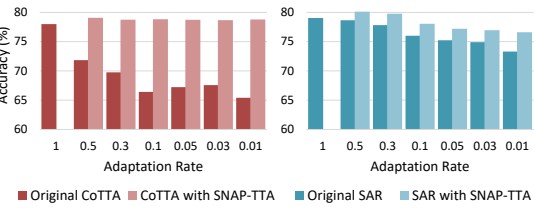

Figure 5: Classification accuracy on CIFAR10-C with varying adaptation rates. SNAP-TTA consistently mitigates accuracy drop across all rates.

Table 4: Classification accuracy (%) comparison of ablative settings on the STTA (adaptation rate 0.1). Performance averaged over 15 CIFAR10-C corruptions.

| Methods | Tent | CoTTA | EATA | SAR | RoTTA |
|---|---|---|---|---|---|
| Naïve | 76.81 | 66.42 | 76.29 | 76.01 | 74.78 |
| Random | 77.08 | 65.61 | 76.59 | 76.33 | 75.01 |
| LowEntropy | 75.66 | 63.19 | 74.89 | 74.41 | 72.60 |
| CRM | 77.77 | 65.71 | 77.18 | 74.36 | 75.27 |
| CnDRM | 77.46 | 77.69 | 77.17 | 76.85 | 75.64 |
| CnDRM+EMA | 78.02 | 72.19 | 77.05 | 76.84 | 76.18 |
| CnDRM+IoBMN | **78.95** | **78.83** | **78.61** | **78.06** | **77.07** |

**Contribution of individual components of SNAP-TTA.** We conducted an ablative evaluation to understand the effects of the individual components of SNAP-TTA (Table 4; more results on diverse adaptation rates and datasets are on Appendix D). CRM denotes prediction-balanced sampling with a confidence threshold (same as the Class-Representative criteria of CnDRM), and CnDRM denotes both Class and Domain Representative sampling (the first component of SNAP-TTA). For inference, the default uses test batch normalization statistics, EMA uses the exponential moving average of the test batch, and IoBMN uses memory samples' statistics corrected to match that of the test batch (the second component of SNAP-TTA).

Contrary to the hypothesis that low-entropy samples are beneficial for TTA (Niu et al., 2022; 2023), LowEntropy performed worse than Rand for STTA. This can be attributed to the limited updates of STTA, resulting in poor or longer convergence times due to low entropy minimization loss. CRM, originally used for data-efficient supervised deep learning (Choi et al., 2024; Xia et al., 2022), performed better than Rand. However, as CRM on TTA inevitably relies on uncertain pseudo labels instead of the ground truth, its performance remains lower than utilizing domain representative features (CnDRM) (note that TTA is unsupervised domain adaptation rather than training from scratch (Xia et al., 2022)). The highest accuracy was achieved when inference was performed us-

Table 5: Classification accuracy (%) on ImageNet-C through Adaptation Rate 0.1 using **ViT**-based model. **Bold** numbers are the highest accuracy.

| Methods | Gau. | Shot | Imp. | Def. | Gla. | Mot. | Zoom | Snow | Fro. | Fog | Brit. | Cont. | Elas. | Pix. | JPEG | Avg. |
|---|---|---|---|---|---|---|---|---|---|---|---|---|---|---|---|---|
| Tent | 40.56 | 41.30 | 41.69 | 35.76 | 31.81 | 42.01 | 38.02 | 44.33 | **53.53** | 20.69 | 72.41 | 30.42 | 45.87 | **51.95** | **56.11** | 43.10 |
| + SNAP-TTA | **40.98** | **41.72** | **42.18** | **37.16** | **32.30** | **42.89** | **38.44** | **46.19** | 52.50 | **53.11** | **72.25** | **39.25** | **46.77** | 51.53 | 55.99 | **46.22** |
| EATA | 20.12 | 21.52 | 21.40 | 20.90 | 23.42 | 15.71 | 18.00 | 16.12 | 28.35 | 22.24 | 35.97 | 11.33 | 19.78 | 20.22 | 19.99 | 21.00 |
| + SNAP-TTA | **40.74** | **43.22** | **43.11** | **40.63** | **44.59** | **51.58** | **50.63** | **54.77** | **58.32** | **61.50** | **73.91** | **33.85** | **60.19** | **63.35** | **63.01** | **52.23** |
| SAR | 21.45 | 23.02 | 23.17 | 23.67 | 24.64 | 15.98 | 14.62 | 7.70 | 31.49 | 8.94 | 41.33 | 6.82 | 17.35 | 22.39 | 22.49 | 20.34 |
| + SNAP-TTA | **37.59** | **38.27** | **36.78** | **38.58** | **39.99** | **49.00** | **45.77** | **43.96** | **56.61** | **59.96** | **73.02** | **19.69** | **54.30** | **61.16** | **61.85** | **47.77** |

ing IoBMN, which primarily utilizes memory statistics and only shifts slightly to the test batch on demand. These results collectively indicate that utilizing CnDRM and IoBMN of SNAP-TTA enhances performance in a low-latency STTA scenario.

**Validation of SNAP-TTA on Vision Transformer (ViT) based Model.** To validate the effectiveness of SNAP-TTA on the Vision Transformer (ViT) (Dosovitskiy, 2020), we conducted experiments on ImageNet-C with adaptation rate of 0.1. Since ViT uses layer normalization (LN), we adjusted CnDRM and IoBMN to use LN from instances, demonstrating that the core concepts of selecting domain-representative samples and mitigating shift in normalization statistics can be applied effectively to a different normalization type (details in Appendix F.3). The results in Table 5 confirm consistent accuracy gains of SNAP-TTA with significant latency decrease, regardless of model and normalization types.

## 5 DISCUSSION AND CONCLUSION

**Limitations and future work.** Our work could be optimized for more realistic data streams, such as continuous domain adaptation scenarios (Appendix F.2). For instance, the adaptation rate can be dynamically altered based on the need for adaptation (i.e., the data distribution just changed). Additionally, while SNAP-TTA employed a fixed confidence threshold in CnDRM as a safeguard to filter noisy samples, its adaptability could be improved. Dynamically adjusting the threshold based on data characteristics presents a promising direction for future research to enhance sampling efficiency and overall performance.

Moreover, while we focused on reducing adaptation latency, memory overhead is another concern. We note that SNAP-TTA introduces negligible additional memory overhead, as detailed in the Appendix E.4, where related analysis and tracking information from real-device experiments are provided. Additionally, we demonstrate in the Appendix E.5 that SNAP-TTA can be effectively used alongside memory-efficient TTA methods such as MECTA (Hong et al., 2023), showcasing its compatibility and practicality. Future works could further explore optimizing SNAP-TTA for both latency and memory.

**Conclusion** We raised the overlooked issue of latency of TTA methods, which is particularly relevant for applications on resource-constrained edge devices. To this end, we propose SNAP-TTA, a Sparse TTA (STTA) framework that could be applied to existing TTA methods to significantly reduce their latency while maintaining competitive accuracy. For effective performance in an STTA setting, we utilize class-domain representative memory of samples for adaptation. Furthermore, we optimize inference by adapting normalization layers using representative samples to account for domain shifts. Extensive experiments and ablative studies demonstrate SNAP-TTA's effectiveness in latency and adaptation accuracy.

## REPRODUCIBILITY STATEMENT

Details of the experiments, including datasets, scenarios, and hyperparameters for reproducibility, are provided in the Appendix B. Additionally, we share the link (https://anonymous.4open.science/r/SNAPTTA-DD0E) of an anonymous repository containing our source code and instructions to validate the reproducibility.

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

## A  RELATED WORK

**Test-time adaptation.**  Test-time adaptation (TTA) aims to improve model performance on Out-of-Distribution (OOD) data by using only the unlabeled test data stream to adapt the model. Test-time normalization (Nado et al., 2020; Schneider et al., 2020) adjusts the batch normalization (BN) statistics using test data to improve performance. Other works mainly involve updating the parameters of the model during test-time. Tent (Wang et al., 2021) adapts the affine parameters of the BN layers to minimize the entropy of its predictions. EATA (Niu et al., 2022) builds upon Tent, sampling reliable and non-redundant samples and utilizing an anti-forgetting regularizer for efficiency. Other works introduce more complex schemes, primarily to improve robustness against more practical test-time scenarios. CoTTA (Wang et al., 2022) addresses a continually changing test-time environment by using weight-averaged and augmentation-averaged predictions with stochastic restoring. SAR (Niu et al., 2023) filters samples with large and noisy gradients to stabilize the model during wilder test-time scenarios. RoTTA (Yuan et al., 2023) targets a practical test-time setting of changing distributions and correlative sampling by introducing a memory bank and a teacher-student model.

**Test-time adaptation on edge devices.**  TTA on edge devices primarily inherit the challenges of on-device learning: limited memory and increased latency from general resource constraints (Lin et al., 2020). Several memory-efficient TTA works have been proposed in this regard. MECTA (Hong et al., 2023) aims to reduce the memory consumption of gradient-based TTA, proposing an adaptive normalization layer to reduce the intermediate caches for backpropagation. Another work EcoTTA (Song et al., 2023) proposes memory-efficient continual TTA by adapting lightweight meta networks instead of the originals to reduce the size of intermediate activations. Despite works to promote memory-efficiency, the latency of TTA, especially on resource-constrained edge devices, has been generally overlooked. While many adaptation-based TTA (Wang et al., 2021; Niu et al., 2022; 2023; Yuan et al., 2023) update only the affine parameters for general time and memory concerns, they still involve computationally-heavy operations every batch, which can lead to high latency on edge devices. A recent work (Alfarra et al., 2024) introduces a more realistic TTA evaluation protocol that penalizes slow TTA methods by providing them with fewer samples for adaptation. We build on from this notion, proposing a sparse TTA setting to reduce the latency of existing TTA methods, but at a minimal cost to performance.

**Data-efficient deep learning.**  Data-efficient deep learning methods enable deep learning models to achieve competitive performance with less data. Among these methods, data selection, or data sampling, involves utilizing a small subset of the training data in an attempt to match that of full-dataset training. A branch of data-selection is score-based selection, which scores each sample based on some predefined metric, such as a sample's influence (Koh & Liang, 2017), difficulty (Toneva et al., 2019; Paul et al., 2021), prediction confidence (Pleiss et al., 2020), or consistency (Jiang et al., 2021), and selects samples with scores in a certain range. Another set of data-selection methods involve optimization-based selection, which formulates an optimization problem to find a optimal subset that can best approximate full-dataset training (Mirzasoleiman et al., 2020; Yang et al., 2023; Pooladzandi et al., 2022). While these approaches work well in their preconceived settings, they generally suffer performance drop as their settings change, such as a change in sampling ratio. More recent works like the Moderate Coreset (Xia et al., 2022) proposes a more robust selection approach by using the distance of a sample to the class center as a score criterion, for an effective representation of the dataset. While our proposed sparse TTA setting is more challenging than the conventional data-efficient setting, as we cannot access ground truths labels nor make assumptions regarding the model, we utilize similar ideas of representative sampling as motivation for our method.

## B  EXPERIMENT DETAILS

All experiments presented in this paper were conducted using three random seeds (0, 1, 2), and we report the average accuracies along with their corresponding standard deviations. To ensure efficiency in experimentation, accuracy measurements were obtained using NVIDIA GeForce RTX 3090 GPUs, as the performance differences attributable to the random seed are negligible. Latency measurements were conducted on a Raspberry Pi 4 (Raspberry Pi Foundation, 2019), equipped with a Quad-core Cortex-A72 (ARM v8) 64-bit SoC @ 1.8GHz CPU and 4GB RAM.

## B.1 BASELINE IMPLEMENTATION DETAILS

In this study, we utilized the official implementations of the baseline methods. To ensure consistency, we adopted the reported best hyperparameters documented in the respective papers or source code repositories as much as possible. Also, we present information about the implementation specifics of the baseline methods and provide a comprehensive overview of our experimental setup, including detailed descriptions of the employed hyperparameters.

We adopt hyperparameters from the original papers or the official code of the baselines for consistency. To assess the generality of SNAP-TTA, the test batch sizes were set to 16 for all baseline methods to ensure a fair comparison. To minimize overhead and maintain consistency with inference batches, we set the size of CnDRM equal to the batch size. TTA is conducted in an online manner, with adaptation or inference performed per batch. When there was a conflict between the implementation of SNAP-TTA and certain components of the existing baseline methods, we prioritized SNAP-TTA's features for fair evaluation at the STTA setting.

For Tent (Wang et al., 2021), we update the BN affine parameters using the SGD optimizer (Loshchilov & Hutter, 2017) with a learning rate of $l = 1e - 3$ for CIFAR10/100C and $l = 1e - 4$ for ImageNet-C. For separate experimentation on the ViT, we used a learning rate of $l = 2e - 4$. For CoTTA (Wang et al., 2022), we update all model parameters using the Adam optimizer (Kingma & Ba, 2015) with a learning rate of $l = 1e - 4$. Furthermore, we set CoTTA's teacher model EMA factor to $\alpha = 0.99$, the restoration factor to $p = 0.1$, and the anchor probability to $p_{\text{th}} = 0.9$. For EATA (Niu et al., 2022), we use the SGD optimizer with a learning rate of $l = 1e - 4$. We set the entropy threshold as $E_0 = 0.4 \times \ln |N|$, where $N$ is the total number of classes. For SAR (Niu et al., 2023), we use SAM (Foret et al., 2021) with the base optimizer as SGD with a learning rate of $l = 1e - 3$. For fair evaluation, we replaced the sample filtering scheme with SNAP-TTA's CnDRM. For RoTTA (Yuan et al., 2023), we use the SGD optimizer with a learning rate of $l = 1e - 3$. For fair evaluation, we replaced RoTTA's RBN and CSTU with SNAP-TTA's CnDRM and IoBMN. For the teacher-student structure, we set the teacher model's exponential moving average update rate as $v = 1e - 3$.

Finally, we list the hyperparameters specific to the components of SNAP-TTA. The confidence threshold for CnDRM $\tau_{conf}$ is set to 0.4 for CIFAR10-C, 0.45 for CIFAR100-C, and 0.5 for ImageNet-C. The entropy threshold for our ablation study $\tau_{entr}$ is set to $\log(10) \times 0.40$ for CIFAR10-C and $\log(100) \times 0.40$ for CIFAR100-C, as referenced in a previous work using entropy-based filtering (Niu et al., 2022). Additionally, the parameters for the soft shrinkage function in IoBMN are fixed with $\alpha = 4$ for Tent, CoTTA, SAR, RoTTA, and $\alpha = 2$ for EATA.

## C DETAILED EXPERIMENT RESULTS

In this section, we provide detailed experimental results for the performance comparison of SNAP-TTA across a wide range of adaptation rates. We evaluated the performance on CIFAR10-C, CIFAR100-C, and ImageNet-C datasets with adaptation rates of 0.01, 0.03, 0.05, 0.1, 0.3, and 0.5, and across five state-of-the-art (SOTA) TTA algorithms: Tent, EATA, SAR, CoTTA, and RoTTA. This comprehensive evaluation resulted in a total of 150 combinations (3 datasets, 6 adaptation rates, 5 algorithms).

The results demonstrate that, regardless of the adaptation rate, dataset, or the TTA algorithm, integrating SNAP-TTA consistently outperforms the baseline methods. Specifically, SNAP-TTA achieved the highest accuracy across nearly all of these 150 combinations, effectively demonstrating its robustness in both high and low adaptation settings. For CIFAR10-C and CIFAR100-C, SNAP-TTA showed substantial performance improvements compared to the baseline, even at very low adaptation rates (e.g., 0.01 and 0.05). Similarly, for ImageNet-C, SNAP-TTA maintained superior accuracy across diverse corruption types.

These results highlight that SNAP-TTA effectively balances adaptation and latency, ensuring optimal performance even when the adaptation rate is sparse and regardless of the underlying TTA algorithm. This consistent superiority across all 150 combinations underscores SNAP-TTA's suitability for practical, real-world applications on resource-constrained devices.

## C.1 IMAGENET-C

Table 6: STTA classification accuracy (%) comparing with and without SNAP-TTA on ImageNet-C through Adaptation Rates(AR) (0.5, 0.3, and 0.1), including results for full adaptation (AR=1). **Bold** numbers are the highest accuracy.

| AR | Methods | Gau. | Shot | Imp. | Def. | Gla. | Mot. | Zoom | Snow | Fro. | Fog | Brit. | Cont. | Elas. | Pix. | JPEG | Avg. |
|---|---|---|---|---|---|---|---|---|---|---|---|---|---|---|---|---|---|
| 1 | Source | 3.00 ±0.00 | 3.70 ±0.00 | 2.64 ±0.00 | 17.90 ±0.00 | 9.74 ±0.00 | 14.72 ±0.00 | 22.45 ±0.00 | 16.60 ±0.00 | 23.06 ±0.00 | 24.00 ±0.00 | 59.11 ±0.00 | 5.37 ±0.00 | 16.50 ±0.00 | 20.88 ±0.00 | 32.63 ±0.00 | 18.15 ±0.00 |
| | BN stats | 14.29 ±0.05 | 15.06 ±0.02 | 14.89 ±0.08 | 13.30 ±0.08 | 13.38 ±0.08 | 23.78 ±0.05 | 35.22 ±0.06 | 31.78 ±0.04 | 30.26 ±0.07 | 44.40 ±0.14 | 62.39 ±0.11 | 15.14 ±0.05 | 40.42 ±0.10 | 45.25 ±0.04 | 36.53 ±0.16 | 29.07 ±0.07 |
| | Tent | 27.03 ±0.05 | 28.98 ±0.08 | 28.64 ±0.29 | 24.66 ±0.27 | 23.63 ±0.25 | 38.70 ±0.10 | 45.77 ±0.12 | 44.82 ±0.08 | 38.06 ±0.35 | 54.59 ±0.08 | 64.61 ±0.10 | 16.84 ±1.51 | 51.64 ±0.10 | 55.54 ±0.15 | 49.38 ±0.07 | 39.53 ±0.24 |
| | CoTTA | 13.12 ±0.08 | 13.98 ±0.07 | 13.94 ±0.01 | 12.44 ±0.10 | 12.18 ±0.04 | 23.74 ±0.04 | 35.22 ±0.06 | 31.78 ±0.05 | 30.26 ±0.06 | 44.40 ±0.14 | 62.40 ±0.11 | 15.13 ±0.03 | 40.42 ±0.10 | 45.26 ±0.04 | 36.53 ±0.16 | 28.72 ±0.07 |
| | EATA | 29.62 ±0.02 | 31.79 ±0.09 | 31.17 ±0.19 | 26.89 ±0.03 | 26.30 ±0.15 | 40.65 ±0.12 | 47.44 ±0.06 | 46.29 ±0.09 | 40.78 ±0.05 | 55.57 ±0.08 | 64.97 ±0.08 | 38.02 ±0.08 | 52.66 ±0.20 | 56.03 ±0.04 | 50.26 ±0.16 | 42.56 ±0.10 |
| | SAR | 17.49 ±0.40 | 22.04 ±1.44 | 21.21 ±0.96 | 11.62 ±0.72 | 12.60 ±0.97 | 39.76 ±0.63 | 44.13 ±0.11 | 45.98 ±0.23 | 29.39 ±0.30 | 55.13 ±0.20 | 63.71 ±0.08 | 17.34 ±0.61 | 52.31 ±0.08 | 56.09 ±0.18 | 49.35 ±0.13 | 35.21 ±0.47 |
| | RoTTA | 20.60 ±0.07 | 22.83 ±0.09 | 19.81 ±0.24 | 10.46 ±0.04 | 10.10 ±0.26 | 21.31 ±0.27 | 31.83 ±0.23 | 39.66 ±0.18 | 32.09 ±0.18 | 46.08 ±0.23 | 62.22 ±0.27 | 20.27 ±0.49 | 42.54 ±0.29 | 47.47 ±0.23 | 40.67 ±0.10 | 31.20 ±0.21 |
| 0.5 | Tent | 25.24 ±0.10 | 26.86 ±0.27 | 26.35 ±0.08 | 23.26 ±0.06 | 22.41 ±0.05 | 35.99 ±0.09 | 44.60 ±0.10 | 42.96 ±0.13 | 37.68 ±0.17 | 53.60 ±0.15 | 64.40 ±0.12 | 21.35 ±0.94 | 50.23 ±0.12 | 54.32 ±0.15 | 47.93 ±0.04 | 38.48 ±0.17 |
| | **+ SNAP-TTA** | **28.05** ±0.00 | **29.97** ±0.04 | **29.39** ±0.19 | **25.73** ±0.15 | **23.39** ±0.06 | **38.49** ±0.17 | **45.65** ±0.03 | **44.21** ±0.09 | **39.57** ±0.10 | **53.90** ±0.10 | **64.52** ±0.09 | **34.39** ±1.83 | **49.99** ±0.14 | **54.88** ±0.07 | **48.72** ±0.09 | **40.72** ±0.21 |
| | CoTTA | 11.99 ±0.13 | 13.04 ±0.20 | 12.86 ±0.10 | 11.90 ±0.07 | 11.64 ±0.07 | 22.92 ±0.02 | 35.06 ±0.06 | 31.20 ±0.09 | 29.97 ±0.06 | 44.28 ±0.07 | 62.16 ±0.07 | 14.02 ±0.09 | 40.39 ±0.05 | 45.29 ±0.09 | 36.58 ±0.12 | 28.22 ±0.09 |
| | **+ SNAP-TTA** | **15.16** ±0.14 | **15.96** ±0.02 | **15.86** ±0.14 | **13.98** ±0.00 | **14.13** ±0.00 | **24.69** ±0.09 | **36.51** ±0.07 | **32.59** ±0.16 | **31.71** ±0.06 | **45.98** ±0.04 | **63.62** ±0.09 | **15.72** ±0.05 | **42.05** ±0.04 | **46.71** ±0.09 | **37.93** ±0.24 | **30.17** ±0.09 |
| | EATA | 28.62 ±0.10 | 30.12 ±0.10 | 29.94 ±0.14 | 25.34 ±0.20 | 24.48 ±0.44 | 38.94 ±0.10 | 46.85 ±0.25 | 45.20 ±0.12 | 40.03 ±0.01 | 55.04 ±0.06 | 64.84 ±0.07 | 34.48 ±0.41 | 52.06 ±0.24 | 55.57 ±0.13 | 49.85 ±0.05 | 41.42 ±0.16 |
| | **+ SNAP-TTA** | **30.00** ±0.29 | **31.88** ±0.17 | **31.47** ±0.13 | **26.93** ±0.21 | **26.64** ±0.28 | **39.16** ±0.15 | **47.23** ±0.07 | **45.36** ±0.13 | 39.75 ±0.14 | **55.30** ±0.14 | 64.52 ±0.10 | 33.75 ±0.07 | **52.29** ±0.09 | **55.66** ±0.18 | **50.48** ±0.08 | **42.03** ±0.15 |
| | SAR | 26.74 ±0.25 | 28.56 ±1.75 | 28.77 ±0.13 | 19.90 ±0.21 | 21.50 ±0.38 | 39.97 ±0.10 | 44.98 ±0.12 | 45.95 ±0.17 | 34.22 ±0.80 | 55.04 ±0.05 | 63.93 ±0.03 | 6.58 ±0.64 | 52.50 ±0.10 | 55.98 ±0.19 | 49.71 ±0.09 | 38.29 ±0.33 |
| | **+ SNAP-TTA** | **31.58** ±0.38 | **33.22** ±2.44 | **33.77** ±0.56 | **26.47** ±1.69 | **26.26** ±0.94 | **44.01** ±0.10 | **47.94** ±0.04 | **48.77** ±0.12 | **42.51** ±0.09 | **56.96** ±0.13 | **64.86** ±0.10 | 28.31 ±10.99 | **54.23** ±0.08 | **57.55** ±0.16 | **51.90** ±0.19 | **43.22** ±1.20 |
| | RoTTA | 18.17 ±0.05 | 19.59 ±0.03 | 18.49 ±0.10 | 12.32 ±0.11 | 11.79 ±0.13 | 23.56 ±0.15 | 34.62 ±0.16 | 37.84 ±0.14 | 32.91 ±0.11 | 47.86 ±0.06 | 63.94 ±0.13 | 18.68 ±0.42 | 43.21 ±0.08 | 48.54 ±0.23 | 40.20 ±0.23 | 31.45 ±0.14 |
| | **+ SNAP-TTA** | **20.43** ±0.03 | **22.03** ±0.08 | **21.05** ±0.11 | **15.47** ±0.11 | **14.49** ±0.07 | **26.36** ±0.06 | **36.46** ±0.10 | **38.98** ±0.09 | **34.15** ±0.12 | **48.41** ±0.13 | **64.02** ±0.13 | **20.74** ±0.23 | **43.66** ±0.10 | **49.16** ±0.10 | **41.05** ±0.15 | **33.10** ±0.11 |
| 0.3 | Tent | 23.63 ±0.08 | 25.18 ±0.37 | 24.80 ±0.28 | 21.81 ±0.02 | 20.97 ±0.18 | 34.11 ±0.07 | 43.60 ±0.04 | 41.44 ±0.05 | 36.98 ±0.04 | 52.66 ±0.15 | 64.21 ±0.13 | 22.74 ±0.04 | 48.96 ±0.16 | 53.46 ±0.07 | 46.80 ±0.09 | 37.42 ±0.12 |
| | **+ SNAP-TTA** | **26.60** ±0.20 | **28.21** ±0.19 | **27.94** ±0.33 | **24.37** ±0.12 | **22.39** ±0.07 | **36.45** ±0.13 | **44.36** ±0.07 | **42.64** ±0.15 | **38.54** ±0.06 | **52.91** ±0.10 | **64.26** ±0.03 | **33.47** ±0.10 | 48.58 ±0.14 | **53.90** ±0.11 | **47.41** ±0.11 | **39.47** ±0.17 |
| | CoTTA | 11.74 ±0.09 | 12.74 ±0.06 | 12.68 ±0.07 | 11.77 ±0.17 | 11.62 ±0.14 | 22.64 ±0.14 | 34.97 ±0.07 | 31.05 ±0.01 | 29.81 ±0.13 | 44.24 ±0.05 | 62.12 ±0.06 | 13.73 ±0.02 | 40.31 ±0.15 | 45.19 ±0.08 | 36.71 ±0.09 | 28.09 ±0.09 |
| | **+ SNAP-TTA** | **15.26** ±0.16 | **16.00** ±0.09 | **15.83** ±0.06 | **13.81** ±0.04 | **14.13** ±0.01 | **24.84** ±0.03 | **36.46** ±0.13 | **32.58** ±0.03 | **31.73** ±0.08 | **46.04** ±0.21 | **63.52** ±0.06 | **15.69** ±0.08 | **42.18** ±0.07 | **46.74** ±0.05 | **38.00** ±0.14 | **30.19** ±0.08 |
| | EATA | 27.35 ±0.04 | 29.03 ±0.15 | 28.62 ±0.27 | 23.94 ±0.06 | 23.45 ±0.60 | 37.21 ±0.30 | 46.18 ±0.13 | 44.05 ±0.20 | 39.19 ±0.22 | 54.52 ±0.01 | 64.54 ±0.06 | 32.20 ±0.62 | 51.22 ±0.16 | 55.00 ±0.10 | 49.27 ±0.21 | 40.38 ±0.21 |
| | **+ SNAP-TTA** | **29.48** ±0.14 | **31.20** ±0.04 | **30.69** ±0.11 | **26.68** ±0.14 | **25.90** ±0.25 | **38.24** ±0.01 | **46.60** ±0.22 | **44.62** ±0.06 | **39.31** ±0.19 | **54.82** ±0.13 | 64.44 ±0.09 | **32.87** ±0.25 | **51.41** ±0.06 | **55.41** ±0.14 | **49.78** ±0.14 | **41.43** ±0.14 |
| | SAR | 28.12 ±0.13 | 29.30 ±0.89 | 29.63 ±0.17 | 22.37 ±0.47 | 23.88 ±0.33 | 39.34 ±0.18 | 45.36 ±0.11 | 45.69 ±0.18 | 36.73 ±0.79 | 54.91 ±0.07 | 64.11 ±0.02 | 10.96 ±1.33 | 52.22 ±0.19 | 55.76 ±0.13 | 49.60 ±0.08 | 39.20 ±0.34 |
| | **+ SNAP-TTA** | **32.63** ±0.11 | **34.69** ±0.23 | **34.26** ±0.18 | **28.91** ±0.27 | **27.96** ±0.29 | **43.51** ±0.14 | **47.79** ±0.03 | **48.27** ±0.11 | **42.41** ±0.13 | **56.45** ±0.09 | **64.77** ±0.07 | **32.76** ±3.04 | **53.74** ±0.13 | **57.21** ±0.28 | **51.67** ±0.12 | **43.80** ±0.35 |
| | RoTTA | 16.90 ±0.15 | 17.88 ±0.11 | 17.25 ±0.08 | 12.89 ±0.17 | 12.51 ±0.05 | 23.96 ±0.03 | 35.26 ±0.16 | 36.26 ±0.01 | 32.32 ±0.07 | 47.25 ±0.02 | 63.98 ±0.13 | 17.46 ±0.18 | 42.77 ±0.09 | 48.21 ±0.24 | 39.35 ±0.15 | 30.95 ±0.11 |
| | **+ SNAP-TTA** | **18.63** ±0.07 | **19.94** ±0.08 | **19.35** ±0.06 | **14.88** ±0.06 | **14.34** ±0.08 | **25.88** ±0.05 | **36.47** ±0.03 | **37.13** ±0.04 | **33.32** ±0.11 | **47.74** ±0.17 | 63.96 ±0.06 | **19.08** ±0.21 | **42.98** ±0.07 | **48.73** ±0.17 | **40.27** ±0.20 | **32.18** ±0.09 |
| 0.1 | Tent | 22.00 ±3.47 | 23.51 ±3.92 | 23.07 ±3.85 | 19.38 ±2.30 | 18.86 ±2.06 | 32.15 ±3.40 | 42.29 ±2.45 | 39.70 ±3.27 | 34.33 ±0.60 | 51.62 ±2.30 | 63.70 ±0.29 | 15.79 ±4.61 | 47.74 ±2.84 | 52.35 ±2.27 | 45.54 ±2.98 | 35.47 ±2.71 |
| | **+ SNAP-TTA** | **26.21** ±4.92 | **27.85** ±5.36 | **27.50** ±5.30 | **23.62** ±4.28 | **22.73** ±4.11 | **36.01** ±5.57 | **44.11** ±3.72 | **42.19** ±4.49 | **38.15** ±3.37 | **52.95** ±3.47 | **64.57** ±1.18 | **30.23** ±5.15 | **48.56** ±4.29 | **53.71** ±3.31 | **47.09** ±4.09 | **39.03** ±4.17 |
| | CoTTA | 10.97 ±0.32 | 11.92 ±0.32 | 11.98 ±0.18 | 11.45 ±0.04 | 11.38 ±0.34 | 22.39 ±0.02 | 34.96 ±0.15 | 30.88 ±0.14 | 29.89 ±0.09 | 44.09 ±0.28 | 61.96 ±0.18 | 13.08 ±0.16 | 40.20 ±0.10 | 45.27 ±0.10 | 36.71 ±0.17 | 27.81 |
| | **+ SNAP-TTA** | **15.13** ±0.06 | **16.03** ±0.09 | **15.91** ±0.04 | **13.86** ±0.00 | **14.02** ±0.07 | **24.90** ±0.05 | **36.51** ±0.05 | **32.56** ±0.06 | **31.81** ±0.12 | **46.02** ±0.06 | **63.60** ±0.10 | **15.69** ±0.04 | **41.94** ±0.09 | **46.78** ±0.09 | **38.03** ±0.12 | **30.19** ±0.07 |
| | EATA | 22.43 ±0.05 | 23.78 ±0.16 | 23.26 ±0.43 | 19.38 ±0.26 | 19.42 ±0.51 | 32.18 ±0.31 | 43.22 ±0.19 | 40.65 ±0.15 | 36.64 ±0.16 | 52.38 ±0.27 | 63.87 ±0.05 | 24.59 ±1.52 | 48.13 ±0.40 | 52.89 ±0.12 | 46.33 ±0.14 | 36.61 ±0.32 |
| | **+ SNAP-TTA** | **26.10** ±0.09 | **27.29** ±0.13 | **27.13** ±0.20 | **22.38** ±0.32 | **22.15** ±0.14 | **33.45** ±0.27 | **43.92** ±0.08 | **40.96** ±0.16 | **36.68** ±0.01 | **52.71** ±0.09 | 63.77 ±0.10 | **27.93** ±0.18 | **48.47** ±0.24 | **53.23** ±0.10 | **47.46** ±0.17 | **38.24** ±0.15 |
| | SAR | 26.12 ±0.17 | 27.56 ±0.01 | 26.93 ±0.11 | 22.51 ±0.24 | 23.35 ±0.21 | 36.03 ±0.21 | 44.48 ±0.09 | 43.19 ±0.09 | 37.26 ±0.32 | 53.82 ±0.10 | 64.15 ±0.07 | 19.87 ±2.10 | 50.78 ±0.12 | 54.78 ±0.18 | 48.43 ±0.07 | 38.62 ±0.28 |
| | **+ SNAP-TTA** | **30.28** ±0.16 | **31.97** ±0.24 | **31.30** ±0.12 | **26.67** ±0.34 | **26.31** ±0.37 | **39.66** ±0.25 | **46.08** ±0.04 | **45.43** ±0.09 | **40.26** ±0.13 | **54.76** ±0.23 | **64.62** ±0.05 | **36.12** ±0.67 | **51.26** ±0.06 | **55.42** ±0.20 | **49.63** ±0.06 | **41.99** ±0.20 |
| | RoTTA | 14.77 ±0.04 | 15.59 ±0.04 | 15.33 ±0.04 | 13.17 ±0.07 | 13.19 ±0.10 | 23.85 ±0.05 | 35.38 ±0.05 | 32.73 ±0.03 | 30.77 ±0.04 | 45.22 ±0.15 | 63.08 ±0.12 | 15.62 ±0.02 | 41.05 ±0.10 | 46.15 ±0.07 | 37.19 ±0.13 | 29.54 ±0.07 |
| | **+ SNAP-TTA** | **15.35** ±0.03 | **16.20** ±0.01 | **16.01** ±0.07 | **13.67** ±0.09 | **13.66** ±0.07 | **24.27** ±0.03 | **35.62** ±0.01 | **33.04** ±0.07 | **31.02** ±0.04 | **45.38** ±0.11 | 62.95 ±0.08 | **15.96** ±0.08 | **41.06** ±0.11 | **46.17** ±0.07 | **37.44** ±0.19 | **29.85** ±0.07 |

Table 7: STTA classification accuracy (%) comparing with and without SNAP-TTA on ImageNet-C through Adaptation Rates(AR) (0.05, 0.03, and 0.01). **Bold** numbers are the highest accuracy.

| AR | Methods | Gau. | Shot | Imp. | Def. | Gla. | Mot. | Zoom | Snow | Fro. | Fog | Brit. | Cont. | Elas. | Pix. | JPEG | Avg. |
|---|---|---|---|---|---|---|---|---|---|---|---|---|---|---|---|---|---|
| 0.05 | Tent | 23.77 ±0.40 | 24.65 ±0.43 | 24.44 ±0.58 | 20.54 ±0.70 | 20.27 ±0.69 | 32.73 ±0.30 | 43.57 ±0.14 | 40.82 ±0.15 | 35.92 ±0.33 | 52.78 ±0.12 | 63.82 ±0.02 | 15.95 ±1.18 | 49.33 ±0.18 | 53.46 ±0.09 | 47.19 ±0.03 | 36.62 ±0.35 |
| | **+ SNAP-TTA** | **29.12** ±0.09 | **30.46** ±0.22 | **30.30** ±0.48 | **25.77** ±0.20 | **25.22** ±0.23 | **38.21** ±0.43 | **46.14** ±0.00 | **44.29** ±0.13 | **39.95** ±0.07 | **54.65** ±0.15 | **65.47** ±0.09 | **33.81** ±1.10 | **50.83** ±0.13 | **55.59** ±0.10 | **49.21** ±0.03 | **41.27** ±0.23 |
| | CoTTA | 11.03 ±0.30 | 11.91 ±0.57 | 11.75 ±0.33 | 11.03 ±0.24 | 11.20 ±0.46 | 22.30 ±0.18 | 34.98 ±0.05 | 30.87 ±0.08 | 29.78 ±0.01 | 43.99 ±0.11 | 61.87 ±0.06 | 12.92 ±0.36 | 40.26 ±0.19 | 45.23 ±0.17 | 36.63 ±0.07 | 27.72 ±0.21 |
| | **+ SNAP-TTA** | **15.22** ±0.08 | **15.97** ±0.11 | **15.93** ±0.03 | **13.91** ±0.06 | **14.05** ±0.12 | **24.87** ±0.04 | **36.48** ±0.00 | **32.60** ±0.07 | **31.65** ±0.04 | **46.09** ±0.03 | **63.59** ±0.07 | **15.67** ±0.05 | **42.00** ±0.03 | **46.71** ±0.09 | **37.96** ±0.09 | **30.18** ±0.06 |
| | EATA | 19.53 ±0.31 | 20.65 ±0.66 | 20.72 ±0.75 | 16.74 ±0.41 | 16.96 ±0.58 | 29.11 ±0.49 | 41.22 ±0.27 | 37.96 ±0.18 | 34.84 ±0.23 | 50.75 ±0.21 | 63.29 ±0.13 | 19.86 ±1.26 | 45.92 ±0.35 | 51.15 ±0.17 | 44.13 ±0.09 | 34.19 ±0.41 |
| | **+ SNAP-TTA** | **22.83** ±0.10 | **23.95** ±0.34 | **23.62** ±0.30 | **19.43** ±0.09 | **19.70** ±0.19 | **30.34** ±0.56 | **41.59** ±0.08 | **38.06** ±0.11 | **35.06** ±0.21 | **50.98** ±0.18 | **63.30** ±0.13 | **23.72** ±0.30 | **46.26** ±0.16 | **51.52** ±0.16 | **45.46** ±0.18 | **35.72** ±0.21 |
| | SAR | 23.25 ±0.21 | 24.23 ±0.34 | 23.66 ±0.30 | 19.98 ±0.09 | 20.38 ±0.16 | 33.05 ±0.30 | 43.04 ±0.16 | 40.73 ±0.02 | 36.06 ±0.09 | 52.61 ±0.07 | 64.09 ±0.05 | 20.17 ±0.84 | 49.00 ±0.11 | 53.35 ±0.10 | 46.73 ±0.11 | 36.69 ±0.20 |
| | **+ SNAP-TTA** | **27.54** ±0.16 | **29.03** ±0.05 | **28.66** ±0.04 | **24.05** ±0.16 | **23.42** ±0.08 | **36.28** ±0.12 | **44.12** ±0.10 | **42.89** ±0.11 | **38.54** ±0.07 | **53.24** ±0.07 | **64.25** ±0.05 | **31.83** ±0.24 | **48.79** ±0.23 | **54.04** ±0.19 | **47.80** ±0.08 | **39.63** ±0.12 |
| | RoTTA | 14.42 ±0.06 | 15.22 ±0.05 | 15.02 ±0.10 | 13.25 ±0.11 | 13.31 ±0.07 | 23.79 ±0.03 | 35.27 ±0.08 | 32.09 ±0.05 | 30.43 ±0.07 | 44.71 ±0.13 | 62.64 ±0.14 | 15.24 ±0.09 | 40.63 ±0.10 | 45.55 ±0.07 | 36.75 ±0.16 | 29.22 ±0.09 |
| | **+ SNAP-TTA** | **14.65** ±0.06 | **15.48** ±0.02 | **15.29** ±0.08 | **13.43** ±0.09 | **13.45** ±0.09 | **23.93** ±0.03 | **35.33** ±0.05 | **32.18** ±0.04 | **30.53** ±0.05 | **44.71** ±0.16 | 62.58 ±0.10 | **15.41** ±0.04 | **40.64** ±0.09 | **45.55** ±0.10 | **36.81** ±0.14 | **29.33** ±0.08 |
| 0.03 | Tent | 21.76 ±0.17 | 22.76 ±0.35 | 22.58 ±0.17 | 19.06 ±0.04 | 18.90 ±0.12 | 30.85 ±0.22 | 42.34 ±0.12 | 38.94 ±0.26 | 35.53 ±0.31 | 51.58 ±0.18 | 63.42 ±0.11 | 18.61 ±0.91 | 47.96 ±0.26 | 52.41 ±0.21 | 45.56 ±0.08 | 35.48 ±0.23 |
| | **+ SNAP-TTA** | **26.42** ±0.14 | **28.20** ±0.26 | **27.81** ±0.37 | **23.79** ±0.46 | **22.82** ±0.21 | **35.77** ±0.11 | **44.80** ±0.16 | **42.37** ±0.34 | **38.81** ±0.14 | **53.34** ±0.06 | **64.95** ±0.11 | **30.05** ±0.62 | **49.28** ±0.17 | **54.16** ±0.09 | **47.57** ±0.08 | **39.34** ±0.22 |
| | CoTTA | 10.61 ±0.18 | 12.36 ±0.36 | 11.78 ±0.57 | 11.66 ±0.57 | 11.32 ±0.26 | 22.25 ±0.11 | 35.01 ±0.18 | 30.88 ±0.24 | 29.84 ±0.07 | 44.09 ±0.11 | 61.83 ±0.16 | 12.92 ±0.12 | 40.26 ±0.19 | 45.20 ±0.11 | 36.58 ±0.09 | 27.77 ±0.22 |
| | **+ SNAP-TTA** | **15.29** ±0.08 | **16.02** ±0.07 | **16.00** ±0.00 | **13.99** ±0.07 | **14.06** ±0.11 | **24.78** ±0.05 | **36.54** ±0.07 | **32.62** ±0.06 | **31.70** ±0.08 | **46.01** ±0.04 | **63.49** ±0.04 | **15.69** ±0.18 | **42.05** ±0.09 | **46.75** ±0.08 | **37.97** ±0.08 | **30.20** ±0.08 |
| | EATA | 17.17 ±0.41 | 18.34 ±0.19 | 17.94 ±0.36 | 14.48 ±0.82 | 15.04 ±0.22 | 26.31 ±0.25 | 39.47 ±0.33 | 35.51 ±0.50 | 33.41 ±0.33 | 49.16 ±0.19 | **63.06** ±0.05 | 18.01 ±0.88 | 44.16 ±0.31 | 49.90 ±0.09 | 42.47 ±0.31 | 32.30 ±0.35 |
| | **+ SNAP-TTA** | **20.75** ±0.32 | **21.87** ±0.41 | **21.28** ±0.35 | **17.34** ±0.30 | **17.90** ±0.34 | **28.08** ±0.34 | **39.84** ±0.16 | **36.27** ±0.13 | **33.54** ±0.11 | **49.50** ±0.12 | 63.04 ±0.07 | **20.86** ±0.33 | **44.68** ±0.28 | **49.97** ±0.13 | **43.53** ±0.03 | **33.90** ±0.23 |
| | SAR | 20.38 ±0.10 | 21.34 ±0.14 | 21.18 ±0.36 | 18.24 ±0.18 | 18.28 ±0.27 | 30.56 ±0.08 | 41.63 ±0.12 | 38.57 ±0.17 | 35.23 ±0.28 | 51.19 ±0.22 | 63.74 ±0.04 | 20.40 ±0.20 | 47.32 ±0.09 | 52.02 ±0.09 | 44.81 ±0.19 | 34.99 ±0.17 |
| | **+ SNAP-TTA** | **25.11** ±0.23 | **26.27** ±0.31 | **26.00** ±0.10 | **22.02** ±0.49 | **21.25** ±0.56 | **33.51** ±0.31 | **42.86** ±0.14 | **40.83** ±0.16 | **37.09** ±0.16 | **51.87** ±0.12 | **63.83** ±0.06 | **28.36** ±0.29 | **47.19** ±0.34 | **52.63** ±0.06 | **45.80** ±0.30 | **37.64** ±0.25 |
| | RoTTA | 14.36 ±0.04 | 15.12 ±0.03 | 14.95 ±0.08 | 13.30 ±0.08 | 13.34 ±0.08 | 23.78 ±0.04 | 35.23 ±0.05 | 31.89 ±0.04 | 30.33 ±0.04 | 44.52 ±0.11 | 62.48 ±0.12 | 15.20 ±0.01 | 40.50 ±0.11 | 45.36 ±0.07 | 36.63 ±0.17 | 29.13 ±0.07 |
| | **+ SNAP-TTA** | **14.45** ±0.04 | **15.21** ±0.02 | **15.06** ±0.08 | **13.35** ±0.08 | **13.42** ±0.07 | **23.83** ±0.04 | **35.26** ±0.06 | **31.92** ±0.02 | **30.36** ±0.08 | **44.53** ±0.10 | 62.47 ±0.09 | **15.27** ±0.04 | 40.50 ±0.10 | **45.39** ±0.08 | **36.65** ±0.16 | **29.18** ±0.07 |
| 0.01 | Tent | 17.09 ±0.14 | 17.70 ±0.10 | 17.69 ±0.13 | 14.91 ±0.23 | 15.25 ±0.09 | 25.23 ±0.25 | 38.66 ±0.27 | 34.15 ±0.27 | 32.28 ±0.21 | 48.14 ±0.21 | 62.65 ±0.21 | 15.76 ±0.16 | 43.44 ±0.48 | 49.14 ±0.23 | 41.18 ±0.04 | 31.55 ±0.19 |
| | **+ SNAP-TTA** | **20.66** ±0.02 | **21.73** ±0.12 | **21.55** ±0.18 | **18.46** ±0.34 | **18.28** ±0.33 | **29.88** ±0.12 | **40.63** ±0.14 | **36.97** ±0.21 | **34.89** ±0.10 | **49.85** ±0.26 | **64.29** ±0.10 | **22.64** ±0.26 | **45.13** ±0.08 | **50.77** ±0.29 | **43.17** ±0.07 | **34.59** ±0.19 |
| | CoTTA | 11.11 ±0.61 | 13.24 ±0.12 | 11.86 ±0.65 | 10.85 ±0.59 | 10.97 ±0.98 | 22.18 ±0.05 | 34.96 ±0.18 | 30.88 ±0.14 | 29.63 ±0.21 | 44.09 ±0.21 | 61.71 ±0.22 | 12.81 ±0.53 | 40.16 ±0.20 | 45.14 ±0.22 | 36.73 ±0.12 | 27.75 ±0.34 |
| | **+ SNAP-TTA** | **15.09** ±0.04 | **16.00** ±0.09 | **15.83** ±0.14 | **13.84** ±0.09 | **14.06** ±0.02 | **24.70** ±0.07 | **36.47** ±0.02 | **32.59** ±0.11 | **31.66** ±0.03 | **46.10** ±0.15 | **63.62** ±0.07 | **15.60** ±0.06 | **42.03** ±0.10 | **46.74** ±0.01 | **38.17** ±0.20 | **30.17** ±0.08 |
| | EATA | 14.85 ±0.13 | 15.61 ±0.21 | 15.69 ±0.21 | 13.26 ±0.04 | 13.37 ±0.06 | 23.72 ±0.19 | 36.18 ±0.06 | 32.57 ±0.09 | **31.14** ±0.06 | 46.06 ±0.19 | **62.35** ±0.35 | 13.88 ±0.17 | 41.91 ±0.15 | 47.00 ±0.09 | 38.88 ±0.15 | 29.76 ±0.15 |
| | **+ SNAP-TTA** | **16.73** ±0.12 | **17.55** ±0.10 | **17.30** ±0.19 | **14.35** ±0.09 | **14.64** ±0.10 | **24.13** ±0.36 | **36.83** ±0.23 | **32.81** ±0.08 | 31.09 ±0.10 | **46.63** ±0.19 | 62.20 ±0.01 | **15.26** ±0.54 | **42.34** ±0.12 | **47.44** ±0.18 | **39.81** ±0.34 | **30.61** ±0.19 |
| | SAR | 16.08 ±0.08 | 17.04 ±0.07 | 16.69 ±0.10 | 14.72 ±0.16 | 14.78 ±0.12 | 25.92 ±0.13 | 37.85 ±0.05 | 34.07 ±0.24 | 32.25 ±0.11 | 47.66 ±0.13 | 63.15 ±0.05 | 17.20 ±0.15 | 43.05 ±0.20 | 48.78 ±0.09 | 40.14 ±0.20 | 31.29 ±0.13 |
| | **+ SNAP-TTA** | **18.89** ±0.15 | **19.45** ±0.15 | **19.70** ±0.12 | **16.70** ±0.14 | **16.55** ±0.15 | **27.69** ±0.16 | **38.57** ±0.11 | **35.34** ±0.22 | **33.09** ±0.09 | **48.08** ±0.31 | 63.04 ±0.07 | **20.39** ±0.12 | 42.95 ±0.29 | 48.76 ±0.26 | **40.99** ±0.33 | **32.68** ±0.18 |
| | RoTTA | 14.30 ±0.05 | 15.06 ±0.03 | 14.89 ±0.07 | **13.30** ±0.07 | 13.37 ±0.08 | 23.78 ±0.07 | **35.22** ±0.06 | 31.79 ±0.04 | 30.27 ±0.06 | 44.40 ±0.14 | **62.40** ±0.11 | **15.16** ±0.06 | 40.42 ±0.10 | **45.27** ±0.05 | **36.54** ±0.16 | **29.08** ±0.07 |
| | **+ SNAP-TTA** | **14.30** ±0.06 | **15.07** ±0.03 | **14.92** ±0.08 | **13.30** ±0.08 | **13.38** ±0.07 | 23.78 ±0.04 | **35.22** ±0.06 | 31.78 ±0.04 | 30.26 ±0.07 | **44.41** ±0.14 | **62.40** ±0.11 | 15.15 ±0.05 | **40.43** ±0.09 | **45.27** ±0.04 | **36.54** ±0.15 | **29.08** ±0.07 |

## C.2 CIFAR10-C

Table 8: STTA classification accuracy (%) comparing with and without SNAP-TTA on CIFAR10-C through Adaptation Rates(AR) (0.5, 0.3, and 0.1), including results for full adaptation (AR=1). **Bold** numbers are the highest accuracy.

| AR | Methods | Gau. | Shot | Imp. | Def. | Gla. | Mot. | Zoom | Snow | Fro. | Fog | Brit. | Cont. | Elas. | Pix. | JPEG | Avg. |
|---|---|---|---|---|---|---|---|---|---|---|---|---|---|---|---|---|---|
| 1 | Source | 22.13±0.00 | 29.25±0.00 | 22.53±0.00 | 54.54±0.00 | 55.10±0.00 | 67.45±0.00 | 64.37±0.00 | 78.25±0.00 | 69.93±0.00 | 74.26±0.00 | 91.29±0.00 | 35.45±0.00 | 77.20±0.00 | 46.56±0.00 | 73.38±0.00 | 57.45±0.00 |
| | BN stats | 63.72±0.48 | 65.67±0.12 | 57.14±0.25 | 84.99±0.31 | 62.72±0.23 | 83.86±0.48 | 84.26±0.30 | 78.98±0.30 | 76.95±0.08 | 83.32±0.17 | 88.46±0.16 | 84.60±0.17 | 73.96±0.18 | 76.61±0.02 | 68.79±0.42 | 75.60±0.24 |
| | Tent | 73.66±0.88 | 76.18±0.94 | 68.04±1.32 | 86.61±0.50 | 67.12±0.76 | 85.73±0.38 | 86.24±0.09 | 82.34±0.94 | 81.56±0.64 | 86.02±0.18 | 89.99±0.16 | 87.16±2.50 | 76.40±0.82 | 82.95±0.15 | 76.45±0.46 | 80.43±0.71 |
| | CoTTA | 71.95±0.32 | 73.97±0.48 | 67.03±0.66 | 83.91±0.20 | 66.75±0.08 | 82.64±0.34 | 83.34±0.19 | 79.92±0.09 | 79.49±0.13 | 82.41±0.23 | 88.39±0.18 | 80.14±0.17 | 75.38±0.09 | 79.24±0.07 | 75.42±0.25 | 78.00±0.23 |
| | EATA | 75.82±0.50 | 77.61±0.27 | 69.63±0.87 | 87.14±0.29 | 69.41±0.68 | 85.96±0.39 | 87.08±0.27 | 83.42±0.38 | 82.28±0.29 | 86.58±0.41 | 90.40±0.17 | 89.26±0.39 | 77.62±0.28 | 83.35±0.32 | 77.77±0.20 | 81.56±0.38 |
| | SAR | 73.52±1.53 | 74.03±0.46 | 65.45±1.81 | 85.69±0.37 | 65.01±0.35 | 84.63±0.53 | 85.01±0.34 | 81.47±0.37 | 80.91±0.72 | 84.18±0.09 | 88.70±0.12 | 86.23±0.16 | 74.94±0.03 | 81.20±0.28 | 74.84±0.69 | 79.05±0.52 |
| | RoTTA | 66.54±0.46 | 68.60±0.23 | 60.27±0.46 | 85.73±0.35 | 64.84±0.63 | 84.68±0.36 | 85.01±0.45 | 80.15±0.56 | 78.02±0.06 | 84.13±0.09 | 89.00±0.27 | 84.91±0.19 | 75.06±0.15 | 77.96±0.16 | 70.12±0.36 | 77.00±0.32 |
| 0.5 | Tent | 73.44±0.61 | 75.93±0.44 | 67.18±0.78 | 86.52±0.17 | 67.28±1.78 | 85.25±0.49 | 86.23±0.27 | 82.24±0.77 | 80.35±0.14 | 85.39±0.20 | 89.80±0.28 | 87.77±0.27 | 77.00±0.65 | 82.08±0.68 | 75.58±0.60 | 80.14±0.55 |
| | **+ SNAP-TTA** | **75.17±0.00** | **77.66±0.78** | **68.78±1.26** | **88.25±0.38** | **69.18±0.51** | **87.11±0.18** | **88.19±0.13** | **84.21±0.29** | **82.72±0.45** | **87.34±0.51** | **91.63±0.12** | **86.30±1.07** | **78.76±0.28** | **83.43±0.18** | **77.28±0.50** | **81.74±0.44** |
| | CoTTA | 65.08±0.26 | 66.67±0.21 | 61.30±0.16 | 77.50±0.48 | 61.36±0.15 | 77.70±0.37 | 77.37±0.37 | 74.05±0.22 | 72.86±0.44 | 77.43±0.19 | 82.69±0.30 | 72.44±0.72 | 70.52±0.07 | 70.94±0.27 | 69.79±0.10 | 71.85±0.29 |
| | **+ SNAP-TTA** | **71.89±0.45** | **74.18±0.33** | **66.92±0.19** | **85.46±0.32** | **67.57±0.26** | **84.27±0.22** | **84.91±0.18** | **81.10±0.09** | **80.62±0.46** | **84.06±0.24** | **90.16±0.17** | **82.14±0.33** | **76.75±0.16** | **80.23±0.38** | **75.98±0.50** | **79.08±0.28** |
| | EATA | 73.95±0.22 | 75.82±0.18 | 68.00±0.70 | 86.83±0.25 | 67.83±0.50 | 85.27±0.39 | 86.48±0.13 | 82.63±0.50 | 80.99±0.05 | 85.45±0.16 | 89.86±0.18 | 87.61±0.53 | 77.01±0.31 | 82.13±0.18 | 76.11±0.45 | 80.40±0.32 |
| | **+ SNAP-TTA** | **74.85±0.51** | **77.63±0.46** | **68.43±0.43** | **88.53±0.17** | **69.70±0.69** | **87.19±0.35** | **88.16±0.18** | **83.87±0.42** | **82.84±0.33** | **87.18±0.15** | **91.54±0.12** | **89.62±0.38** | **78.91±0.48** | **83.76±0.14** | **77.36±0.22** | **81.97±0.33** |
| | SAR | 69.10±1.63 | 72.37±1.05 | 63.22±0.44 | 85.18±0.25 | 64.30±1.02 | 83.94±0.12 | 85.07±0.45 | 80.11±0.17 | 79.64±0.60 | 83.91±0.37 | 88.64±0.10 | 84.21±0.30 | 75.70±0.34 | 79.10±0.52 | 72.92±0.09 | 77.83±0.50 |
| | **+ SNAP-TTA** | **73.98±0.48** | **75.48±0.65** | **66.41±1.26** | **86.63±0.15** | **68.15±0.07** | **85.50±0.15** | **86.53±0.10** | **81.62±0.39** | **80.20±0.17** | **85.06±0.27** | **91.46±0.03** | **87.04±0.11** | **77.22±0.45** | **81.16±0.27** | **75.53±0.23** | **80.13±0.32** |
| | RoTTA | 65.02±0.04 | 66.84±0.52 | 58.38±0.33 | 85.26±0.42 | 63.51±0.18 | 83.81±0.15 | 84.66±0.20 | 79.26±0.29 | 76.76±0.49 | 83.46±0.21 | 88.27±0.04 | 83.47±0.05 | 74.43±0.16 | 77.39±0.29 | 69.13±0.41 | 75.98±0.25 |
| | **+ SNAP-TTA** | **66.03±0.14** | **68.09±0.15** | **58.88±0.06** | **87.09±0.27** | **64.55±0.07** | **85.70±0.03** | **86.48±0.02** | **80.97±0.22** | **78.87±0.20** | **85.29±0.22** | **90.28±0.13** | **86.22±0.10** | **76.05±0.22** | **78.76±0.22** | **70.51±0.35** | **77.58±0.16** |
| 0.3 | Tent | 71.18±0.99 | 74.06±0.80 | 65.44±1.17 | 85.93±0.28 | 66.01±0.97 | 84.37±0.14 | 85.90±0.17 | 81.31±0.40 | 79.80±0.09 | 84.80±0.25 | 89.58±0.23 | 84.01±0.30 | 75.96±0.30 | 80.46±0.39 | 74.09±0.54 | 78.86±0.47 |
| | **+ SNAP-TTA** | **74.95±0.84** | **77.29±0.55** | **67.59±0.46** | **88.27±0.27** | **67.46±0.26** | **86.97±0.21** | **87.64±0.16** | **83.46±0.40** | **82.45±0.19** | **86.72±0.19** | **91.22±0.19** | **87.79±0.98** | **78.26±0.35** | **82.61±0.38** | **75.79±0.32** | **81.23±0.39** |
| | CoTTA | 63.01±0.12 | 64.38±0.64 | 58.95±0.74 | 75.43±0.61 | 59.65±0.48 | 76.08±0.58 | 75.47±0.16 | 71.75±0.55 | 70.33±0.48 | 75.52±0.32 | 80.94±0.49 | 70.53±0.51 | 68.75±0.65 | 67.87±0.30 | 67.55±0.37 | 69.75±0.47 |
| | **+ SNAP-TTA** | **71.39±0.31** | **73.57±0.27** | **66.29±0.10** | **85.22±0.22** | **66.71±0.19** | **84.20±0.18** | **84.64±0.13** | **80.77±0.21** | **80.56±0.32** | **84.06±0.15** | **89.85±0.17** | **81.86±0.08** | **76.48±0.07** | **79.94±0.24** | **75.69±0.27** | **78.75±0.19** |
| | EATA | 70.98±1.05 | 73.70±0.28 | 65.73±1.68 | 86.01±0.35 | 66.71±0.81 | 84.36±0.23 | 86.10±0.38 | 80.92±0.47 | 79.87±0.09 | 84.48±0.04 | 89.29±0.19 | 86.33±0.31 | 76.19±0.20 | 80.66±0.58 | 73.98±0.52 | 79.02±0.48 |
| | **+ SNAP-TTA** | **74.19±0.38** | **76.64±0.68** | **67.89±0.19** | **87.93±0.25** | **68.56±0.20** | **87.08±0.05** | **87.89±0.34** | **83.56±0.30** | **82.20±0.23** | **86.60±0.22** | **91.11±0.61** | **88.94±0.23** | **78.10±0.14** | **83.03±0.20** | **75.83±0.43** | **81.30±0.30** |
| | SAR | 69.10±1.63 | 72.37±1.05 | 63.22±0.44 | 85.18±0.25 | 64.30±1.02 | 83.94±0.12 | 85.07±0.45 | 80.11±0.17 | 79.64±0.60 | 83.91±0.37 | 88.64±0.10 | 84.21±0.30 | 75.70±0.34 | 79.10±0.52 | 72.92±0.09 | 77.83±0.50 |
| | **+ SNAP-TTA** | **72.72±0.94** | **75.25±0.30** | **65.78±1.06** | **86.53±0.16** | **66.19±0.60** | **85.53±0.26** | **86.40±0.27** | **81.61±0.45** | **80.53±0.64** | **85.08±0.23** | **91.41±0.14** | **86.74±0.08** | **77.23±0.41** | **81.00±0.37** | **74.52±1.04** | **79.77±0.46** |
| | RoTTA | 64.09±0.44 | 66.07±0.13 | 57.58±0.63 | 84.97±0.20 | 62.66±0.15 | 83.06±0.18 | 84.08±0.17 | 78.60±0.34 | 76.40±0.36 | 82.86±0.05 | 88.03±0.22 | 83.21±0.24 | 74.14±0.58 | 76.35±0.47 | 68.70±0.17 | 75.39±0.29 |
| | **+ SNAP-TTA** | **65.83±0.18** | **67.57±0.19** | **58.39±0.16** | **86.97±0.18** | **64.22±0.08** | **85.63±0.16** | **86.39±0.09** | **80.75±0.15** | **78.90±0.18** | **85.21±0.16** | **90.19±0.21** | **85.92±0.09** | **75.92±0.05** | **78.91±0.37** | **70.42±0.04** | **77.41±0.18** |
| 0.1 | Tent | 67.32±0.93 | 69.39±0.96 | 60.69±0.36 | 85.34±0.24 | 63.82±0.41 | 83.52±0.13 | 84.70±0.15 | 79.68±0.41 | 77.79±0.50 | 83.75±0.08 | 88.53±0.49 | 83.12±0.66 | 75.18±0.68 | 77.82±0.69 | 71.47±0.44 | 76.81±0.48 |
| | **+ SNAP-TTA** | **70.22±0.44** | **71.48±0.91** | **63.08±0.04** | **87.35±0.20** | **65.74±0.26** | **85.89±0.25** | **86.38±0.32** | **81.93±0.33** | **80.00±0.21** | **85.62±0.14** | **90.34±0.22** | **87.47±0.11** | **76.44±0.12** | **79.63±0.14** | **72.72±0.39** | **78.95±0.27** |
| | CoTTA | 59.11±0.43 | 60.26±0.56 | 56.07±0.65 | 72.23±0.69 | 56.77±0.64 | 73.55±0.68 | 72.20±0.63 | 68.05±0.52 | 66.68±1.15 | 72.88±1.17 | 77.66±0.83 | 65.95±0.95 | 65.67±0.58 | 64.12±0.07 | 65.16±0.73 | 66.42±0.73 |
| | **+ SNAP-TTA** | **71.70±0.40** | **73.54±0.21** | **66.70±0.02** | **85.16±0.19** | **66.83±0.39** | **84.30±0.08** | **84.88±0.20** | **81.02±0.25** | **80.61±0.24** | **84.20±0.23** | **89.84±0.08** | **81.71±0.20** | **76.60±0.20** | **79.66±0.14** | **75.71±0.25** | **78.83±0.20** |
| | EATA | 66.65±0.43 | 68.96±0.47 | 59.73±0.15 | 84.93±0.27 | 63.26±0.36 | 83.10±0.24 | 84.53±0.15 | 79.28±0.44 | 77.46±0.42 | 83.48±0.13 | 88.12±0.09 | 82.46±0.24 | 74.49±0.20 | 77.48±0.69 | 70.43±0.25 | 76.29±0.30 |
| | **+ SNAP-TTA** | **69.29±0.39** | **70.49±0.57** | **61.71±0.37** | **87.32±0.42** | **65.48±0.38** | **85.96±0.29** | **86.64±0.21** | **81.44±0.34** | **79.56±0.47** | **85.47±0.23** | **90.50±0.38** | **86.84±0.36** | **76.32±0.21** | **79.64±0.12** | **72.51±0.32** | **78.61±0.34** |
| | SAR | 66.11±0.59 | 68.18±0.83 | 59.15±0.72 | 84.91±0.69 | 62.87±0.64 | 82.33±0.68 | 84.27±0.13 | 79.23±0.32 | 77.58±0.43 | 83.21±0.18 | 88.29±0.09 | 82.60±0.57 | 74.65±0.46 | 75.92±0.77 | 70.79±0.40 | 76.01±0.45 |
| | **+ SNAP-TTA** | **67.76±0.22** | **70.68±0.14** | **60.82±1.08** | **86.78±0.26** | **64.73±0.43** | **85.29±0.10** | **86.22±0.11** | **80.82±0.23** | **79.30±0.48** | **84.95±0.28** | **91.33±0.17** | **86.59±0.14** | **75.72±0.26** | **78.72±0.35** | **71.24±0.46** | **78.06±0.31** |
| | RoTTA | 63.12±0.33 | 64.84±0.21 | 56.72±0.30 | 84.49±0.04 | 62.15±0.17 | 82.53±0.30 | 83.84±0.02 | 78.03±0.29 | 76.13±0.71 | 82.88±0.16 | 87.48±0.08 | 81.49±0.11 | 73.75±0.14 | 76.04±0.29 | 68.24±0.27 | 74.78±0.23 |
| | **+ SNAP-TTA** | **65.35±0.20** | **66.99±0.15** | **58.09±0.18** | **86.77±0.18** | **63.63±0.18** | **85.47±0.13** | **86.01±0.21** | **80.54±0.11** | **78.38±0.24** | **84.99±0.43** | **90.00±0.23** | **85.99±0.03** | **75.67±0.17** | **78.14±0.06** | **70.09±0.23** | **77.07±0.18** |

Table 9: STTA classification accuracy (%) comparing with and without SNAP-TTA on CIFAR10-C through Adaptation Rates(AR) (0.05, 0.03, and 0.01). **Bold** numbers are the highest accuracy.

| AR | Methods | Gau. | Shot | Imp. | Def. | Gla. | Mot. | Zoom | Snow | Fro. | Fog | Brit. | Cont. | Elas. | Pix. | JPEG | Avg. |
|---|---|---|---|---|---|---|---|---|---|---|---|---|---|---|---|---|---|
| | Tent | 64.65 ±0.55 | 67.08 ±0.58 | 58.48 ±0.42 | 85.00 ±0.60 | 62.61 ±0.44 | 82.76 ±0.70 | 84.63 ±0.55 | 79.01 ±0.74 | 77.66 ±0.91 | 83.32 ±0.48 | 88.00 ±0.56 | 82.34 ±0.93 | 74.16 ±0.10 | 77.11 ±0.60 | 69.40 ±0.48 | 75.75 ±0.57 |
| | + SNAP-TTA | 67.71 ±0.38 | 69.84 ±0.82 | 59.53 ±1.10 | 87.10 ±0.15 | 64.66 ±0.25 | 85.73 ±0.20 | 86.35 ±0.20 | 80.68 ±0.23 | 78.92 ±0.14 | 85.60 ±0.08 | 90.19 ±0.31 | 86.72 ±0.20 | 76.16 ±0.17 | 78.86 ±0.42 | 70.95 ±0.30 | 77.93 ±0.33 |
| | CoTTA | 59.27 ±0.66 | 61.18 ±1.12 | 56.33 ±0.06 | 72.22 ±1.43 | 57.37 ±1.10 | 74.27 ±1.46 | 72.61 ±1.11 | 70.03 ±1.02 | 68.68 ±0.92 | 74.82 ±1.09 | 79.72 ±1.07 | 65.57 ±1.38 | 66.92 ±1.14 | 64.13 ±1.27 | 65.25 ±0.98 | 67.22 ±1.05 |
| | + SNAP-TTA | 71.42 ±0.29 | 73.31 ±0.12 | 65.91 ±0.13 | 85.23 ±0.11 | 67.01 ±0.21 | 84.19 ±0.20 | 84.91 ±0.14 | 80.80 ±0.19 | 80.56 ±0.34 | 84.19 ±0.14 | 90.00 ±0.23 | 82.09 ±0.35 | 76.31 ±0.05 | 79.79 ±0.29 | 75.18 ±0.21 | 78.73 ±0.20 |
| 0.05 | EATA | 64.68 ±0.31 | 67.01 ±0.37 | 58.07 ±0.24 | 84.90 ±0.54 | 62.56 ±0.33 | 82.64 ±0.67 | 84.57 ±0.61 | 78.77 ±0.71 | 77.16 ±0.92 | 83.09 ±0.44 | 87.80 ±0.47 | 81.62 ±0.59 | 74.05 ±0.28 | 76.99 ±0.41 | 69.31 ±0.71 | 75.55 ±0.51 |
| | + SNAP-TTA | 67.36 ±0.33 | 68.73 ±0.26 | 59.35 ±0.37 | 87.05 ±0.22 | 64.36 ±0.18 | 85.62 ±0.18 | 86.48 ±0.25 | 81.31 ±0.24 | 78.73 ±0.22 | 85.33 ±0.15 | 90.03 ±0.24 | 86.31 ±0.07 | 76.04 ±0.12 | 78.79 ±0.27 | 70.90 ±0.38 | 77.76 ±0.23 |
| | SAR | 64.79 ±0.13 | 66.32 ±0.86 | 57.58 ±0.69 | 84.66 ±0.72 | 62.46 ±0.26 | 81.42 ±1.52 | 84.13 ±0.34 | 78.87 ±0.26 | 77.20 ±0.81 | 82.62 ±1.24 | 88.10 ±0.41 | 82.12 ±0.74 | 74.04 ±0.05 | 75.38 ±0.80 | 69.13 ±0.52 | 75.25 ±0.62 |
| | + SNAP-TTA | 66.00 ±0.17 | 68.85 ±0.75 | 58.47 ±0.42 | 86.54 ±0.25 | 63.06 ±0.28 | 85.26 ±0.09 | 86.13 ±0.38 | 80.38 ±0.09 | 78.17 ±0.27 | 85.17 ±0.13 | 90.93 ±0.36 | 85.96 ±0.20 | 75.27 ±0.31 | 77.37 ±0.28 | 70.61 ±0.30 | 77.21 ±0.29 |
| | RoTTA | 63.21 ±0.37 | 64.87 ±0.62 | 56.60 ±0.28 | 84.64 ±0.52 | 62.16 ±0.31 | 82.31 ±0.63 | 84.13 ±0.56 | 78.16 ±0.71 | 76.39 ±0.95 | 82.90 ±0.62 | 87.44 ±0.46 | 81.47 ±0.65 | 73.59 ±0.42 | 76.02 ±0.40 | 68.09 ±0.33 | 74.80 ±0.52 |
| | + SNAP-TTA | 65.28 ±0.32 | 66.91 ±0.22 | 57.88 ±0.06 | 86.75 ±0.25 | 63.51 ±0.13 | 85.48 ±0.13 | 86.17 ±0.10 | 80.46 ±0.23 | 78.38 ±0.26 | 85.24 ±0.13 | 89.99 ±0.23 | 85.82 ±0.03 | 75.66 ±0.16 | 77.98 ±0.19 | 70.15 ±0.29 | 77.05 ±0.18 |
| | Tent | 64.36 ±0.43 | 66.21 ±0.16 | 57.65 ±1.01 | 84.73 ±0.48 | 62.95 ±0.52 | 83.07 ±0.50 | 84.50 ±0.32 | 78.46 ±0.82 | 76.99 ±0.32 | 83.00 ±0.36 | 88.07 ±0.43 | 82.62 ±0.34 | 73.93 ±0.23 | 76.50 ±0.46 | 68.82 ±0.48 | 75.46 ±0.46 |
| | + SNAP-TTA | 66.32 ±0.61 | 68.38 ±0.71 | 59.00 ±0.52 | 86.93 ±0.19 | 64.04 ±0.24 | 85.58 ±0.34 | 86.35 ±0.05 | 80.78 ±0.10 | 78.68 ±0.02 | 85.34 ±0.05 | 90.08 ±0.10 | 86.19 ±0.31 | 75.77 ±0.05 | 78.37 ±0.06 | 70.49 ±0.08 | 77.49 ±0.23 |
| | CoTTA | 60.38 ±1.71 | 61.26 ±1.94 | 56.71 ±2.47 | 72.44 ±2.23 | 57.58 ±1.85 | 74.64 ±1.74 | 72.73 ±2.61 | 69.68 ±2.03 | 68.34 ±2.02 | 74.64 ±2.52 | 79.52 ±2.37 | 67.28 ±1.89 | 67.42 ±1.77 | 64.89 ±0.79 | 66.19 ±1.73 | 67.58 ±1.98 |
| | + SNAP-TTA | 71.12 ±0.47 | 73.68 ±0.29 | 66.34 ±0.24 | 85.30 ±0.12 | 66.64 ±0.34 | 84.25 ±0.14 | 84.55 ±0.13 | 80.88 ±0.15 | 80.11 ±0.14 | 84.06 ±0.37 | 89.89 ±0.19 | 81.98 ±0.26 | 76.27 ±0.08 | 79.77 ±0.21 | 75.35 ±0.08 | 78.68 ±0.21 |
| 0.03 | EATA | 63.99 ±0.87 | 65.95 ±0.44 | 57.39 ±1.05 | 84.71 ±0.48 | 62.66 ±0.62 | 83.11 ±0.52 | 84.44 ±0.33 | 78.42 ±0.75 | 76.63 ±0.26 | 82.97 ±0.20 | 88.00 ±0.47 | 82.55 ±0.34 | 73.85 ±0.33 | 76.46 ±0.29 | 68.91 ±0.56 | 75.34 ±0.50 |
| | + SNAP-TTA | 66.16 ±0.03 | 67.60 ±0.41 | 58.81 ±0.36 | 86.95 ±0.13 | 64.06 ±0.17 | 85.49 ±0.36 | 86.34 ±0.08 | 80.79 ±0.01 | 78.65 ±0.25 | 85.24 ±0.13 | 90.09 ±0.12 | 86.23 ±0.08 | 75.88 ±0.18 | 78.48 ±0.10 | 70.56 ±0.47 | 77.42 ±0.19 |
| | SAR | 63.72 ±0.46 | 65.75 ±0.29 | 57.89 ±0.65 | 84.37 ±0.81 | 62.45 ±0.69 | 81.47 ±1.61 | 82.46 ±2.95 | 78.32 ±0.81 | 76.79 ±0.24 | 81.93 ±1.33 | 88.60 ±0.68 | 82.72 ±0.29 | 73.89 ±0.43 | 74.55 ±0.98 | 68.79 ±0.61 | 74.91 ±0.85 |
| | + SNAP-TTA | 65.40 ±0.33 | 67.68 ±0.60 | 58.37 ±0.45 | 86.72 ±0.18 | 63.11 ±0.16 | 85.10 ±0.29 | 86.18 ±0.17 | 79.93 ±0.24 | 78.05 ±0.22 | 84.92 ±0.13 | 90.93 ±0.35 | 85.58 ±0.14 | 75.30 ±0.14 | 77.22 ±0.30 | 69.97 ±0.30 | 76.96 ±0.27 |
| | RoTTA | 63.36 ±0.80 | 65.10 ±0.55 | 56.64 ±0.56 | 84.62 ±0.49 | 62.41 ±0.67 | 82.96 ±0.79 | 84.35 ±0.43 | 78.10 ±0.80 | 76.42 ±0.23 | 82.69 ±0.25 | 87.90 ±0.53 | 82.34 ±0.32 | 73.56 ±0.25 | 76.09 ±0.44 | 68.39 ±0.31 | 75.00 ±0.50 |
| | + SNAP-TTA | 65.27 ±0.32 | 67.05 ±0.19 | 58.05 ±0.22 | 86.79 ±0.21 | 63.48 ±0.18 | 85.46 ±0.33 | 86.25 ±0.09 | 80.39 ±0.08 | 78.34 ±0.15 | 85.19 ±0.10 | 90.10 ±0.16 | 85.94 ±0.08 | 75.67 ±0.12 | 78.04 ±0.09 | 69.75 ±0.27 | 77.05 ±0.17 |
| | Tent | 62.43 ±1.70 | 64.13 ±1.51 | 55.85 ±1.35 | 84.03 ±1.07 | 62.21 ±1.20 | 82.47 ±0.88 | 83.87 ±0.66 | 77.71 ±0.93 | 76.55 ±0.66 | 82.75 ±0.18 | 87.35 ±1.11 | 81.83 ±1.81 | 73.24 ±1.33 | 75.34 ±1.18 | 67.73 ±1.50 | 74.50 ±1.10 |
| | + SNAP-TTA | 65.51 ±0.24 | 67.26 ±0.31 | 58.05 ±0.34 | 86.89 ±0.07 | 63.53 ±0.33 | 85.44 ±0.20 | 85.97 ±0.12 | 80.58 ±0.12 | 78.35 ±0.16 | 85.12 ±0.21 | 90.09 ±0.11 | 85.86 ±0.08 | 75.66 ±0.20 | 78.38 ±0.33 | 70.12 ±0.20 | 77.12 ±0.21 |
| | CoTTA | 59.75 ±4.69 | 59.44 ±6.21 | 54.47 ±5.57 | 71.12 ±5.10 | 57.11 ±4.35 | 72.47 ±4.52 | 72.83 ±4.80 | 66.05 ±7.60 | 65.14 ±7.65 | 69.75 ±9.79 | 75.12 ±6.79 | 64.31 ±6.46 | 66.22 ±4.50 | 62.65 ±5.27 | 64.76 ±5.36 | 65.41 ±5.91 |
| | + SNAP-TTA | 71.79 ±0.22 | 73.61 ±0.29 | 65.98 ±0.58 | 85.34 ±0.36 | 66.76 ±0.26 | 84.26 ±0.12 | 84.93 ±0.21 | 80.64 ±0.45 | 80.38 ±0.30 | 83.94 ±0.42 | 89.98 ±0.08 | 82.47 ±0.64 | 76.48 ±0.26 | 79.61 ±0.24 | 75.60 ±0.29 | 78.79 ±0.31 |
| 0.01 | EATA | 62.36 ±1.73 | 63.92 ±1.66 | 55.73 ±1.39 | 84.05 ±1.10 | 62.24 ±1.18 | 82.38 ±0.85 | 83.90 ±0.93 | 77.66 ±0.72 | 76.48 ±0.15 | 82.67 ±0.17 | 87.34 ±1.12 | 81.82 ±1.81 | 73.30 ±1.24 | 75.31 ±1.20 | 67.76 ±1.52 | 74.46 ±1.12 |
| | + SNAP-TTA | 65.49 ±0.29 | 67.19 ±0.04 | 57.93 ±0.40 | 86.92 ±0.41 | 63.65 ±0.18 | 85.42 ±0.28 | 85.97 ±0.24 | 80.46 ±0.18 | 78.13 ±0.27 | 85.07 ±0.13 | 90.03 ±0.10 | 85.87 ±0.20 | 75.69 ±0.11 | 78.20 ±0.13 | 70.03 ±0.46 | 77.07 ±0.23 |
| | SAR | 62.50 ±1.69 | 64.13 ±1.83 | 55.65 ±1.38 | 82.30 ±3.37 | 62.22 ±1.21 | 77.21 ±6.27 | 80.11 ±6.19 | 77.66 ±0.80 | 76.75 ±0.34 | 79.12 ±3.28 | 89.45 ±1.79 | 81.97 ±1.97 | 73.39 ±1.21 | 69.39 ±5.48 | 67.83 ±1.65 | 73.31 ±2.57 |
| | + SNAP-TTA | 65.06 ±0.17 | 66.93 ±0.11 | 57.66 ±0.51 | 86.76 ±0.29 | 62.78 ±0.24 | 85.05 ±0.21 | 85.94 ±0.48 | 79.95 ±0.18 | 77.62 ±0.37 | 84.65 ±0.21 | 90.72 ±0.62 | 85.48 ±0.35 | 75.34 ±0.13 | 75.72 ±1.35 | 69.61 ±0.25 | 76.62 ±0.36 |
| | RoTTA | 62.25 ±1.65 | 63.71 ±1.68 | 55.59 ±1.46 | 84.05 ±1.12 | 62.17 ±1.37 | 82.32 ±0.83 | 83.86 ±0.90 | 77.56 ±0.75 | 76.39 ±0.24 | 82.64 ±0.10 | 87.27 ±1.12 | 81.75 ±1.82 | 73.21 ±1.21 | 75.15 ±1.27 | 67.75 ±1.48 | 74.38 ±1.13 |
| | + SNAP-TTA | 65.32 ±0.25 | 66.94 ±0.12 | 57.85 ±0.29 | 86.91 ±0.31 | 63.44 ±0.24 | 85.32 ±0.22 | 85.98 ±0.14 | 80.49 ±0.24 | 78.22 ±0.20 | 85.04 ±0.15 | 90.01 ±0.06 | 85.77 ±0.24 | 75.75 ±0.11 | 78.15 ±0.07 | 70.06 ±0.47 | 77.02 ±0.21 |

## C.3 CIFAR100-C

Table 10: STTA classification accuracy (%) comparing with and without SNAP-TTA on CIAFR100-C through Adaptation Rates(AR) (0.5, 0.3, and 0.1), including results for full adaptation (AR=1). **Bold** numbers are the highest accuracy.

| AR | Methods | Gau. | Shot | Imp. | Def. | Gla. | Mot. | Zoom | Snow | Fro. | Fog | Brit. | Cont. | Elas. | Pix. | JPEG | Avg. |
|----|---------|------|------|------|------|------|------|------|------|------|-----|-------|-------|-------|------|------|------|
| 1 | Source | 10.26 ±0.00 | 11.87 ±0.00 | 6.48 ±0.00 | 35.16 ±0.00 | 20.33 ±0.00 | 44.42 ±0.00 | 42.13 ±0.00 | 45.99 ±0.00 | 34.84 ±0.00 | 41.12 ±0.00 | 66.37 ±0.00 | 19.54 ±0.00 | 50.59 ±0.00 | 22.68 ±0.00 | 45.48 ±0.00 | 33.15 ±0.00 |
| | BN stats | 36.90 ±0.10 | 37.96 ±0.24 | 32.13 ±0.44 | 62.65 ±0.26 | 39.14 ±0.19 | 60.05 ±0.42 | 61.16 ±0.05 | 50.68 ±0.13 | 50.38 ±0.09 | 54.81 ±0.24 | 64.40 ±0.05 | 60.33 ±0.12 | 50.48 ±0.24 | 53.49 ±0.11 | 41.98 ±0.49 | 50.44 ±0.21 |
| | Tent | 46.71 ±0.29 | 48.06 ±0.47 | 40.98 ±0.13 | 65.19 ±0.40 | 44.10 ±0.41 | 62.78 ±0.24 | 63.95 ±0.23 | 55.43 ±0.36 | 55.46 ±0.49 | 59.32 ±0.30 | 67.43 ±0.17 | 63.83 ±0.42 | 53.89 ±0.15 | 59.40 ±0.32 | 49.91 ±0.66 | 55.76 ±0.33 |
| | CoTTA | 42.14 ±0.34 | 42.92 ±0.44 | 37.92 ±0.18 | 55.40 ±0.12 | 41.01 ±0.39 | 55.18 ±0.10 | 55.39 ±0.58 | 49.46 ±0.23 | 50.61 ±0.63 | 50.86 ±0.31 | 61.35 ±0.27 | 47.44 ±0.37 | 48.69 ±0.18 | 54.38 ±0.16 | 48.11 ±0.65 | 49.39 ±0.33 |
| | EATA | 38.42 ±0.41 | 39.96 ±0.47 | 32.64 ±0.71 | 62.35 ±0.41 | 38.73 ±0.33 | 59.93 ±0.17 | 61.07 ±0.19 | 50.50 ±0.36 | 50.79 ±0.34 | 55.30 ±0.23 | 64.38 ±0.12 | 60.63 ±0.13 | 49.66 ±0.32 | 53.63 ±0.41 | 43.02 ±0.20 | 50.74 ±0.32 |
| | SAR | 50.75 ±0.44 | 52.00 ±0.22 | 43.87 ±0.40 | 65.44 ±0.39 | 46.30 ±0.22 | 63.60 ±0.17 | 64.68 ±0.09 | 58.41 ±0.48 | 58.26 ±0.09 | 61.34 ±0.40 | 68.03 ±0.15 | 67.68 ±0.31 | 54.53 ±0.25 | 61.52 ±0.21 | 52.72 ±0.21 | 57.94 ±0.27 |
| | RoTTA | 38.54 ±0.22 | 39.85 ±0.24 | 33.73 ±0.37 | 63.45 ±0.17 | 40.74 ±0.32 | 60.54 ±0.19 | 62.03 ±0.26 | 51.61 ±0.09 | 51.75 ±0.14 | 56.20 ±0.31 | 65.14 ±0.10 | 61.55 ±0.14 | 51.22 ±0.14 | 54.42 ±0.22 | 42.50 ±0.35 | 51.55 ±0.22 |
| 0.5 | Tent | 43.96 ±0.85 | 45.42 ±1.34 | 36.57 ±1.57 | 62.28 ±0.13 | 36.57 ±2.97 | 59.96 ±0.59 | 61.90 ±0.48 | 53.25 ±0.72 | 53.14 ±1.70 | 57.36 ±0.22 | 65.20 ±0.20 | 60.14 ±2.77 | 49.72 ±0.08 | 57.62 ±0.61 | 46.83 ±0.52 | 52.66 ±0.98 |
| | + SNAP-TTA | **49.06** ±0.00 | **50.43** ±0.13 | **41.49** ±0.80 | **65.55** ±0.24 | **44.09** ±0.06 | **63.31** ±0.53 | **65.62** ±0.37 | **57.62** ±0.09 | **56.81** ±0.31 | **60.75** ±0.48 | **68.72** ±0.31 | **67.52** ±0.64 | **54.08** ±0.19 | **61.15** ±0.14 | **51.54** ±0.11 | **57.18** ±0.29 |
| | CoTTA | 34.31 ±0.09 | 35.16 ±0.46 | 31.42 ±0.28 | 47.78 ±0.45 | 34.99 ±0.40 | 48.91 ±0.48 | 47.79 ±0.46 | 41.27 ±0.86 | 41.42 ±0.37 | 43.77 ±0.57 | 52.16 ±0.27 | 38.30 ±0.46 | 42.25 ±0.49 | 44.12 ±0.41 | 41.58 ±0.22 | 41.68 ±0.42 |
| | + SNAP-TTA | 41.28 ±0.46 | 42.23 ±0.16 | 37.17 ±0.19 | 58.29 ±0.21 | 40.70 ±0.08 | 57.32 ±0.12 | 57.78 ±0.09 | 49.85 ±0.38 | 50.82 ±0.11 | 52.21 ±0.28 | 63.69 ±0.18 | 51.30 ±0.23 | 49.41 ±0.14 | 55.15 ±0.09 | 47.92 ±0.25 | 50.34 ±0.20 |
| | EATA | 38.02 ±0.22 | 39.48 ±0.15 | 32.77 ±0.17 | 61.68 ±0.38 | 38.42 ±0.07 | 59.11 ±0.09 | 60.63 ±0.18 | 50.15 ±0.25 | 49.92 ±0.13 | 54.60 ±0.21 | 63.43 ±0.09 | 58.70 ±0.44 | 49.42 ±0.22 | 53.08 ±0.20 | 42.62 ±0.21 | 50.13 ±0.24 |
| | + SNAP-TTA | 39.75 ±0.11 | 41.14 ±0.26 | 34.15 ±0.10 | 63.75 ±0.23 | 40.55 ±0.21 | 61.09 ±0.08 | 62.81 ±0.19 | 52.12 ±0.08 | 52.12 ±0.30 | 56.47 ±0.18 | 65.73 ±0.23 | 61.85 ±0.34 | 51.14 ±0.28 | 55.75 ±0.15 | 44.86 ±0.51 | 52.22 ±0.22 |
| | SAR | 49.00 ±0.61 | 50.00 ±0.42 | 42.99 ±0.30 | 65.10 ±0.44 | 45.21 ±0.41 | 62.51 ±0.20 | 64.43 ±0.43 | 55.78 ±0.27 | 56.59 ±0.46 | 60.21 ±0.48 | 67.33 ±0.44 | 65.17 ±0.46 | 53.90 ±0.50 | 60.22 ±0.29 | 51.28 ±0.23 | 56.65 ±0.40 |
| | + SNAP-TTA | **51.71** ±0.46 | **52.79** ±0.08 | **44.95** ±0.54 | **66.59** ±0.10 | **47.84** ±0.01 | **64.40** ±0.18 | **66.15** ±0.28 | **59.00** ±0.20 | **59.12** ±0.37 | **62.62** ±0.16 | **69.15** ±0.06 | **68.20** ±0.16 | **55.89** ±0.26 | **62.66** ±0.31 | **53.77** ±0.23 | **58.99** ±0.23 |
| | RoTTA | 37.12 ±0.09 | 38.34 ±0.20 | 32.54 ±0.22 | 62.25 ±0.09 | 38.91 ±0.13 | 59.52 ±0.19 | 61.19 ±0.21 | 50.22 ±0.23 | 49.91 ±0.56 | 54.69 ±0.15 | 63.74 ±0.47 | 59.40 ±0.29 | 50.32 ±0.29 | 53.29 ±0.15 | 41.94 ±0.23 | 50.22 ±0.23 |
| | + SNAP-TTA | 38.33 ±0.30 | 39.12 ±0.24 | 32.93 ±0.28 | 64.01 ±0.15 | 40.36 ±0.44 | 61.30 ±0.38 | 62.96 ±0.16 | 51.77 ±0.22 | 51.54 ±0.19 | 56.15 ±0.28 | 66.13 ±0.05 | 61.67 ±0.17 | 51.60 ±0.24 | 54.90 ±0.23 | 43.14 ±0.36 | 51.73 ±0.25 |
| 0.3 | Tent | 44.41 ±0.80 | 46.79 ±0.72 | 38.72 ±1.17 | 62.98 ±0.28 | 39.79 ±0.92 | 60.38 ±0.53 | 62.25 ±0.33 | 52.47 ±0.76 | 53.69 ±0.65 | 57.47 ±0.63 | 65.80 ±0.28 | 60.13 ±2.70 | 50.03 ±0.60 | 58.21 ±0.81 | 47.23 ±0.43 | 53.36 ±0.77 |
| | + SNAP-TTA | **49.23** ±0.04 | **50.15** ±0.48 | **42.19** ±0.75 | **65.85** ±0.15 | **45.12** ±1.15 | **63.39** ±0.28 | **64.91** ±0.26 | **57.45** ±0.51 | **57.13** ±0.37 | **60.72** ±0.17 | **68.86** ±0.31 | **66.65** ±1.52 | **54.25** ±0.41 | **61.38** ±0.54 | **51.80** ±0.68 | **57.27** ±0.51 |
| | CoTTA | 31.74 ±0.43 | 32.66 ±0.38 | 29.28 ±0.15 | 44.98 ±0.45 | 32.96 ±0.56 | 46.51 ±0.48 | 44.96 ±0.37 | 38.57 ±0.90 | 38.16 ±0.78 | 41.91 ±0.39 | 49.38 ±0.86 | 35.53 ±0.33 | 40.04 ±0.61 | 40.77 ±0.67 | 39.12 ±0.43 | 39.11 ±0.52 |
| | + SNAP-TTA | 41.44 ±0.38 | 42.49 ±0.09 | 37.08 ±0.13 | 58.27 ±0.24 | 40.99 ±0.37 | 57.24 ±0.37 | 57.68 ±0.17 | 50.36 ±0.22 | 51.09 ±0.18 | 51.66 ±0.22 | 63.50 ±0.13 | 50.90 ±0.52 | 49.49 ±0.26 | 54.75 ±0.42 | 47.81 ±0.13 | 50.32 ±0.26 |
| | EATA | 37.97 ±0.04 | 39.47 ±0.34 | 32.69 ±0.12 | 61.45 ±0.19 | 37.96 ±0.17 | 59.02 ±0.28 | 60.79 ±0.12 | 49.73 ±0.05 | 49.55 ±0.38 | 54.63 ±0.41 | 63.38 ±0.07 | 58.16 ±0.21 | 49.07 ±0.24 | 53.17 ±0.41 | 42.49 ±0.44 | 49.97 ±0.23 |
| | + SNAP-TTA | 40.03 ±0.26 | 41.39 ±0.29 | 34.91 ±0.58 | 63.58 ±0.15 | 40.29 ±0.28 | 61.58 ±0.12 | 62.56 ±0.25 | 51.85 ±0.25 | 51.78 ±0.21 | 56.13 ±0.01 | 65.70 ±0.20 | 61.68 ±0.29 | 51.25 ±0.35 | 55.28 ±0.23 | 44.80 ±0.17 | 52.19 ±0.24 |
| | SAR | 49.00 ±0.61 | 50.00 ±0.42 | 42.99 ±0.30 | 65.10 ±0.44 | 45.21 ±0.41 | 62.51 ±0.20 | 64.43 ±0.43 | 55.78 ±0.27 | 56.59 ±0.46 | 60.21 ±0.48 | 67.33 ±0.44 | 65.17 ±0.46 | 53.90 ±0.50 | 60.22 ±0.29 | 51.28 ±0.23 | 56.65 ±0.40 |
| | + SNAP-TTA | **50.63** ±0.31 | **52.03** ±0.32 | **44.89** ±0.54 | **66.28** ±0.13 | **47.08** ±0.26 | **64.32** ±0.09 | **65.90** ±0.21 | **57.98** ±0.27 | **58.09** ±0.49 | **61.88** ±0.24 | **69.17** ±0.42 | **67.82** ±0.29 | **55.47** ±0.29 | **62.02** ±0.31 | **53.09** ±0.15 | **58.44** ±0.29 |
| | RoTTA | 36.83 ±0.18 | 37.94 ±0.22 | 32.00 ±0.05 | 61.90 ±0.20 | 38.67 ±0.10 | 59.15 ±0.14 | 60.97 ±0.24 | 49.92 ±0.23 | 49.32 ±0.38 | 54.62 ±0.21 | 63.71 ±0.18 | 58.31 ±0.11 | 49.79 ±0.22 | 52.88 ±0.34 | 41.59 ±0.27 | 49.84 ±0.21 |
| | + SNAP-TTA | 38.11 ±0.13 | 39.21 ±0.23 | 32.80 ±0.13 | 63.72 ±0.23 | 40.01 ±0.13 | 61.51 ±0.16 | 62.74 ±0.13 | 51.37 ±0.16 | 51.49 ±0.15 | 55.68 ±0.30 | 65.90 ±0.25 | 61.56 ±0.29 | 51.50 ±0.08 | 54.67 ±0.13 | 43.01 ±0.19 | 51.55 ±0.18 |
| 0.1 | Tent | 43.55 ±0.66 | 44.25 ±0.54 | 37.95 ±0.72 | 62.56 ±0.47 | 41.80 ±0.04 | 59.45 ±0.20 | 62.13 ±0.21 | 53.04 ±0.84 | 51.60 ±0.39 | 56.76 ±0.15 | 64.60 ±0.56 | 61.19 ±1.68 | 51.01 ±0.39 | 56.42 ±0.27 | 46.28 ±0.49 | 52.84 ±0.51 |
| | + SNAP-TTA | **46.51** ±0.35 | **47.68** ±0.23 | **39.92** ±0.48 | **65.39** ±0.11 | **44.14** ±0.60 | **63.29** ±0.18 | **64.53** ±0.38 | **55.20** ±0.47 | **55.55** ±0.11 | **59.71** ±0.33 | **68.05** ±0.17 | **64.90** ±0.90 | **53.91** ±0.30 | **59.28** ±0.16 | **49.58** ±0.75 | **55.84** ±0.37 |
| | CoTTA | 28.53 ±0.90 | 29.53 ±0.86 | 26.45 ±0.60 | 42.19 ±1.19 | 30.34 ±0.77 | 44.69 ±1.07 | 41.88 ±0.62 | 34.44 ±0.84 | 33.93 ±1.07 | 39.03 ±0.09 | 45.49 ±0.60 | 31.17 ±0.80 | 37.25 ±1.20 | 36.17 ±0.71 | 36.84 ±0.90 | 35.86 ±0.90 |
| | + SNAP-TTA | 41.72 ±0.25 | 42.62 ±0.60 | 37.46 ±0.13 | 58.43 ±0.13 | 41.24 ±0.21 | 57.33 ±0.07 | 57.96 ±0.30 | 50.34 ±0.38 | 51.17 ±0.18 | 52.29 ±0.16 | 63.59 ±0.20 | 51.32 ±0.36 | 49.68 ±0.21 | 54.78 ±0.28 | 47.89 ±0.35 | 50.52 ±0.25 |
| | EATA | 38.41 ±0.53 | 39.03 ±0.45 | 32.29 ±0.32 | 61.07 ±0.36 | 38.45 ±0.29 | 58.21 ±0.47 | 60.62 ±0.36 | 49.59 ±0.30 | 49.19 ±0.34 | 54.23 ±0.50 | 62.88 ±0.28 | 57.39 ±0.62 | 49.00 ±0.65 | 53.01 ±0.60 | 42.05 ±0.15 | 49.70 ±0.42 |
| | + SNAP-TTA | 40.62 ±0.26 | 41.53 ±0.49 | 34.31 ±0.24 | 64.08 ±0.30 | 40.29 ±0.21 | 61.32 ±0.24 | 63.04 ±0.16 | 52.00 ±0.53 | 51.77 ±0.40 | 56.85 ±0.43 | 65.98 ±0.09 | 61.96 ±0.34 | 51.05 ±0.09 | 55.67 ±0.28 | 44.80 ±0.15 | 52.35 ±0.28 |
| | SAR | 43.92 ±0.52 | 45.28 ±0.55 | 38.64 ±0.28 | 63.36 ±0.44 | 42.58 ±0.28 | 60.36 ±0.42 | 62.78 ±0.23 | 53.39 ±0.86 | 52.23 ±0.32 | 57.54 ±0.41 | 65.41 ±0.88 | 60.88 ±0.59 | 52.07 ±0.13 | 56.80 ±0.20 | 47.16 ±0.43 | 53.49 ±0.43 |
| | + SNAP-TTA | **46.29** ±0.68 | **47.60** ±0.06 | **39.95** ±0.21 | **65.26** ±0.18 | **44.00** ±0.22 | **63.09** ±0.25 | **64.97** ±0.36 | **55.08** ±0.24 | **55.17** ±0.17 | **59.73** ±0.24 | **68.13** ±0.09 | **64.72** ±0.44 | **53.84** ±0.31 | **58.98** ±0.35 | **49.54** ±0.65 | **55.76** ±0.30 |
| | RoTTA | 36.28 ±0.15 | 37.12 ±0.41 | 31.38 ±0.27 | 61.20 ±0.07 | 38.36 ±0.15 | 58.26 ±0.24 | 60.30 ±0.47 | 49.20 ±0.23 | 48.21 ±0.14 | 53.54 ±0.23 | 62.80 ±0.40 | 56.78 ±0.51 | 49.61 ±0.24 | 52.28 ±0.41 | 41.26 ±0.11 | 49.11 ±0.27 |
| | + SNAP-TTA | 37.83 ±0.13 | 38.42 ±0.36 | 32.38 ±0.20 | 63.73 ±0.09 | 39.72 ±0.38 | 61.32 ±0.18 | 62.58 ±0.19 | 51.38 ±0.18 | 51.18 ±0.13 | 55.61 ±0.07 | 65.70 ±0.29 | 61.39 ±0.21 | 51.36 ±0.09 | 54.51 ±0.24 | 42.85 ±0.33 | 51.33 ±0.21 |

Table 11: STTA classification accuracy (%) comparing with and without SNAP-TTA on CIFAR100-C through Adaptation Rates(AR) (0.05, 0.03, and 0.01). **Bold** numbers are the highest accuracy.

| AR | Methods | Gau. | Shot | Imp. | Def. | Gla. | Mot. | Zoom | Snow | Fro. | Fog | Brit. | Cont. | Elas. | Pix. | JPEG | Avg. |
|---|---|---|---|---|---|---|---|---|---|---|---|---|---|---|---|---|---|
| 0.05 | Tent | 40.69 ±0.35 | 41.55 ±0.62 | 35.14 ±0.38 | 62.26 ±0.52 | 40.26 ±0.23 | 58.92 ±0.60 | 61.06 ±0.43 | 51.21 ±0.88 | 50.00 ±0.31 | 55.52 ±0.33 | 64.05 ±0.62 | 58.45 ±1.06 | 50.50 ±0.80 | 54.68 ±0.26 | 44.36 ±0.69 | 51.24 ±0.54 |
| | + SNAP-TTA | **42.87** ±0.37 | **44.87** ±0.70 | **37.60** ±0.08 | **65.01** ±0.01 | **42.22** ±0.35 | **62.22** ±0.31 | **63.72** ±0.45 | **54.03** ±0.46 | **53.68** ±0.39 | **58.03** ±0.47 | **67.05** ±0.50 | **63.08** ±0.10 | **52.97** ±0.15 | **57.67** ±0.12 | **46.94** ±0.13 | **54.13** ±0.31 |
| | CoTTA | 26.15 ±0.60 | 26.89 ±0.32 | 25.26 ±0.44 | 39.48 ±0.71 | 28.34 ±0.74 | 41.41 ±0.76 | 38.77 ±1.14 | 32.06 ±0.85 | 30.84 ±0.65 | 35.56 ±1.12 | 41.60 ±1.36 | 28.52 ±0.79 | 34.99 ±0.45 | 33.60 ±0.82 | 34.54 ±0.54 | 33.20 ±0.75 |
| | + SNAP-TTA | **42.02** ±0.21 | **42.70** ±0.13 | **37.67** ±0.31 | **58.30** ±0.26 | **41.57** ±0.37 | **57.47** ±0.14 | **58.02** ±0.18 | **50.55** ±0.27 | **51.31** ±0.32 | **52.34** ±0.17 | **63.63** ±0.16 | **51.25** ±0.49 | **49.76** ±0.18 | **54.94** ±0.05 | **47.98** ±0.12 | **50.63** ±0.22 |
| | EATA | 38.46 ±0.14 | 39.05 ±0.58 | 33.47 ±0.23 | 61.07 ±0.63 | 38.52 ±0.29 | 58.16 ±0.46 | 60.59 ±0.48 | 49.60 ±0.55 | 49.18 ±0.47 | 54.41 ±0.24 | 63.15 ±0.43 | 57.06 ±1.37 | 49.09 ±0.88 | 52.87 ±0.42 | 42.49 ±0.34 | 49.81 ±0.50 |
| | + SNAP-TTA | **40.49** ±0.21 | **41.64** ±0.43 | **34.37** ±0.15 | **64.28** ±0.20 | **40.38** ±0.51 | **61.52** ±0.30 | **63.17** ±0.18 | **51.66** ±0.53 | **52.12** ±0.52 | **56.50** ±0.21 | **66.03** ±0.36 | **62.01** ±0.12 | **51.76** ±0.12 | **55.66** ±0.23 | **44.83** ±0.32 | **52.43** ±0.29 |
| | SAR | 40.28 ±0.07 | 41.62 ±0.62 | 35.35 ±0.04 | 62.84 ±0.26 | 40.37 ±0.41 | 59.51 ±0.38 | 61.68 ±0.28 | 51.29 ±0.81 | 50.66 ±0.38 | 55.60 ±0.40 | 64.43 ±0.62 | 58.49 ±0.82 | 50.90 ±0.64 | 54.82 ±0.27 | 44.64 ±0.43 | 51.50 ±0.43 |
| | + SNAP-TTA | **41.76** ±0.29 | **44.24** ±0.44 | **36.89** ±0.21 | **64.34** ±0.38 | **41.54** ±0.37 | **62.13** ±0.15 | **63.39** ±0.24 | **53.24** ±0.33 | **52.91** ±0.02 | **57.54** ±0.22 | **66.89** ±0.60 | **62.41** ±0.50 | **52.70** ±0.15 | **57.23** ±0.47 | **46.63** ±0.57 | **53.59** ±0.33 |
| | RoTTA | 36.38 ±0.12 | 37.38 ±0.42 | 31.78 ±0.45 | 61.44 ±0.06 | 38.26 ±0.20 | 58.18 ±0.42 | 60.19 ±0.53 | 48.98 ±0.18 | 48.30 ±0.28 | 53.50 ±0.17 | 62.73 ±0.42 | 56.52 ±0.90 | 49.37 ±0.49 | 52.19 ±0.19 | 41.60 ±0.28 | 49.12 ±0.34 |
| | + SNAP-TTA | **37.67** ±0.12 | **38.66** ±0.21 | **32.47** ±0.12 | **63.95** ±0.16 | **40.18** ±0.20 | **61.33** ±0.47 | **62.52** ±0.35 | **51.47** ±0.14 | **51.32** ±0.36 | **55.67** ±0.21 | **65.89** ±0.24 | **61.24** ±0.15 | **51.47** ±0.14 | **54.52** ±0.15 | **42.84** ±0.38 | **51.41** ±0.23 |
| 0.03 | Tent | 38.55 ±0.17 | 39.28 ±0.15 | 33.77 ±0.16 | 61.64 ±0.25 | 39.66 ±0.39 | 58.83 ±0.48 | 60.89 ±0.29 | 49.45 ±0.51 | 49.51 ±0.78 | 54.64 ±0.42 | 63.48 ±0.58 | 57.29 ±0.33 | 50.34 ±0.34 | 53.44 ±0.38 | 43.28 ±0.26 | 50.27 ±0.37 |
| | + SNAP-TTA | **41.22** ±0.33 | **42.20** ±0.27 | **35.31** ±0.36 | **64.48** ±0.06 | **40.82** ±0.60 | **61.96** ±0.02 | **63.50** ±0.30 | **52.84** ±0.40 | **52.36** ±0.40 | **57.18** ±0.33 | **66.50** ±0.02 | **62.17** ±0.41 | **52.12** ±0.17 | **56.48** ±0.18 | **45.72** ±0.40 | **52.99** ±0.28 |
| | CoTTA | 27.11 ±1.11 | 27.73 ±2.05 | 25.87 ±1.41 | 40.25 ±2.62 | 29.52 ±1.49 | 42.16 ±2.21 | 39.60 ±2.51 | 32.74 ±2.42 | 32.23 ±1.71 | 36.60 ±2.75 | 43.33 ±2.80 | 29.13 ±2.42 | 36.45 ±1.82 | 34.51 ±1.66 | 35.96 ±1.75 | 34.21 ±2.05 |
| | + SNAP-TTA | **41.77** ±0.24 | **42.85** ±0.19 | **37.50** ±0.08 | **58.61** ±0.22 | **41.15** ±0.16 | **57.65** ±0.22 | **58.05** ±0.32 | **50.45** ±0.65 | **51.34** ±0.20 | **52.72** ±0.35 | **63.49** ±0.07 | **51.63** ±0.61 | **49.87** ±0.17 | **55.24** ±0.13 | **48.14** ±0.36 | **50.70** ±0.26 |
| | EATA | 37.94 ±0.32 | 38.63 ±0.21 | 32.00 ±0.91 | 61.02 ±0.33 | 39.08 ±0.30 | 58.52 ±0.66 | 60.28 ±0.42 | 48.73 ±0.32 | 49.15 ±0.97 | 53.89 ±0.53 | 63.03 ±0.34 | 56.64 ±0.49 | 49.45 ±0.47 | 52.93 ±0.35 | 42.11 ±0.44 | 49.56 ±0.47 |
| | + SNAP-TTA | **39.87** ±0.89 | **41.12** ±0.20 | **34.48** ±0.08 | **64.14** ±0.23 | **40.27** ±0.09 | **61.91** ±0.00 | **63.09** ±0.43 | **52.37** ±0.42 | **51.93** ±0.44 | **56.36** ±0.26 | **66.02** ±0.05 | **61.88** ±0.15 | **51.83** ±0.04 | **55.60** ±0.11 | **44.59** ±0.45 | **52.36** ±0.26 |
| | SAR | 38.33 ±0.25 | 39.19 ±0.26 | 33.15 ±0.43 | 61.77 ±0.21 | 39.78 ±0.06 | 59.09 ±0.33 | 61.02 ±0.25 | 49.67 ±0.54 | 49.86 ±0.65 | 54.71 ±0.31 | 63.59 ±0.49 | 57.45 ±0.18 | 50.37 ±0.39 | 53.67 ±0.32 | 42.88 ±0.51 | 50.30 ±0.35 |
| | + SNAP-TTA | **39.84** ±0.07 | **41.83** ±0.78 | **34.94** ±0.28 | **63.70** ±0.28 | **40.49** ±0.16 | **61.45** ±0.28 | **63.17** ±0.07 | **52.27** ±0.51 | **51.91** ±0.07 | **56.69** ±0.05 | **65.91** ±0.27 | **61.31** ±0.52 | **51.68** ±0.22 | **56.06** ±0.18 | **44.95** ±0.16 | **52.41** ±0.28 |
| | RoTTA | 36.24 ±0.03 | 36.94 ±0.21 | 31.15 ±0.09 | 60.87 ±0.17 | 38.28 ±0.14 | 58.25 ±0.53 | 59.88 ±0.36 | 48.43 ±0.52 | 48.17 ±0.47 | 53.32 ±0.46 | 62.73 ±0.34 | 56.18 ±0.39 | 49.23 ±0.61 | 52.12 ±0.31 | 41.28 ±0.61 | 48.87 ±0.35 |
| | + SNAP-TTA | **37.85** ±0.20 | **38.68** ±0.20 | **32.78** ±0.31 | **63.97** ±0.24 | **39.75** ±0.17 | **61.41** ±0.16 | **62.57** ±0.52 | **51.53** ±0.27 | **51.38** ±0.28 | **55.68** ±0.37 | **65.56** ±0.20 | **61.25** ±0.13 | **51.53** ±0.19 | **54.84** ±0.26 | **42.96** ±0.33 | **51.45** ±0.25 |
| 0.01 | Tent | 36.08 ±0.42 | 36.95 ±0.21 | 31.31 ±0.47 | 61.03 ±0.51 | 38.09 ±0.56 | 57.63 ±0.53 | 58.76 ±0.31 | 48.24 ±0.47 | 48.65 ±0.87 | 53.45 ±0.19 | 62.14 ±0.49 | 55.07 ±2.13 | 48.59 ±0.25 | 51.82 ±0.58 | 40.68 ±0.04 | 48.57 ±0.54 |
| | + SNAP-TTA | **38.40** ±0.06 | **39.40** ±0.16 | **33.26** ±0.10 | **63.85** ±0.11 | **40.36** ±0.36 | **61.23** ±0.34 | **62.79** ±0.24 | **51.92** ±0.06 | **51.73** ±0.00 | **56.20** ±0.34 | **65.83** ±0.17 | **60.95** ±0.29 | **51.82** ±0.00 | **54.75** ±0.30 | **43.53** ±0.16 | **51.73** ±0.18 |
| | CoTTA | 26.59 ±1.64 | 27.92 ±1.79 | 24.86 ±1.51 | 41.34 ±2.21 | 28.91 ±1.96 | 43.09 ±2.85 | 40.11 ±2.87 | 34.33 ±1.61 | 33.32 ±2.67 | 37.99 ±2.03 | 44.78 ±3.61 | 28.80 ±2.18 | 36.26 ±1.90 | 34.70 ±1.66 | 35.67 ±1.47 | 34.58 ±2.13 |
| | + SNAP-TTA | **42.05** ±0.05 | **42.91** ±0.17 | **37.50** ±0.08 | **58.70** ±0.12 | **41.22** ±0.36 | **57.38** ±0.17 | **58.14** ±0.33 | **50.39** ±0.68 | **51.13** ±0.43 | **52.23** ±0.12 | **63.42** ±0.35 | **51.74** ±0.17 | **49.87** ±0.50 | **54.84** ±0.09 | **47.72** ±0.25 | **50.62** ±0.26 |
| | EATA | 36.10 ±0.27 | 37.05 ±0.59 | 31.03 ±0.34 | 60.86 ±0.50 | 37.83 ±0.37 | 57.64 ±0.57 | 58.77 ±0.42 | 48.02 ±0.50 | 48.75 ±1.26 | 53.37 ±0.09 | 62.18 ±0.43 | 54.95 ±2.22 | 48.55 ±0.15 | 51.89 ±0.65 | 40.75 ±0.02 | 48.51 ±0.55 |
| | + SNAP-TTA | **38.54** ±0.14 | **39.78** ±0.15 | **33.11** ±0.22 | **63.82** ±0.10 | **39.98** ±0.53 | **61.33** ±0.20 | **62.53** ±0.24 | **51.76** ±0.12 | **51.50** ±0.32 | **56.03** ±0.44 | **65.94** ±0.19 | **61.16** ±0.11 | **51.47** ±0.04 | **54.52** ±0.27 | **43.67** ±0.04 | **51.68** ±0.21 |
| | SAR | 36.04 ±0.00 | 37.02 ±0.26 | 31.38 ±0.30 | 61.13 ±0.35 | 38.07 ±0.44 | 58.00 ±0.59 | 59.08 ±0.36 | 48.44 ±0.47 | 48.84 ±0.92 | 53.52 ±0.16 | 62.57 ±0.50 | 55.19 ±2.20 | 48.87 ±0.15 | 52.01 ±0.57 | 40.71 ±0.19 | 48.72 ±0.50 |
| | + SNAP-TTA | **37.91** ±0.39 | **38.85** ±0.25 | **32.92** ±0.38 | **63.17** ±0.23 | **39.35** ±0.45 | **60.51** ±0.51 | **62.01** ±0.26 | **51.11** ±0.11 | **50.48** ±0.28 | **55.47** ±0.41 | **65.07** ±0.16 | **59.69** ±0.15 | **51.24** ±0.15 | **54.10** ±0.47 | **42.80** ±0.06 | **50.98** ±0.28 |
| | RoTTA | 35.55 ±0.33 | 36.34 ±0.31 | 30.55 ±0.45 | 60.76 ±0.50 | 37.42 ±0.50 | 57.50 ±0.56 | 58.57 ±0.30 | 47.87 ±0.28 | 48.31 ±0.97 | 53.11 ±0.23 | 61.90 ±1.98 | 54.70 ±0.08 | 48.25 ±0.62 | 51.37 ±0.11 | 40.29 ±0.52 | 48.16 |
| | + SNAP-TTA | **37.82** ±0.16 | **38.72** ±0.05 | **32.60** ±0.10 | **63.53** ±0.01 | **39.80** ±0.49 | **61.00** ±0.37 | **62.27** ±0.23 | **51.42** ±0.06 | **51.33** ±0.12 | **55.71** ±0.42 | **65.64** ±0.14 | **60.89** ±0.18 | **51.50** ±0.18 | **54.27** ±0.19 | **42.92** ±0.47 | **51.30** ±0.21 |

# D ADDITIONAL RESULTS ON ABLATION STUDY

In this section, we provide additional details on the ablation study to evaluate the contributions of the CnDRM and IoBMN components in SNAP-TTA. Specifically, we measured the average accuracy across 15 corruption types on CIFAR10-C and CIFAR100-C datasets under varying adaptation rates (0.3, 0.1, 0.05) to thoroughly assess the effectiveness of each component.

Tables 12, 13, 14, 15, and 16 summarize the results for different combinations of CnDRM and IoBMN across these adaptation rates. The results indicate that the combination of CnDRM (Class and Domain Representative sampling) and IoBMN (inference using memory statistics corrected to match the test batch) consistently yields the highest accuracy. This trend is observed across all evaluated adaptation rates, suggesting that both components contribute significantly to enhancing adaptation performance.

Moreover, individual evaluations show that each component has a distinct positive effect, as evidenced by consistently higher accuracy compared to using no adaptation or only a single component. This emphasizes the complementary nature of CnDRM and IoBMN, which together provide

robust adaptation capabilities for domain-shifted scenarios. These tables provide further insight into the benefits of each configuration and how the synergy of CnDRM and IoBMN results in improved robustness against various corruptions.

Table 12: STTA classification accuracy (%) of ablative settings on the CIFAR10-C, adaptation rate 0.5. Averaged over all 15 corruptions. **Bold** numbers are the highest accuracy.

| Methods | Tent | CoTTA | EATA | SAR | RoTTA |
|---|---|---|---|---|---|
| naïve | 78.86 | 69.75 | 79.02 | 77.83 | 75.39 |
| Random | 78.90 | 66.04 | 78.97 | 77.77 | 75.06 |
| LowEntropy | 78.68 | 63.74 | 78.42 | 76.21 | 72.83 |
| CRM | 80.32 | 66.50 | 80.14 | 75.78 | 75.49 |
| CnDRM | 79.62 | 77.68 | 79.63 | 78.22 | 75.85 |
| CnDRM+EMA | 80.96 | 72.42 | 80.27 | 78.19 | 76.73 |
| **CnDRM+IoDMN** | **81.23** | **78.75** | **81.30** | **79.77** | **77.41** |

Table 13: STTA classification accuracy (%) of ablative settings on the CIFAR10-C, adaptation rate 0.05. Averaged over all 15 corruptions. **Bold** numbers are the highest accuracy.

| Methods | Tent | CoTTA | EATA | SAR | RoTTA |
|---|---|---|---|---|---|
| naïve | 75.75 | 67.22 | 75.55 | 75.25 | 74.80 |
| Random | 75.82 | 65.90 | 75.56 | 75.27 | 74.91 |
| LowEntropy | 74.07 | 64.08 | 73.73 | 73.58 | 72.83 |
| CRM | 76.55 | 66.14 | 76.06 | 74.02 | 75.23 |
| CnDRM | 76.53 | 77.67 | 76.29 | 76.18 | 75.61 |
| CnDRM+EMA | 76.86 | 71.69 | 75.98 | 75.43 | 75.95 |
| **CnDRM+IoDMN** | **77.93** | **78.73** | **77.76** | **77.21** | **77.05** |

Table 14: STTA classification accuracy (%) of ablative settings on the CIFAR100-C, adaptation rate 0.3. Averaged over all 15 corruptions. **Bold** numbers are the highest accuracy.

| Methods | Tent | CoTTA | EATA | SAR | RoTTA |
|---|---|---|---|---|---|
| naïve | 53.36 | 39.11 | 49.97 | 56.65 | 49.84 |
| Random | 53.00 | 33.49 | 49.24 | 56.06 | 49.00 |
| LowEntropy | 53.53 | 32.29 | 45.51 | 55.84 | 44.77 |
| CRM | 54.21 | 32.86 | 47.42 | 56.40 | 46.68 |
| CnDRM | 55.15 | 50.02 | 51.36 | 57.72 | 50.74 |
| CnDRM+EMA | 55.39 | 41.34 | 50.11 | 57.68 | 49.88 |
| **CnDRM+IoDMN** | **57.27** | **50.32** | **52.19** | **58.44** | **51.55** |

Table 15: STTA classification accuracy (%) of ablative settings on the CIFAR100-C, adaptation rate 0.1. Averaged over all 15 corruptions. **Bold** numbers are the highest accuracy.

| Methods | Tent | CoTTA | EATA | SAR | RoTTA |
|---|---|---|---|---|---|
| naïve | 52.84 | 35.86 | 49.70 | 53.49 | 49.11 |
| Random | 52.68 | 33.18 | 49.39 | 53.42 | 48.84 |
| LowEntropy | 51.76 | 32.30 | 46.03 | 52.15 | 45.18 |
| CRM | 52.43 | 32.54 | 47.68 | 53.12 | 47.01 |
| CnDRM | 54.46 | 50.06 | 51.41 | 55.24 | 50.47 |
| CnDRM+EMA | 54.36 | 41.63 | 50.21 | 54.84 | 49.95 |
| **CnDRM+IoDMN** | **55.84** | **50.52** | **52.35** | **55.76** | **51.33** |

Table 16: STTA classification accuracy (%) of ablative settings on the CIFAR100-C, adaptation rate 0.05. Averaged over all 15 corruptions. **Bold** numbers are the highest accuracy.

| Methods | Tent | CoTTA | EATA | SAR | RoTTA |
|---|---|---|---|---|---|
| naïve | 51.24 | 33.20 | 49.81 | 51.50 | 49.12 |
| Random | 51.35 | 33.71 | 49.57 | 51.48 | 48.98 |
| LowEntropy | 49.79 | 32.36 | 46.65 | 49.51 | 45.41 |
| CRM | 50.17 | 32.74 | 47.47 | 50.49 | 46.58 |
| CnDRM | 52.86 | 50.08 | 51.47 | 53.09 | 50.44 |
| CnDRM+EMA | 52.68 | 41.43 | 50.32 | 52.80 | 50.04 |
| **CnDRM+IoDMN** | **54.13** | **50.63** | **52.43** | **53.59** | **51.41** |

# E ADDITIONAL ABLATE ANALYSIS

## E.1 DOMAIN INFLUENCE IN EARLY LAYER REPRESENTATIONS

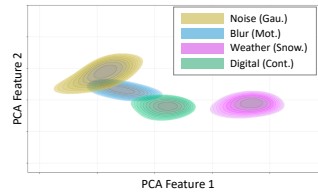

In deep learning models, early layers capture low-level features such as textures, edges, and frequency components (Zeiler & Fergus, 2014). These features are inherently domain-specific, making these layers more sensitive to shifts in input data distribution—a critical challenge for tasks requiring domain adaptation and generalization (Lee et al., 2018; Segu et al., 2023). This sensitivity arises because early layers encapsulate domain-specific patterns that may not generalize to new distributions. Under the covariate shift assumption (Quiñonero-Candela et al., 2008), while input distributions differ between source and target domains, the conditional distribution of labels remains the same. This discrepancy between input distributions makes early layers particularly vulnerable to domain shifts.

Figure 6: PCA embedding of early layer features for one domain from each of the four main CIFAR10-C corruption categories, showing clear separation between domains.

Visualizing early layer feature embeddings using 2D PCA on CIFAR-10C domains reveals distinct domain-specific patterns, highlighting the significant influence of domain information in these representations (Figure 6). Our preliminary experiments further confirm that sparse TTA, using the Wasserstein distance between moving batch normalization statistics and instance-specific statistics derived from early layer hidden features, can significantly improve performance. Selecting instances closer to the target domain distribution center using this distance metric yields better adaptation results, as demonstrated by performance comparisons between the top 20% and bottom 20% of samples (Figure 3). These findings emphasize the crucial role of domain-sensitive early layers in achieving effective adaptation.

## E.2 ANALYSIS ON CONFIDENCE THRESHOLD ON PSEUDO-LABEL ACCURACY

We analyzed the impact of using a confidence threshold for pseudo-label selection by comparing random sampling with high-confidence sampling across three benchmarks: CIFAR10-C, CIFAR100-C, and ImageNet-C. Table 17 shows that high-confidence sampling consistently outperformed random sampling, achieving significantly higher pseudo-label accuracy in all datasets. This result demonstrates the effectiveness of selecting high-confidence samples to improve the quality of pseudo-labels, thereby enhancing model adaptation under domain shift conditions.

Table 17: Pseudo-label accuracy comparison between random and high-confidence sampling on three benchmakrs: CIFAR10-C, CIFAR100-C, and ImageNet-C. **Bold** numbers are the highest accuracy.

|          | CIFAR10-C | CIFAR100-C | ImageNet-C |
|----------|-----------|------------|------------|
| Random   | 69.91     | 45.30      | 23.90      |
| **HighConf** | **74.80** | **59.38** | **59.40** |

## E.3 LATENCY TRACKING OF SNAP-TTA ON DIVERSE EDGE-DEVICES

To evaluate the latency efficiency of SNAP-TTA on resource-constrained edge devices, we measured the adaptation latency across three devices: NVIDIA Jetson Nano (NVIDIA Corporation, 2019), Raspberry Pi 4 (Raspberry Pi Foundation, 2019), and Raspberry Pi Zero 2 W (Raspberry Pi Foundation, 2021). These experiments compared the latency of SNAP-TTA with the Original TTA framework, specifically focusing on five state-of-the-art TTA algorithms: Tent (Wang et al., 2021), EATA (Niu et al., 2022), SAR (Niu et al., 2023), RoTTA (Yuan et al., 2023), and CoTTA (Wang et al., 2022). The experiments were conducted at an adaptation rate of 0.1, demonstrating the effectiveness of SNAP-TTA in reducing adaptation latency while maintaining competitive accuracy.

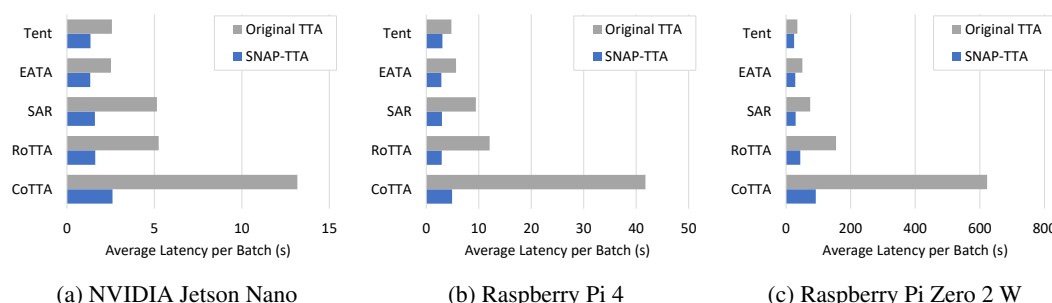

Figure 7: Latency comparison between SNAP-TTA and Original TTA across five state-of-the-art TTA algorithms (Tent, EATA, SAR, RoTTA, CoTTA) on three edge devices: (a) NVIDIA Jetson Nano, (b) Raspberry Pi 4, and (c) Raspberry Pi Zero 2 W. SNAP-TTA demonstrates significant latency reductions while maintaining competitive adaptation performance. The experiments were conducted at an adaptation rate of 0.1.

Figure 7 illustrates the latency performance for each device. It is evident that SNAP-TTA achieves a significant reduction in adaptation latency compared to the Original TTA framework. Notably, the latency reduction was proportional to the adaptation rate, validating the efficiency of SNAP-TTA in sparse adaptation scenarios. For instance, the latency for CoTTA was reduced by up to 87.5% on the Raspberry Pi 4, emphasizing the practical benefits of SNAP-TTA in latency-sensitive environments. Additionally, similar trends were observed across other devices, including the resource-limited Raspberry Pi Zero 2 W.

The results confirm that SNAP-TTA not only ensures substantial latency reductions but also adapts effectively to real-world conditions on diverse edge devices, proving its suitability for deployment in latency-sensitive applications.

### E.4 MEMORY OVERHEAD OF SNAP-TTA

The SNAP-TTA framework achieves substantial latency reduction and accuracy improvements with minimal memory overhead, even under resource-constrained scenarios like edge devices. In this section, we present both a theoretical analysis of the memory requirements and empirical results obtained from evaluations on a Raspberry Pi 4(Raspberry Pi Foundation, 2019) (CPU-only edge device).

The memory overhead of SNAP-TTA arises from two main components: (1) the memory buffer in Class and Domain Representative Memory (CnDRM) for storing representative samples, including both feature statistics (mean and variance) and the raw image samples, and (2) the statistics required for Inference-only Batch-aware Memory Normalization (IoBMN). For a batch size $B$, the total theoretical memory overhead can be expressed as: Memory Overhead $= B \times$ (Image Size $+ 2 \times$ Feature Dimension $\times$ Bytes per Value)+Feature Dimension$\times$Bytes per Value$\times$ 2. The last term accounts for the storage of IoBMN statistics (mean and variance for each feature channel). The image size is calculated based on the dataset resolution and data type.

For ResNet18 on CIFAR10-C, CIFAR10 images have a resolution of $32 \times 32 \times 3$ with each value stored as 1 byte. For a feature dimension of 512 and batch size $B = 16$, the total overhead is: Image Overhead $= 16 \times (32 \times 32 \times 3 \times 1) = 49,152$ bytes (48 KB), Feature Overhead (CnDRM) $= 16 \times (512 \times 2 \times 4) = 65,536$ bytes (64 KB), Feature Overhead (IoBMN) $= 512 \times 2 \times 4 = 4,096$ bytes (4 KB). Thus, the total memory overhead is: Total Overhead $= 48$ KB $+ 64$ KB $+ 4$ KB $= 116$ KB.

For ResNet50 on ImageNet-C, ImageNet images have a resolution of $224 \times 224 \times 3$, stored as 1 byte per value. For a feature dimension of 2048 and batch size $B = 16$, the total overhead is: Image Overhead $= 16 \times (224 \times 224 \times 3 \times 1) = 12,044,928$ bytes (11.5 MB), Feature Overhead (CnDRM) $= 16 \times (2048 \times 2 \times 4) = 262,144$ bytes (256 KB), Feature Overhead (IoBMN) $= 2048 \times 2 \times 4 = 16,384$ bytes (16 KB). Thus, the total memory overhead is: Total Overhead $= 11.5$ MB $+ 256$ KB $+ 16$ KB $\approx 11.77$ MB.

Table 18 shows the empirical memory usage of SNAP-TTA compared to Original TTA methods (Tent, EATA, CoTTA, SAR, and RoTTA). The results were averaged across three seeds of experiments and represent the memory footprint observed in a CPU-only edge device, Raspberry Pi 4. While minor variations in measurements are expected due to the nature of CPU memory footprint tracking, the results robustly indicate that the actual memory overhead of SNAP-TTA on edge devices is extremely low across all algorithms, ranging from 0.02% to 1.74%. Furthermore, while peak memory usage is either slightly increased or remains comparable to Original TTA methods, the average memory usage of SNAP-TTA is consistently lower. This is because SNAP-TTA performs backpropagation infrequently, which is the most memory-intensive operation in TTA.

Table 18: Comparison of memory usage (Average Memory, Peak Memory, and Memory Overhead) between Original TTA and SNAP-TTA (adaptation rate 0.3) across various methods (Tent, EATA, CoTTA, SAR, and RoTTA) tested on Raspberry Pi 4. **Bold** numbers are the lowest memory usage.

| Methods | Average Mem (MB) | | Peak Mem (MB) | | Mem Overhead (MB) |
| | Original TTA | SNAP-TTA | Original TTA | SNAP-TTA | SNAP - Original |
| --- | --- | --- | --- | --- | --- |
| Tent | 764.24 | **751.35** | **822.93** | 828.46 | 5.52 (0.67%) |
| CoTTA | 1133.52 | **1099.64** | **1211.21** | 1227.99 | 16.78 (1.13%) |
| EATA | 816.69 | **749.95** | **847.73** | 862.51 | 14.78 (1.74%) |
| SAR | 786.65 | **753.69** | **863.77** | 865.18 | 1.41 (0.02%) |
| RoTTA | 933.23 | **871.64** | **972.23** | 983.94 | 11.71 (1.20%) |

These findings demonstrate that **SNAP-TTA's memory overhead is negligible compared to its benefits in latency reduction and accuracy improvements**. By leveraging a small memory buffer for representative samples and minimizing backpropagation operations, SNAP-TTA not only achieves a lightweight memory profile but also becomes more efficient in terms of average memory usage compared to Original TTA. This lightweight design, combined with its advantages in latency and accuracy, underscores the practicality of SNAP-TTA for deployment in latency-sensitive applications on edge devices.

### E.5 INTEGRATION OF SNAP-TTA WITH MEMORY-EFFICIENT TTA ALGORITHM: MECTA (HONG ET AL., 2023)

This section evaluates the integration of SNAP-TTA with MECTA, a memory-efficient TTA algorithm, to demonstrate its applicability for resource-constrained edge devices. The experimental setup follows the evaluation settings presented in the MECTA paper to ensure a fair and consistent comparison. Specifically, we analyze the performance of Tent and EATA, enhanced with MECTA and further integrated with SNAP-TTA, using the ResNet50 model with a batch size of 64 on the ImageNet-C dataset.

Table 19 presents the classification accuracy and peak memory usage for Tent+MECTA and EATA+MECTA configurations with and without SNAP-TTA. Integrating SNAP-TTA with Tent+MECTA improves accuracy from 35.21% to 39.52%, while reducing peak memory usage by approximately 30% compared to the Tent baseline. Similarly, SNAP-TTA boosts the accuracy of EATA+MECTA from 35.55% to 42.86% while maintaining an efficient memory footprint.

Table 19: Comparison of classification (%) and memory peak (MB) in STTA with an adaptation rate of 0.1. MECTA significantly reduces memory consumption, and SNAP-TTA is applied alongside it to boost the performance of sparse adaptation. The accuracy is the average over 15 corruptions in ImageNet-C. **Bold** numbers indicate either the lowest memory usage or the highest accuracy.

| Methods | Accuracy (%) | Max Memory (MB) |
| --- | --- | --- |
| Tent | 35.21 | 6805.26 |
| +MECTA | 37.62 | **4620.25 (-32.10%)** |
| **+ MECTA + SNAP-TTA** | **39.52** | 4622.12 (-32.08%) |
| EATA | 35.55 | 6541.02 |
| +MECTA | 41.41 | **4512.38 (-31.01%)** |
| **+ MECTA + SNAP-TTA** | **42.86** | 4535.44 (-30.66%) |

Further details are provided in Table 20, which evaluates the combination of SNAP-TTA with MECTA across various corruption types and adaptation rates (AR = 0.3, 0.1, and 0.05). These results show that SNAP-TTA consistently outperforms baseline configurations across all adaptation rates and corruption types. This demonstrates the robustness of SNAP-TTA when integrated with MECTA and its suitability for real-world applications.

By adhering to the evaluation settings of the MECTA paper, this study ensures high reliability and comparability of results. The findings confirm that SNAP-TTA is highly compatible with MECTA, significantly improving both accuracy and memory efficiency. This synergy highlights the potential of combining SNAP-TTA and MECTA for deployment in resource-constrained environments such as edge devices.

Table 20: Evaluation of SNAP-TTA with MECTA on ImageNet-C through Adaptation Rates(AR) (0.3, 0.1, and 0.05). **Bold** numbers are the highest accuracy.

| AR | Methods | Gau. | Shot | Imp. | Def. | Gla. | Mot. | Zoom | Snow | Fro. | Fog | Brit. | Cont. | Elas. | Pix. | JPEG | Avg. |
|---|---|---|---|---|---|---|---|---|---|---|---|---|---|---|---|---|---|
| 0.3 | Tent + MECTA | 28.20 ±0.30 | 30.13 ±0.41 | 29.58 ±0.08 | 23.07 ±0.22 | 23.35 ±0.47 | 34.49 ±0.13 | 45.95 ±0.13 | 40.97 ±0.15 | 35.68 ±0.41 | 55.66 ±0.04 | 66.56 ±0.06 | 14.72 ±0.47 | 53.09 ±0.18 | 57.16 ±0.05 | 50.74 ±0.15 | 39.29 ±0.22 |
| | + SNAP-TTA | **30.49** ±0.26 | **31.98** ±0.14 | **31.66** ±0.21 | 26.29 ±0.32 | **38.47** ±0.02 | **47.38** ±0.30 | 43.79 ±0.11 | 40.12 ±0.11 | **56.38** ±0.12 | 66.81 ±0.05 | 28.87 ±0.07 | **53.53** ±0.28 | **57.61** ±0.09 | **50.86** ±0.10 | **42.03** ±0.08 | **±0.15** |
| | EATA + MECTA | 32.18 ±0.60 | 34.85 ±0.49 | 33.06 ±0.31 | 28.80 ±0.22 | 29.18 ±0.18 | 41.02 ±0.26 | 49.24 ±0.08 | 47.10 ±0.20 | 41.56 ±0.25 | 57.35 ±0.12 | **66.27** ±0.05 | 34.56 ±0.12 | 55.38 ±0.10 | 58.19 ±0.04 | **52.87** ±0.26 | 44.11 ±0.22 |
| | + SNAP-TTA | **33.67** ±0.19 | **35.76** ±0.24 | **34.86** ±0.31 | **30.35** ±0.11 | **30.29** ±0.04 | **42.78** ±0.06 | **49.55** ±0.10 | **47.46** ±0.10 | **42.32** ±0.10 | 57.50 ±0.05 | 66.18 ±0.15 | **39.08** ±0.06 | 55.38 ±0.81 | **58.35** ±0.16 | 52.72 ±0.12 | **45.08** ±0.15 |
| 0.1 | Tent + MECTA | 24.94 ±0.15 | 26.73 ±0.20 | 25.63 ±0.07 | 21.11 ±0.22 | 21.46 ±0.18 | 32.11 ±0.02 | 44.05 ±0.19 | 38.22 ±0.27 | 36.36 ±0.09 | **53.92** ±0.12 | 66.48 ±0.02 | 18.50 ±0.45 | 50.80 ±0.12 | **55.67** ±0.18 | **48.33** ±0.11 | 37.62 ±0.16 |
| | + SNAP-TTA | **27.49** ±0.08 | **28.90** ±0.14 | **28.26** ±0.16 | **23.49** ±0.17 | 23.76 ±0.12 | **34.92** ±0.06 | **45.18** ±0.13 | **40.21** ±0.09 | **38.40** ±0.18 | 53.78 ±0.14 | **66.54** ±0.03 | 27.72 ±0.20 | **51.00** ±0.20 | 55.48 ±0.13 | 47.61 ±0.17 | **39.52** ±0.13 |
| | EATA + MECTA | 29.42 ±0.67 | 31.72 ±0.30 | 29.44 ±0.32 | 24.41 ±0.74 | 25.48 ±0.45 | 37.04 ±0.18 | 47.10 ±0.15 | 43.60 ±0.19 | 39.43 ±0.38 | 55.95 ±0.13 | 66.42 ±0.14 | 28.85 ±0.15 | **53.70** ±0.15 | 57.34 ±0.15 | **51.20** ±0.36 | 41.41 ±0.37 |
| | + SNAP-TTA | **31.26** ±0.11 | **32.71** ±0.17 | **32.22** ±0.17 | **27.31** ±0.46 | **27.61** ±0.28 | **38.88** ±0.28 | **47.83** ±0.09 | **44.52** ±0.14 | **40.58** ±0.05 | **56.42** ±0.06 | 66.24 ±0.21 | **35.38** ±0.63 | 53.67 ±0.17 | **57.39** ±0.13 | 50.83 ±0.12 | **42.86** ±0.20 |
| 0.05 | Tent + MECTA | 21.22 ±0.13 | 23.19 ±0.22 | 21.90 ±0.13 | 18.69 ±0.18 | 19.39 ±0.20 | 29.89 ±0.13 | 42.02 ±0.10 | 36.53 ±0.22 | 35.23 ±0.05 | **51.75** ±0.15 | **66.23** ±0.04 | 19.64 ±0.27 | 48.43 ±0.03 | **53.54** ±0.13 | **45.43** ±0.11 | 35.54 ±0.14 |
| | + SNAP-TTA | **23.93** ±0.27 | **25.37** ±0.22 | **24.10** ±0.15 | **20.42** ±0.18 | **21.14** ±0.07 | **31.83** ±0.06 | **42.68** ±0.04 | **37.53** ±0.16 | **36.31** ±0.20 | 51.42 ±0.17 | 66.19 ±0.04 | **23.84** ±0.24 | **48.62** ±0.05 | 53.20 ±0.17 | 44.57 ±0.17 | **36.74** ±0.15 |
| | EATA + MECTA | 24.97 ±0.42 | 26.95 ±0.27 | 21.87 ±3.29 | 21.19 ±0.90 | 21.94 ±0.45 | 33.61 ±0.08 | 45.11 ±0.11 | 40.92 ±0.19 | 37.73 ±0.42 | 54.64 ±0.10 | 66.60 ±0.07 | 23.03 ±0.59 | 51.87 ±0.35 | **56.60** ±0.25 | 49.15 ±0.23 | 38.41 ±0.51 |
| | + SNAP-TTA | **28.39** ±0.57 | **30.10** ±0.38 | **29.45** ±0.22 | **24.32** ±0.20 | **25.12** ±0.07 | **35.54** ±0.20 | **46.04** ±0.27 | **41.87** ±0.07 | **39.16** ±0.15 | **55.12** ±0.01 | **66.61** ±0.09 | **30.34** ±0.34 | **52.06** ±0.24 | 56.42 ±0.11 | 49.11 ±0.07 | **40.64** ±0.20 |

# F    ADDITIONAL DISCUSSIONS

## F.1    EFFICIENT STRATEGY FOR RE-CALCULATION OF SAMPLE'S DISTANCE

The domain centroid in our framework is updated using a momentum-based approach to effectively capture recent shifts in the target domain. This ensures that the centroid remains adaptive to evolving distributions without being overly influenced by temporary fluctuations. However, during sparse adaptation (SA), where model updates occur at extended intervals, the data distribution can shift substantially between updates. Consequently, distances calculated for older samples may become outdated, leading to inconsistencies when comparing them to more recently added samples that are evaluated based on the updated centroid.

To address this issue efficiently, our Class and Domain Representative Memory (CnDRM) recalculates the distance of samples only when the shift in the domain centroid exceeds a predefined significance threshold. Specifically, if the change in the domain centroid $\Delta c_{domain}$ surpasses a threshold $\tau_\Delta$, the distances of all samples in memory are updated to reflect the new domain conditions. This threshold-based approach ensures that recalculations occur only when necessary, thereby minimizing computational costs while maintaining the representativeness of the memory.

In practice, we observed that the performance was not significantly affected as long as the threshold $\tau_\Delta$ was not set too high, indicating robustness to the choice of threshold. Based on these observations, we set $\tau_\Delta = 0.1$ and used this value consistently for all evaluations. By focusing recalculations on significant shifts, this strategy preserves consistency in sample selection, ensuring that both older and newer samples are compared fairly in the context of the current domain characteristics without excessive computational overhead.

## F.2 STRATEGY FOR CONTINUOUS DOMAIN SHIFT SETTING

In our proposed framework, the centroid used for selecting domain-representative samples naturally adapts to changes in the domain as new data is encountered. This mechanism inherently ensures that the centroid evolves to reflect the characteristics of the current domain, allowing for effective performance even under continual Test-Time Adaptation (TTA) scenarios, where the domain may gradually or abruptly shift during adaptation.

Instead of employing additional mechanisms like z-score evaluation to detect domain shifts, we rely on the natural adaptability of the centroid to adjust to the incoming data. This simplifies the design and avoids unnecessary overhead while maintaining robustness. As the domain characteristics evolve, the centroid continuously aligns with the new domain without requiring explicit detection of changes or manual intervention.

To validate the effectiveness of SNAP-TTA under continual domain shift scenarios, we conducted experiments across various benchmark datasets with incremental and abrupt domain shifts. Table 21 summarizes the results, demonstrating that SNAP-TTA maintains strong performance across evolving domains without requiring additional computational overhead for explicit domain shift detection.

Table 21: Performance of SNAP-TTA under continual domain shift scenarios. The table reports the accuracy (%) for different datasets with incremental and abrupt shifts. **Bold** numbers are the highest accuracy.

| AR | Method | Gau. | Shot | Imp. | Def. | Gla. | Mot. | Zoom | Snow | Fro. | Fog | Brit. | Cont. | Elas. | Pix. | JPEG | Avg. |
|---|---|---|---|---|---|---|---|---|---|---|---|---|---|---|---|---|---|
| | Tent | 24.68 ±0.45 | 19.65 ±1.27 | 5.12 ±1.22 | 0.63 ±0.05 | 0.43 ±0.02 | 0.40 ±0.04 | 0.44 ±0.06 | 0.41 ±0.03 | 0.30 ±0.03 | 0.33 ±0.04 | 0.42 ±0.05 | 0.24 ±0.04 | 0.32 ±0.02 | 0.31 ±0.05 | 0.31 ±0.04 | 3.60 ±0.23 |
| 0.1 | + SNAP-TTA | **28.71** ±0.66 | **30.60** ±1.82 | **22.91** ±2.25 | 6.13 ±0.90 | 1.62 ±0.20 | 0.87 ±0.13 | 0.88 ±0.07 | 0.64 ±0.08 | 0.64 ±0.06 | 0.66 ±0.05 | **0.75** ±0.01 | 0.44 ±0.05 | 0.60 ±0.08 | 0.63 ±0.07 | 0.61 ±0.07 | 6.45 ±0.43 |
| | CoTTA | 10.99 ±0.40 | 12.21 ±0.04 | 11.54 ±0.30 | 11.28 ±0.13 | 11.13 ±0.15 | 22.08 ±0.07 | 34.80 ±0.18 | 30.69 ±0.10 | 29.45 ±0.04 | 43.87 ±0.19 | 61.92 ±0.09 | 12.76 ±0.16 | 40.03 ±0.13 | 44.99 ±0.14 | 36.43 ±0.16 | 27.61 ±0.15 |
| | + SNAP-TTA | 15.19 ±0.17 | 15.97 ±0.11 | 15.91 ±0.02 | **13.94** ±0.04 | **14.18** ±0.03 | 24.76 ±0.07 | 36.50 ±0.23 | **32.61** ±0.04 | 31.76 ±0.06 | 46.14 ±0.10 | 63.60 ±0.14 | 15.60 ±0.04 | 42.17 ±0.02 | 46.77 ±0.06 | 38.08 ±0.12 | 30.21 ±0.08 |
| | Tent | 23.31 ±0.37 | 27.08 ±1.13 | 22.71 ±2.50 | 9.72 ±3.35 | 4.14 ±3.00 | 2.03 ±1.53 | 1.16 ±0.75 | 0.66 ±0.22 | 0.45 ±0.12 | 0.47 ±0.09 | 0.61 ±0.16 | 0.33 ±0.09 | 0.47 ±0.08 | 0.47 ±0.08 | 0.46 ±0.07 | 6.27 ±0.90 |
| 0.05 | + SNAP-TTA | 27.10 ±0.23 | **33.41** ±0.10 | **31.78** ±0.62 | **19.85** ±0.79 | **16.94** ±1.50 | 14.75 ±2.53 | 12.46 ±4.27 | 5.53 ±2.30 | 2.69 ±1.18 | 1.47 ±0.49 | 1.52 ±0.40 | 0.67 ±0.09 | 0.88 ±0.10 | 0.89 ±0.10 | 0.84 ±0.07 | **11.39** ±0.98 |
| | CoTTA | 11.04 ±0.38 | 12.25 ±0.39 | 11.73 ±0.42 | 11.62 ±0.10 | 11.25 ±0.59 | 22.05 ±0.13 | 34.89 ±0.13 | 30.73 ±0.20 | 29.50 ±0.17 | 44.09 ±0.18 | 61.87 ±0.09 | 12.87 ±0.18 | 40.15 ±0.17 | 45.06 ±0.19 | 36.53 ±0.14 | 27.71 ±0.23 |
| | + SNAP-TTA | **15.20** ±0.15 | 15.89 ±0.02 | 15.93 ±0.10 | 13.81 ±0.04 | 14.15 ±0.03 | 24.74 ±0.16 | 36.68 ±0.27 | 32.51 ±0.04 | 31.71 ±0.20 | 46.11 ±0.05 | 63.48 ±0.09 | 15.73 ±0.19 | 42.20 ±0.12 | 46.69 ±0.10 | 38.05 ±0.04 | 30.19 ±0.10 |

These results indicate that SNAP-TTA effectively handles both incremental and abrupt domain shifts, consistently outperforming baseline methods. By leveraging the natural adaptability of the centroid, SNAP-TTA provides a robust solution for continual domain adaptation in real-world scenarios. Notably, SNAP-TTA mitigates catastrophic forgetting not only through its sparse adaptation strategy but also by leveraging domain centroid-based sampling, allowing performance to be sustained longer in continual shift scenarios. Unlike Tent, CoTTA is specifically designed for continual domain shift environments, which highlights its superior performance under such conditions.

Future work could explore augmenting this adaptive mechanism by incorporating techniques like z-score evaluation to enable even more responsive adjustments. For instance, a z-score-based approach could further refine the centroid's responsiveness to subtle, gradual domain shifts by monitoring discrepancies between incoming data statistics and the current centroid. Such enhancements could make the system even more effective at handling continual domain evolution, particularly in scenarios with complex or noisy data streams.

## F.3 MODIFICATION FOR LAYER NORMALIZATION OF VIT

The main text describes the use of Batch Normalization (BN) statistics for calculating domain centroids and centroid-instance distances, with subsequent adjustment of memory statistics to match the target test batch using the Inference-only Batch-aware Memory Normalization (IoBMN) method. Specifically, these calculations leverage the mean and variance across batches as follows:

$$\bar{\mu}_c = \frac{1}{B \times L} \sum_{b=1}^{B} \sum_{l=1}^{L} f_{b,c,l}, \quad \bar{\sigma}_c^2 = \frac{1}{B \times L} \sum_{b=1}^{B} \sum_{l=1}^{L} (f_{b,c,l} - \mu_{b,c})^2, \tag{6}$$

where $B$ represents the batch size, $L$ the number of spatial locations, and $c$ the channel index.

However, modern models like Vision Transformer (ViT) utilize Layer Normalization (LN) instead of BN. Unlike BN, which calculates statistics across the entire batch, LN normalizes each instance independently by using the statistics calculated over individual feature dimensions. Specifically, for a feature vector $\mathbf{f}_b$ belonging to the $b$-th instance, LN computes:

$$\mu_b = \frac{1}{C} \sum_{c=1}^{C} f_{b,c}, \quad \sigma_b^2 = \frac{1}{C} \sum_{c=1}^{C} (f_{b,c} - \mu_b)^2, \tag{7}$$

where $C$ is the number of channels. This difference implies that LN operates without batch-level interactions, focusing solely on within-instance normalization, which makes the method inherently more suitable for handling variable batch sizes, particularly in latency-sensitive applications like those considered in our Test-Time Adaptation (TTA) setting.

Despite the differences between BN and LN, the fundamental mechanism of using feature statistics to capture domain information remains valid. The key domain characteristics in early layer features are preserved in both normalization types, enabling the construction of a domain centroid that reflects the distributional characteristics of the test data. For LN, this centroid can be computed by aggregating across instances instead of across batches:

$$\bar{\mu}_c^{\text{LN}} = \frac{1}{M} \sum_{b=1}^{M} \mu_b, \quad \bar{\sigma}_c^{2\text{LN}} = \frac{1}{M} \sum_{b=1}^{M} \sigma_b^2, \tag{8}$$

where $M$ is memory capacity. This modified approach allows the domain centroid to still represent the overall domain-specific characteristics effectively, despite the lack of direct batch-level statistics.

Furthermore, this methodology extends seamlessly to other normalization layers, such as Group Normalization (GN). In GN, the statistics are computed across smaller groups of channels within each instance, but the procedure for aggregating these statistics to form a domain centroid remains the same—by averaging the group-level statistics across instances.

To maintain the core concept of selecting domain-representative samples with minimal modifications, we continue to use the memory of high-confidence domain-representative samples in the Inference-only Batch-aware Memory Normalization (IoBMN) strategy. The adjustment for LN requires: 1. Calculating LN-specific centroids as described in Equation 8. 2. Replacing BN statistics with LN statistics in the IoBMN module, thereby aligning the feature normalization during inference with the domain-representative information derived from memory.

The effectiveness of this modification was validated experimentally, as shown in Table 5, where ViT models using LN showed improved performance even under sparse TTA conditions. This indicates that, with minimal adjustments, SNAP-TTA remains effective for ViT with LN. The core principle of utilizing domain-representative statistics for aligning test-time feature distributions continues to provide significant benefits, ensuring robust adaptation in shifting domains with limited latency and computational overhead.

## F.4 IMPACT OF MEMORY SIZE ON SNAP-TTA PERFORMANCE

The memory size of the Class and Domain Representative Memory (CnDRM) in SNAP-TTA has implications for both performance and privacy. Increasing memory size allows storing more samples, which intuitively could improve adaptation. However, such an approach raises privacy concerns and needs additional memory and latency when storing sensitive samples. To evaluate the trade-off, we conducted experiments on ImageNet-C under Gaussian noise corruption, using Tent + SNAP-TTA(adaptation rate 0.3) with a batch size of 16 and varying the memory size.

As shown in Table 22, increasing the memory size beyond the base configuration of 16 does not lead to significant performance gains. This observation highlights the efficiency of SNAP-TTA's representative sampling strategy, which prioritizes storing samples based on proximity to class and domain centroids. The saturation in accuracy suggests that a carefully aligned memory size to the batch size is sufficient to balance computational efficiency, performance, and privacy considerations.

Table 22: Performance comparison with varying memory sizes on ImageNet-C (Gaussian noise).

| Memory Size | Accuracy (%) |
|---|---|
| 16 (Base) | 26.60 |
| 32 | 28.44 |
| 64 | 28.89 |
| 128 | 28.60 |

In conclusion, to minimize computational overhead while ensuring robust test-time adaptation, the memory size in SNAP-TTA is designed to align with the batch size. This configuration addresses privacy and memory overhead risks by limiting the number of stored samples without compromising adaptation effectiveness.

## F.5 EFFECT OF LEARNING RATE ON SPARSE AND FULL ADAPTATION

To investigate the impact of learning rates on the performance of SNAP-TTA and baseline methods, we conducted experiments under sparse adaptation settings. Initially, the same learning rate was applied for each SOTA TTA algorithms across all adaptation rates to ensure fair comparisons (Table 6, 7, 8, 9, 10,and 11). However, as sparse adaptation inherently limits the number of updates, the updates might be insufficient at lower adaptation rates and explored the effect of increasing the learning rate.

The results, summarized in Table 23, reveal that higher learning rates improve the accuracy of both the naive baseline and SNAP-TTA under sparse settings. Notably, while the naive TTA baseline benefits from a higher learning rate, its performance still falls short of that achieved with full adaptation. In contrast, SNAP-TTA surpasses the performance of full adaptation at optimal learning rates, demonstrating its ability to leverage sparse adaptation effectively. At the same time, applying these higher learning rates to full adaptation results in model instability and collapse, underscoring the need to carefully tune learning rates based on adaptation frequency. Therefore, we selected a stable learning rate of $1 \times 10^{-4}$ for the evaluations in our work that balances model convergence and performance across all adaptation rates. These findings suggest that SNAP-TTA not only adapts effectively under sparse settings but also maintains robustness under optimized learning rates.

Table 23: Accuracy (%) with varying Learning Rates (LR) on ImageNet-C Gaussian noise adaptation rate 0.3.

| LR | Tent(Full) | Tent(STTA) | Tent+SNAP | CoTTA(Full) | CoTTA(STTA) | CoTTA+SNAP | EATA(Full) | EATA(STTA) | EATA+SNAP |
|---|---|---|---|---|---|---|---|---|---|
| $2 \times 10^{-3}$ | 2.31 | 7.04 | 13.69 | 13.31 | 11.88 | 14.67 | 0.36 | 0.59 | 0.75 |
| $1 \times 10^{-3}$ | 4.54 | 16.13 | 27.63 | 13.18 | 11.86 | 14.68 | 1.31 | 0.95 | 24.35 |
| $5 \times 10^{-4}$ | 10.22 | **24.96** | **29.95** | 13.15 | 11.85 | 15.11 | 21.96 | 20.96 | 27.72 |
| $1 \times 10^{-4}$ | **27.03** | 23.63 | 26.60 | 13.12 | 11.74 | **15.26** | **29.42** | **27.35** | **29.48** |
| $5 \times 10^{-5}$ | 26.34 | 20.94 | 24.87 | **13.34** | **11.92** | 14.85 | 29.37 | 26.07 | 27.9 |

In conclusion, selecting an appropriately high learning rate for sparse adaptation significantly enhances performance while ensuring model stability. This strategy is particularly useful for real-world deployment of SNAP-TTA, where computational efficiency and robust performance are paramount.

## G LICENSE OF ASSETS

**Datasets** CIFAR10/CIFAR100 (MIT License), CIFAR10-C/CIFAR100-C (Creative Commons Attribution 4.0 International), and ImageNet-C (Apache 2.0).

**Codes** Torchvision for ResNet18, ResNet50, and VitBase-LN (Apache 2.0), the official repository of CoTTA (MIT License), the official repository of Tent (MIT License), the official repository of EATA (MIT License), the official repository of SAR (BSD 3-Clause License), the official repository of RoTTA (MIT License), and the official repository of MECTA (Sony AI).

