# OpenReview forum: "SNAP-TTA: Sparse Test-Time Adaptation for Latency-Sensitive Applications"
_ICLR.cc/2025/Conference — Submitted to ICLR 2025_

### Official Review · Reviewer_BUbi · 2024-10-28

**Soundness:** 3
**Presentation:** 3
**Contribution:** 2
**Rating:** 6
**Confidence:** 3

**Summary:**

The authors propose a sparse test-time adaptation (TTA) framework, which they call SNAP, that improves the latency-accuracy trade-off of existing TTA algorithms to enable practical use of TTA on ede devices.
To this end, the authors propose "CnDRM", a method for identifying "important" samples for training based on class- and domain-representative sampling, and "IoBMN", a method for mitigating the effects of domain shifts on the model's internal feature distributions.

**Strengths:**

- The method is promising in that, at least on a Raspberry Pi 4 and when used together with STTA, SNAP provides a significant reduction in latency, as shown in Table 4, while being able to maintain accuracy comparable to using STTA alone.
- The authors show empirically that SNAP works well with a number of different TTA algorithms (TENT, CoTTA, EATA, SAR, RoTTA) and with different adaptation rates for different datasets (CIFAR10-C, CIFAR100-C, ImageNet-C).

**Weaknesses:**

- The claimed contribution of the paper is that SNAP can make existing TTA algorithms more latency efficient and suitable for edge devices. However, this is only demonstrated in Table 4 for one algorithm (STTA) and one target device (Raspberry Pi 4). All other experiments focus only on accuracy. And while it is an important and valuable contribution to properly demonstrate that SNAP does not reduce the effectiveness of the TTA algorithms it is applied to, I think the evaluation overall fails to adequately demonstrate the claimed contribution of latency reduction across various edge devices.

**Questions:**

- What are the lower limits of the proposed approach? For example, would SNAP enable TTA on microcontroller units (MCUs) such as Cortex-M MCUs?
- How memory intensive is the approach? There seem to be some mechanisms in place to keep memory requirements fixed (line 264 ff), but could memory, i.e. RAM, availability still become a bottleneck of the approach on edge systems?
- I am a bit confused about the hyperparameter "adaptation rate": Is this parameter specifically implemented by SNAP or is it implemented by the underlying TTA algorithms? I was wondering because, for example, in Table 1 the accuracy for the TTA algorithms without SNAP-TTA also decreases at lower adaptation rates.

---

> ### Author Response · Authors · 2024-11-22
> **Responses to Reviewer BUbi (Part 1)**
>
> We sincerely appreciate the time and effort you have devoted to reviewing our work and offering such valuable feedback. Below, we have addressed each of your points in detail.
> ___
> >W1: The claimed contribution of the paper is that SNAP can make existing TTA algorithms more latency efficient and suitable for edge devices. However, this is only demonstrated in Table 4 for one algorithm (STTA) and one target device (Raspberry Pi 4). All other experiments focus only on accuracy. And while it is an important and valuable contribution to properly demonstrate that SNAP does not reduce the effectiveness of the TTA algorithms it is applied to, I think the evaluation overall fails to adequately demonstrate the claimed contribution of latency reduction across various edge devices.
>
> Thank you for your thoughtful feedback and for identifying areas where our evaluation could be clarified. We would like to elaborate on a key aspect of our work to ensure clarity: **STTA (Sparse TTA)** is not a single algorithm but a **generalized adaptation protocol that selectively skips certain batches to meet latency constraints**. Without this protocol, full adaptation using original TTA methods would result in significantly higher latencies, as illustrated in *Figure 1*.
>
> In *Table 4*, we demonstrated how SNAP-TTA reduces latency while maintaining accuracy when applied to five SOTA TTA algorithms (Tent, CoTTA, EATA, SAR, and RoTTA). While our primary experiments used the Raspberry Pi 4 as a representative edge device, the lightweight nature of SNAP-TTA ensures its compatibility across a variety of hardware platforms, as it introduces minimal additional memory and computational overhead (*Appendix E.4*).
>
> To further address your concern, we **have added latency tracking experiments on additional edge devices (total 3)**, including **NVIDIA Jetson Nano** and **Raspberry Pi Zero 2W** on *Figure 7, Appendix E.3*. These results confirm that SNAP-TTA remains effective and compatible across different hardware. Specifically, the latency tracking results in the table below for CoTTA on **three edge devices demonstrate a substantial reduction in latency with SNAP-TTA** compared to fully adapting with the original TTA approach. These findings highlight the remarkable efficiency and robustness of SNAP-TTA on diverse devices.
> | Methods        | Latency on Jetson Nano (s) | Latency on RPi4 (s) | Latency on RPi Zero2w (s) |
> |----------------|---------------|---------------------|--------------------------|
> | Original TTA   | 13.18| 41.77| 622.28|
> | **+ SNAP-TTA** | **2.61 (-80.2%)**| **4.93 (-88.2%)**   | **92.01 (-85.2%)**|
>
> We hope this expanded evaluation addresses your concern about the demonstration of SNAP-TTA’s latency reduction across a broader range of devices.
> ___
> >Q1: What are the lower limits of the proposed approach? For example, would SNAP enable TTA on microcontroller units (MCUs) such as Cortex-M MCUs?
>
> Thank you for your insightful question. **Yes, SNAP-TTA is compatible with microcontroller units (MCUs)**, including Cortex-M MCUs, and can effectively enable TTA on such resource-constrained devices. Its feasibility depends on the specific model and TTA algorithm, which are suitable for MCU, but the point is the additional overhead introduced by SNAP-TTA is minimal.
>
> The approach relies on **two lightweight components that are both computationally simple and easy to implement on MCUs**: a memory buffer in Class and Domain Representative Memory (CnDRM) for storing representative samples and feature statistics, and Inference-only Batch-aware Memory Normalization (IoBMN) for efficient inference. These components are designed with simplicity in mind, requiring minimal processing and memory overhead.
>
> To address the computational limitations of MCUs, SNAP-TTA **adjusts the adaptation rate** to enable sparse adaptation, **significantly reducing the number of backpropagation** while prioritizing the most informative samples (via CnDRM) and ensuring efficient inference (via IoBMN). This makes it particularly well-suited to devices with limited resources. For detailed memory usage analysis, please refer to the response of Q2, *Appendix E.4 and E.5*. We hope these provides further clarity on the adaptability of SNAP-TTA to MCUs.

---

> ### Author Response · Authors · 2024-11-22
> **Responses to Reviewer BUbi (Part 2)**
>
> >Q2: How memory intensive is the approach? There seem to be some mechanisms in place to keep memory requirements fixed (line 264 ff), but could memory, i.e. RAM, availability still become a bottleneck of the approach on edge systems?
>
> SNAP-TTA introduces minimal memory overhead as it requires only (1) the memory buffer in Class and Domain Representative Memory (CnDRM) for storing representative samples, including both feature statistics (mean and variance) and (2) the statistics required for Inference-only Batch-aware Memory Normalization (IoBMN). Therefore, for a batch size $B$, the total memory overhead can be expressed as: $B \times \left( \text{Image Size} + 2 \times \text{Feature Dimension} \times \text{Bytes per Value} \right) + \text{Feature Dimension} \times \text{Bytes per Value} \times 2. $
>
> Then, in the case of ResNet18 on CIFAR, the total memory overhead is calculated as **only 116 KB**. Also on resource-constrained device Raspberry Pi 4, benchmarking results on ResNet50 and ImageNet-C show that SNAP-TTA incurs **negligible peak memory usage overhead (<1.8%)** compared to original TTA algorithms. Additionally, by reducing backpropagation frequency, SNAP-TTA **lowers average memory consumption**, enabling more flexible memory allocation for multitasking on edge devices. We have added both details of these memory overhead tracking results and theoretical analysis in *Appendix E.4*.
> |         | **Average Mem** |   **(MB)**   | **Peak Mem** | **(MB)** | **Mem Overhead (MB)** |
> |---------|:---------------:|:------------:|:------------:|:--------:|----------------|
> | Methods |   Original TTA  | **SNAP-TTA** | Original TTA | SNAP-TTA |  **SNAP - Original**  |
> | Tent    |          764.24 |   **751.35** |       822.93 |   828.46 |      **5.52 (0.67%)** |
> | CoTTA   |         1133.52 |  **1099.64** |      1211.21 |  1227.99 |     **16.78 (1.13%)** |
>
> Furthermore, SNAP-TTA can **integrate with memory-efficient methods like MECTA**[1] and can independently address latency concerns, enabling broader applicability for edge devices where both memory and latency efficiency are required. To demonstrate this synergy, we have added additional analysis and results in *Appendix E.5*. Below table shows the classification accuracy and peak memory usage for Tent+MECTA and EATA+MECTA configurations with and without SNAP-TTA. Integrating SNAP-TTA with Tent+MECTA **improves accuracy, while reducing peak memory usage by approximately 30\%** compared to the baseline. Similarly, SNAP-TTA boosts the accuracy of EATA+MECTA while maintaining an efficient memory footprint.
> | Method| Accuracy (%) | Max Memory (MB) | Reduced Memory (%) |
> |--------------------------|:------------:|:---------------:|:------------------:|
> | Tent|        35.21 |         6805.26 |- |
> |+ MECTA                |        37.62 |         4620.25 |              32.10 |
> | **+ MECTA + SNAP-TTA** |    **39.52** |     **4622.12** |          **32.08** |
> | EATA|        35.55 |6541.02 |- |
> |   + MECTA|        41.41 |         4512.38 |31.01 |
> | **+ MECTA + SNAP-TTA** |    **42.86** |     **4535.44** |          **30.66** |
>
> We sincerely thank the reviewer for highlighting this perspective, and we believe our approach contributes meaningfully to addressing these pressing challenges in edge-device TTA deployment.
>
> **_References_**
>
> [1] Hong, Junyuan, et al. "Mecta: Memory-economic continual test-time model adaptation." International Conference on Learning Representations. ICLR, 2023.
> ___
> >Q3: I am a bit confused about the hyperparameter "adaptation rate": Is this parameter specifically implemented by SNAP or is it implemented by the underlying TTA algorithms? I was wondering because, for example, in Table 1 the accuracy for the TTA algorithms without SNAP-TTA also decreases at lower adaptation rates.
>
> We appreciate your question and would like to clarify the concept of the "adaptation rate" and its implementation. **The adaptation rate is NOT the parameter specifically implemented by SNAP-TTA**. It is a **general concept that affects the frequency of updates and determines how sparsely adaptation occurs**. Therefore, in Table 1, the accuracy of TTA algorithms without SNAP-TTA decreases at lower adaptation rates because fewer updates result in a significant degradation when applying sparse TTA naively. This phenomenon highlights the challenge of sparse adaptation, which SNAP-TTA is designed to address. By introducing CnDRM and IoBMN, SNAP-TTA effectively mitigates the performance degradation associated with lower adaptation rates, ensuring efficient and reliable sparse adaptation even under stringent resource constraints.
>
> We hope this explanation clarifies the distinction and emphasizes SNAP-TTA’s role in addressing this critical issue.

---

> ### Comment · Reviewer_BUbi · 2024-11-23
>
> Thank you for responding to my concerns about the memory efficiency and latency of your approach on edge devices. While I find it very interesting to hear that you believe your approach is feasible for MCUs, it is unfortunate that you were unable to provide results for an actual Cortex-M based MCU (Jetson, RPi4 and RPi Zero2w are all Cortex-A based as far as I know). However, I understand that such additional experiments may not have been in the cards given the limited time during the rebuttal, and given that the code you shared relies on python/pytorch and does not appear to be easily executable on a platform that cannot run Linux or similar.
>
> Besides that, I would be interested to hear how your approach, and in particular the adjustable adaptation rate, relates to or could be combined with some of the recent developments discussed regarding memory and computation efficient backpropagation and on-device training of quantized DNNs on Cortex-M based MCUs [1, 2].
>
> Nevertheless, with the additional clarifications and additions made to the paper, it now passes my personal acceptance threshold.
>
> [1] Lin, Ji, et al. "On-device training under 256kb memory." Advances in Neural Information Processing Systems (2022).
>
> [2] Deutel, Mark, et al. "On-Device Training of Fully Quantized Deep Neural Networks on Cortex-M Microcontrollers." IEEE Transactions on Computer-Aided Design of Integrated Circuits and Systems (2024).

---

> > ### Author Response · Authors · 2024-11-24
> > **Thank you for responding to our rebuttal**
> >
> > We sincerely thank you for taking the time to read our rebuttal and for your thoughtful comments. Also, we are pleased that our clarifications and improvements have helped our paper meet your acceptance threshold.
> >
> > We agree that verifying SNAP-TTA on Cortex-M MCUs would require specialized libraries and adaptations, which we recognize as an important avenue for future work. Combining SNAP-TTA with recent advances in memory- and computation-efficient backpropagation for quantized DNNs on Cortex-M MCUs is a promising direction, and we appreciate you bringing this connection to our attention.
> >
> > Thank you again for your valuable feedback and consideration. We would welcome any additional suggestions or questions you may have.

---

### Official Review · Reviewer_vuQK · 2024-11-04

**Soundness:** 2
**Presentation:** 2
**Contribution:** 2
**Rating:** 5
**Confidence:** 5

**Summary:**

This paper focuses on Test-Time Adaptation (TTA) for edge devices with limited computational capacity. The authors propose SNAP-TTA, a sparse TTA framework with two key components, Class and Domain Representative Memory (CnDRM) and Inference-only Batch-aware Memory Normalization (IoBMN), aiming to reduce model adaptation frequency and data usage while maintaining accuracy.

**Strengths:**

The proposed SNAP-TTA framework addresses the latency-accuracy trade-off issue in existing TTA methods for edge devices in some cases. It reduces latency while achieving competitive accuracy, as demonstrated by extensive experiments on multiple benchmarks and with integration of several existing TTA algorithms.

**Weaknesses:**

- In the background section, the mention of applications like real-time health monitoring for IoT edge devices may not be entirely appropriate as these devices often have extremely limited memory.
With limited memory, these devices are difficult and even impossible for backward-propagation and gradient decent. In this sense, memory should perhaps be prioritized over latency as the primary concern.
- It is unclear whether the proposed method reduces the delay per batch or the average delay (adaptation occurs once every several batches as shown in Figure 1). If it is the latter, its effectiveness for latency-sensitive applications may be limited as the inference delay could increase significantly every several batches.
- The method reduces the cost of backpropagation by filtering samples to decrease the inference latency. However, EATA also uses a similar strategy, but in Figure 2, the delay of EATA is the same as that of Tent, and the delay of SAR is inconsistent with the results reported in its original paper.
- The paper could compare the inference latency in Tables 1, 2, and 3.
- In Table 6 for ImageNet-C, only the Tent method is compared, ignoring other methods, which could provide more comprehensive and convincing results.
- In the experiments, it is not clear how the number of participating samples is controlled to meet the adaptation rate. Is it through adjusting the $tau_conf$ hyperparameter? Also, it is not described how other compared methods meet the adaptation rate.
- The description of lines 10-15 of the algorithm in the paper is relatively brief, considering its importance for the proposed method. More detailed explanation in the paper would assist readers in understanding.

**Questions:**

NA

---

> ### Author Response · Authors · 2024-11-22
> **Responses to Reviewer vuQK (Part 1)**
>
> We sincerely appreciate the time and effort you have devoted to reviewing our work and offering such valuable feedback. Below, we have addressed each of your points in detail.
> ___
> >W1: In the background section, the mention of applications like real-time health monitoring for IoT edge devices may not be entirely appropriate as these devices often have extremely limited memory. With limited memory, these devices are difficult and even impossible for backward-propagation and gradient descent. In this sense, memory should perhaps be prioritized over latency as the primary concern.
>
> We appreciate the reviewer’s observation about the significance of memory constraints in edge devices. It is indeed true that limited memory can make backpropagation and gradient descent challenging, especially for extremely resource-constrained devices. However, recent advancements in memory-efficient algorithms and models[1-3], have significantly mitigated this issue. **These methods have enabled backpropagation to achieve competitive TTA performance even on devices with highly restricted memory.**
>
> Despite these advancements in addressing memory challenges these days, **another equally critical barrier remains: adaptation latency.** Most of the SOTA TTA methods [4-8] **still face significant hurdles in real-world latency-sensitive applications** due to their high adaptation latency, even when memory concerns are resolved. For example, many SOTA TTA methods require substantial computational resources for operations like backpropagation, augmentation, and ensembling, which makes them impractical for maintaining the inference frame rates required by latency-sensitive edge applications.
>
> Furthermore, latency concerns are not limited to edge devices. With the growing speed and volume of data streams, latency-sensitive applications across various domains demand efficient adaptation strategies. Specifically, **a recent study [9] has highlighted the latency issues in TTA** and proposed practical evaluation strategies for TTA algorithms, but **no solutions have been provided yet**. Our work specifically targets this underexplored issue in on-device TTA research by proposing SNAP-TTA, which addresses latency without sacrificing accuracy.
>
> Finally, we emphasize that **SNAP-TTA’s memory overhead is negligible (<1.8%)** as detailed analysis has been added in *Appendix E.4 *. Furthermore, **SNAP-TTA can integrate with memory-efficient methods like MECTA** and can independently address latency concerns, enabling broader applicability for edge devices where both memory and latency efficiency are required. To demonstrate this synergy, we have added additional analysis and results in Appendix E.5. Below table shows the classification accuracy and peak memory usage for Tent+MECTA and EATA+MECTA configurations with and without SNAP-TTA. Integrating SNAP-TTA with Tent+MECTA improves accuracy, while **reducing peak memory usage by approximately 30\%** compared to the baseline. Similarly, SNAP-TTA boosts the accuracy of EATA+MECTA while maintaining an efficient memory footprint.
> | Method   | Accuracy (%) | Max Memory (MB) | Reduced Memory (%) |
> |------|:-------:|:----------:|:-----:|
> | Tent|35.21|6805.26 |- |
> |+ MECTA|37.62 |4620.25 |32.10 |
> | **+ MECTA + SNAP-TTA** |    **39.52** |**4622.12** |**32.08** |
> | EATA|35.55 |6541.02 |- |
> |+ MECTA|41.41|4512.38 |31.01|
> | **+ MECTA + SNAP-TTA** |**42.86** |**4535.44** |**30.66** |
>
> **_References_**
>
> [1] Hong, Junyuan, et al. "Mecta: Memory-economic continual test-time model adaptation." International Conference on Learning Representations. ICLR, 2023.
>
> [2] Song, Junha, et al. "Ecotta: Memory-efficient continual test-time adaptation via self-distilled regularization." Proceedings of the IEEE/CVF Conference on Computer Vision and Pattern Recognition. CVPR, 2023.
>
> [3] Jia, Hong, et al. "TinyTTA: Efficient Test-time Adaptation via Early-exit Ensembles on Edge Devices." The Thirty-eighth Annual Conference on Neural Information Processing Systems. NeurIPS, 2024.
>
> [4] Wang, Dequan, et al. "Tent: Fully test-time adaptation by entropy minimization." International Conference on Learning Representations. ICLR, 2021.
>
> [5] Wang, Qin, et al. "Continual test-time domain adaptation." Proceedings of the IEEE/CVF Conference on Computer Vision and Pattern Recognition. CVPR, 2022.
>
> [6] Niu, Shuaicheng, et al. "Efficient test-time model adaptation without forgetting." International Conference on Machine Learning. ICML, 2022.
>
> [7] Niu, Shuaicheng, et al. "Towards stable test-time adaptation in dynamic wild world." International Conference on Learning Representations. ICLR, 2023.
>
> [8] Yuan, Longhui, Binhui Xie, and Shuang Li. "Robust test-time adaptation in dynamic scenarios." Proceedings of the IEEE/CVF Conference on Computer Vision and Pattern Recognition. CVPR, 2023.
>
> [9] Alfarra, Motasem, et al. "Evaluation of Test-Time Adaptation Under Computational Time Constraints." International Conference on Machine Learning. ICML, 2024.

---

> ### Author Response · Authors · 2024-11-22
> **Responses to Reviewer vuQK (Part 2)**
>
> >W2: It is unclear whether the proposed method reduces the delay per batch or the average delay (adaptation occurs once every several batches as shown in Figure 1). If it is the latter, its effectiveness for latency-sensitive applications may be limited as the inference delay could increase significantly every several batches.
>
> Thank you for raising this important point. Our current implementation reduces average latency by adapting sparsely across multiple batches, as shown in *Figure 1*. This approach reflects practical constraints in our PyTorch implementation, which applies adaptation at discrete intervals. Conceptually, **SNAP-TTA can also distribute backpropagation steps proportionally across batches, avoiding delay spikes while maintaining low latency**. This is an implementation-specific limitation and does not affect the core concept. In future work, we plan to explore dynamic strategies to further optimize latency-sensitive applications. Thank you for your valuable feedback.
> ___
> >W3: The method reduces the cost of backpropagation by filtering samples to decrease the inference latency. However, EATA also uses a similar strategy, but in Figure 2, the delay of EATA is the same as that of Tent, and the delay of SAR is inconsistent with the results reported in its original paper.
>
> Thank you for your insightful observation regarding the relationship between backpropagation cost, sample filtering, and latency reduction. SNAP-TTA employs a novel approach that significantly reduces backpropagation steps by processing only a small fraction of samples (as low as 10%) while maintaining competitive accuracy. **This distinguishes it from EATA, which filters samples for loss computation but does not skip backpropagation entirely**. As shown in our experiments and supported by the original EATA paper, the number of backpropagation steps—a key factor influencing latency—is not significantly reduced  in EATA, leading to comparable delays with Tent in previous Figure 2 (current *Figure 4*). Additionally, components such as sample filtering criteria computation and Fisher regularization added to EATA require additional calculations, which increase latency on the CPU. This makes the delay gap between the two appear smaller.
>
> Regarding the discrepancies in SAR latency results, the variation arises from differences in evaluation setups. Our evaluations were conducted on a CPU platform, while the original SAR paper used a GPU-based setup. Despite this, **the overall trends observed in our results align with SAR’s original findings, where SAR exhibits slightly higher latency than Tent and EATA**. We appreciate the reviewer’s feedback and hope these clarifications address the concerns while emphasizing the unique contributions of SNAP-TTA in achieving substantial latency reductions.
> ___
> >W4: The paper could compare the inference latency in Tables 1, 2, and 3.
>
> Thank you for the valuable suggestion regarding including latency comparisons in previous Tables 1, 2, and 3. These tables are focused on illustrating the consistent accuracy improvements of SNAP-TTA under sparse adaptation settings. To maintain clarity and avoid overloading the tables with information, we chose to **highlight latency reductions separately in *Table 4* (revised as *Table 3*)**, where both the latency reductions and accuracy gaps of SNAP-TTA (at an adaptation rate of 0.1) are detailed relative to the original TTA methods. We believe this separation ensures a clear presentation of both accuracy and latency performance without redundancy. To further address your feedback, we have added more detailed latency reduction results in the Appendix E.3, Figure 7, tested on additional edge-devices (**Raspberry Pi Zero 2W** and **NVIDIA Jetson Nano**) for readers who wish to explore this aspect more thoroughly. Specifically, the latency tracking results in the table below for CoTTA on **three edge devices demonstrate a substantial reduction in latency with SNAP-TTA compared to fully adapting with the original TTA approach**. These findings highlight the remarkable efficiency and robustness of SNAP-TTA on diverse devices.
>
> | Methods        | Latency on JetsonNano (s) | Latency on RPi4 (s) | Latency on RPiZero2w (s) |
> |----------------|---------------------------|---------------------|--------------------------|
> | Original TTA   | 13.18                     | 41.77               | 622.28                   |
> | **+ SNAP-TTA** | **2.61 (-80.2%)**         | **4.93 (-88.2%)**   | **92.01 (-85.2%)**       |
>
> We hope this approach balances clarity and depth while addressing your concern, and we sincerely appreciate your thoughtful feedback.

---

> ### Author Response · Authors · 2024-11-22
> **Responses to Reviewer vuQK (Part 3)**
>
> >W5: In Table 6 for ImageNet-C, only the Tent method is compared, ignoring other methods, which could provide more comprehensive and convincing results.
>
> We appreciate your feedback on previous Table 6. We have totally agreed that including other methods would make the results more comprehensive and convincing. Therefore, we **have additionally evaluated SNAP-TTA’s performance** not only with the Tent algorithm but also with **EATA and SAR on ViT-base** models. These additional results, presented in *Table 5*, demonstrate that SNAP-TTA consistently achieves higher accuracy gains across all algorithms, further validating its effectiveness and versatility when applied to transformer-based model. Detailed explanations of implementations of SNAP-TTA on ViT are in *Appendix F.3*. We sincerely appreciate your interest in this aspect of our work and hope the additional details in these sections comprehensively address your concerns.
> | Methods|    Gau.   |    Shot   |    Imp.   |    Def.   |    Gla.   |    Mot.   |    Zoom   |    Snow   |    Fro.   |    Fog    |   Brit.   |   Cont.   |   Elas.   |    Pix.   |    JPEG   |    Avg.   |
> |-------------------|:---------:|:---------:|:---------:|:---------:|:---------:|:---------:|:---------:|:---------:|:---------:|:---------:|:---------:|:---------:|:---------:|:---------:|:---------:|:---------:|
> | EATA|     20.12 |     21.52 |     21.40 |     20.90 |     23.42 |     15.71 |     18.00 |     16.12 |     28.35 |     22.24 |     35.97 |     11.33 |     19.78 |     20.22 |     19.99 |     21.00 |
> | **+ SNAP-TTA** | **40.74** | **43.22** | **43.11** | **40.63** | **44.59** | **51.58** | **50.63** | **54.77** | **58.32** |  **61.5** | **73.91** | **33.85** | **60.19** | **63.35** | **63.01** | **52.23** |
> | SAR|21.45 |     23.02 |     23.17 |     23.67 |     24.64 |     15.98 |     14.62 |      7.70 |     31.49 |      8.94 |     41.33 |      6.82 |     17.35 |     22.39 |     22.49 |     20.34 |
> | **+ SNAP-TTA** | **37.59** | **38.27** | **36.78** | **38.58** | **39.99** | **49.00** | **45.77** | **43.96** | **56.61** | **59.96** | **73.02** | **19.69** | **54.30** | **61.16** | **61.85** | **47.77** |
> ___
> >W6: In the experiments, it is not clear how the number of participating samples is controlled to meet the adaptation rate. Is it through adjusting the tau conf hyperparameter? Also, it is not described how other compared methods meet the adaptation rate.
>
> Thank you for raising this question. First of all, **‘tau conf’ doesn’t affect the sample number**, it’s an easy threshold that works in the first step in a multi-stage process of CnDRM designed to prioritize informative samples *(Algorithm 1)*. To clarify, **the number of participating samples for the adaptation is keeping the CnDRM memory size consistent with the initial batch size**. For example, with a batch size of 16 and an adaptation rate (AR) of 0.1, our method processes 160 streaming test samples but only uses 160×0.1=16 samples (i.e., one batch) for model updates. While it is possible to use more samples per update, we deliberately chose this setup to align with real-world edge device constraints, ensuring minimal latency and memory overhead. Thus, while the number of samples per model update remains fixed (matching the batch size), the total number of samples used for updates across all data streams is proportional to the adaptation rate. This consistent number of adaptation samples minimizes memory and latency overhead during backpropagation, which is particularly important for edge-device applications.
> ___
> >W7: The description of lines 10-15 of the algorithm in the paper is relatively brief, considering its importance for the proposed method. More detailed explanation in the paper would assist readers in understanding.
>
> Thank you for pointing out the brevity of the description for lines 10–15 of the algorithm in the paper. These lines describe the implementation of a prediction-balanced method for removing the domain-centroid farthest sample from memory. Since implementation details were not the main contribution of our work, we opted for a conceptual explanation rather than a detailed one. To elaborate further, the memory management mechanism tracks the number of stored samples for each prediction class. When attempting to store a new sample, the algorithm operates as follows:
> - If the prediction of the new sample belongs to the class with the highest number of stored samples, it removes the sample in that class that is farthest from the domain centroid and replaces it with the new one.
> - Otherwise, it removes the farthest sample from the class with the highest number of stored samples overall.
>
> To improve clarity and assist readers in understanding this process, **we have added more detailed explanations** (via comments) in *Algorithm 1*. We hope this additional context ensures that the mechanics of the algorithm are better communicated.

---

> > ### Comment · Reviewer_vuQK · 2024-11-24
> > **Some more questions**
> >
> > For the rebuttal of W1, could the authors provide the memory footprint of baseline without any adaptation (i.e., only forward propagations).
> >
> > For the rebuttal of W2, in my understanding, you mean the implementation would skip some batches and only perform backward-propagations in some batch? I found that in line 19 of Algorithm 1, it says: adaptation occurs every k batch. In this sense, I think the proposed method is not friendly for latency-sensitive applications since it would have an increasing latency every k batch. Since latency-sensitive application often requires a very stable inference latency in all the batch, the proposed method seems to be not suitable for latency-sensitive applications. Or could the authors show some applications that the proposed method is suitable for?
> >
> > For the rebuttal of W6, if the hyper-parameters k in line 19 of Algorithm 1, the adaptation rate is 1/k, right?
> >
> > ---
> >
> > Some more questions: if the adaptation rate is 0.1 (the settings in Table 2), the baseline EATA would only exploit 10% samples for adaptation? And it would further remove some samples in these 10%, leading to less than 10% samples for adaptation?

---

> > > ### Author Response · Authors · 2024-11-24
> > > **Responses to Reviewer vuQK (More questions)**
> > >
> > > Thank you very much for taking the time to read our rebuttal and your thoughtful follow-up questions. We deeply appreciate your feedback and would like to provide further clarification.
> > > > For the rebuttal of W1, could the authors provide the memory footprint of baseline without any adaptation (i.e., only forward propagations).
> > >
> > > We have added the Source (forward-only) row to the table below. It contains cache for model, optimizer, and inference.
> > > | Method    | Accuracy (%) | Max Memory (MB) | Reduced Memory (%) |
> > > |--------------------------|:------------:|:---------------:|:------------------:|
> > > | Source (forward-only)  |  18.15 |         1766.38 |                  - |
> > > | Tent        |        35.21 |         6805.26 |                  - |
> > > |   + MECTA                |        37.62 |         4620.25 |              32.10 |
> > > | **+ MECTA + SNAP-TTA** |    **39.52** |     **4622.12** |          **32.08** |
> > > | EATA     |        35.55 |         6541.02 |                  - |
> > > |   + MECTA                |        41.41 |         4512.38 |              31.01 |
> > > | **+ MECTA + SNAP-TTA**|    **42.86** |     **4535.44** |          **30.66** |
> > > ___
> > > >For the rebuttal of W2, in my understanding, you mean the implementation would skip some batches and only perform backward-propagations in some batch? I found that in line 19 of Algorithm 1, it says: adaptation occurs every k batch. In this sense, I think the proposed method is not friendly for latency-sensitive applications since it would have an increasing latency every k batch. Since latency-sensitive application often requires a very stable inference latency in all the batch, the proposed method seems to be not suitable for latency-sensitive applications. Or could the authors show some applications that the proposed method is suitable for?
> > >
> > > You are correct that the current implementation adapts every k batch, which introduces periodic latency spikes. This reflects practical constraints in the PyTorch framework, where backpropagation must occur as a single block. However, we would like to emphasize that **these spikes are not an inherent limitation of the main idea behind SNAP-TTA**, and can be mitigated by **distributing computational cost (e.g., backpropagation) across batches**. For example, while using a fixed model for inferences (skip-batches), the single backpropagation step for adaptation could be split into smaller portions and executed incrementally during the k batches. At the end of k batches, the accumulated updates are applied seamlessly, ensuring that no individual batch experiences a significant latency spike. This approach **retains the benefits of sparse adaptation while smoothing latency, aligning with the ‘*average latency per batch*’ focus of our current work**. Although not implemented in the present study, this strategy might be an extension of SNAP-TTA for applications with stringent latency requirements.
> > >
> > > Even in its current implementation, SNAP-TTA is suited for applications where occasional latency spikes are acceptable as long as the method ensures overall adaptability and efficiency. For instance, in **real-time video analytics tasks such as wildlife monitoring or environmental surveillance**, the ability to adapt to **gradual changes in lighting or weather** conditions is often more important than maintaining perfectly consistent latency for every frame. Similarly, in **adaptive anomaly detection** for industrial systems, where **periodic updates help refine detection over time**, minor and infrequent delays are not critical to the system's overall effectiveness. These scenarios highlight contexts where adaptability and robust performance across dynamic environments outweigh the need for absolute latency stability, making SNAP-TTA a practical choice despite its periodic latency variations. While we acknowledge the limitations of the current implementation, **our findings demonstrate the feasibility of sparse adaptation and its potential to enable efficient test-time adaptation in latency-sensitive scenarios**.
> > > ___
> > > >For the rebuttal of W6, if the hyper-parameters k in line 19 of Algorithm 1, the adaptation rate is 1/k, right?
> > >
> > > Yes, your understanding is correct.
> > > ___
> > > >Some more questions: if the adaptation rate is 0.1 (the settings in Table 2), the baseline EATA would only exploit 10% samples for adaptation? And it would further remove some samples in these 10%, leading to less than 10% samples for adaptation?
> > >
> > > You are correct. In that scenario, **10% of the total samples are used for adaptation**, but **the number of samples ultimately contributing to the loss calculation during adaptation is less than 10%** following the EATA’s additional step where it selects only reliable samples based on entropy. Note that this is not imposed by our sparse adaptation framework.
> > >
> > > We hope this answers your questions. Thank you again for your valuable feedback, and please don’t hesitate to let us know if there are follow-up questions.

---

> > > > ### Comment · Reviewer_vuQK · 2024-11-28
> > > > **More comments**
> > > >
> > > > The rebuttal partially addressed some of my concerns. I still think the applications of the proposed methods are limited. I am raising the scoring from 3 to 5.

---

> > > > > ### Author Response · Authors · 2024-11-29
> > > > >
> > > > > Thank you for taking the time to review our response and for thoughtfully reconsidering your score. We truly appreciate your feedback and would be happy to address any remaining concerns you may have. Please let us know if there is anything further we can do to completely address your concerns.

---

### Official Review · Reviewer_sYH2 · 2024-11-04

**Soundness:** 3
**Presentation:** 3
**Contribution:** 3
**Rating:** 6
**Confidence:** 5

**Summary:**

This paper addresses the problem of test-time adaptation for out-of-distribution generalization. To reduce the adaptation rate and improve the overall latency of TTA, the authors propose a SNAP framework that selects partial samples for adaptation. Experimental results highlight the potential of the proposed method. However, I still have several concerns as outlined below.

**Strengths:**

The design of the SNAP method is well-motivated and reasonable from the technical perspective.

The proposed approach is a plug-and-play module that can be integrated with existing TTA methods to reduce adaptation steps and enhance efficiency.

Experimental results underscore the effectiveness of the proposed method.

**Weaknesses:**

On edge devices, the most critical factor in determining whether a TTA method is feasible is actually peak memory usage, as highlighted by MECTA [A]. While this work does reduce the number of adaptation steps, it does not decrease peak memory usage. In this sense, the primary motivation for applying the proposed method to edge devices may be misplaced.

[A] MECTA: Memory-Economic Continual Test-time Adaptation

**Questions:**

I am somewhat confused about the latency differences between, Tent, EATA, SAR and SNAP, all of which are sample selection-based methods. Compared to Tent, EATA does not reduce latency because, this is because in the EATA’s code, even filtered samples are still used in back-propagation (due to limitations in PyTorch), despite halving the number of samples involved in adaptation. However, in SNAP, latency is reduced. If this reduction is due to engineering optimizations, the same should ideally apply to EATA and SAR for a fair comparison. If not, the comparison could be seen as unfair.

Another area of confusion is that, based on my experience, EATA generally outperforms Tent and SAR under standard settings. However, the authors’ results show SAR and Tent performing better than EATA, which contradicts my observations. Could the authors provide further clarification on this?

Does the proposed method reduce latency for a single batch or does it show an average improvement over multiple batches?

Lastly, would the proposed method be effective for transformer-based models, such as ViT-base?

I strongly encourage the authors to move Table 1 to the Appendix and provide additional results on ImageNet-C with various adaptation rates in the main paper, as the CIFAR-10 results are less critical and not sufficiently convincing. Currently, Table 1 occupies nearly an entire page, which I feel could be better utilized for more impactful content.

---

> ### Author Response · Authors · 2024-11-22
> **Responses to Reviewer sYH2 (Part 1)**
>
> We sincerely appreciate the time and effort you have devoted to reviewing our work and offering such valuable feedback. Below, we have addressed each of your points in detail.
> ___
> >W1: On edge devices, the most critical factor in determining whether a TTA method is feasible is actually peak memory usage, as highlighted by MECTA. While this work does reduce the number of adaptation steps, it does not decrease peak memory usage. In this sense, the primary motivation for applying the proposed method to edge devices may be misplaced.
>
> We appreciate the reviewer’s thoughtful comment on the critical importance of peak memory usage for edge devices. We agree that memory constraints are a key consideration, and we acknowledge that SNAP-TTA does not directly address peak memory usage compared to methods such as MECTA. However, our work focuses on a complementary and equally critical bottleneck: **adaptation latency**.
>
> While MECTA and similar approaches have made significant strides in mitigating memory constraints, the issue of high adaptation latency in TTA methods has become increasingly prominent. Most of SOTA TTA algorithms [1–5] involve computationally intensive processes, such as backpropagation, augmentation, and ensembling, which render them impractical for latency-sensitive applications on edge devices that require strict inference frame rates. Recent studies [8–9] have begun emphasizing the growing need to address adaptation latency for edge devices, though concrete guidelines or frameworks for latency-focused TTA are still lacking. **Specifically, a recent study [6] has highlighted the latency issues in TTA and proposed practical evaluation strategies for TTA algorithms, but no solutions have been provided yet**.
>
> SNAP-TTA fills this gap by providing **the first general strategy to reduce adaptation latency significantly while maintaining performance**. Unlike other efficient TTA approaches [7,8] that rely on custom algorithms or specialized model structures, SNAP-TTA integrates seamlessly with existing SOTA TTA algorithms. This allows it to retain their benefits while making them practically deployable on edge devices by reducing latency without introducing substantial accuracy trade-offs.
>
> Furthermore, **SNAP-TTA can be integrated with existing memory-efficient TTA method MECTA** [9]. By combining these approaches, we can address both **peak memory usage** and **latency concerns**, enabling broader applicability across various edge environments. To demonstrate this synergy, we have added additional analysis and results in Appendix E.5. The table below shows the classification accuracy and peak memory usage for Tent+MECTA and EATA+MECTA configurations with and without SNAP-TTA. Integrating SNAP-TTA with Tent+MECTA improves accuracy, while **reducing peak memory usage by approximately 30\%** compared to the baseline. Similarly, SNAP-TTA boosts the accuracy of EATA+MECTA while maintaining an efficient memory footprint.
> | Methods| Accuracy (%) | Max Memory (MB) | Reduced Memory (%) |
> |--------------------------|:------------:|:---------------:|:------------------:|
> | Tent | 35.21 |  6805.26 |  - |
> |+ MECTA |37.62 | 4620.25 |32.10 |
> | **+ MECTA + SNAP-TTA** |**39.52** | **4622.12** |          **32.08** |
> | EATA|  35.55 |   6541.02 |- |
> | + MECTA  | 41.41 | 4512.38 |  31.01 |
> | **+ MECTA + SNAP-TTA** | **42.86** |**4535.44** |  **30.66** |
>
> **_References_**
> [1] Wang, Dequan, et al. "Tent: Fully test-time adaptation by entropy minimization." International Conference on Learning Representations. ICLR, 2021.
>
> [2] Wang, Qin, et al. "Continual test-time domain adaptation." Proceedings of the IEEE/CVF Conference on Computer Vision and Pattern Recognition. CVPR, 2022.
>
> [3] Niu, Shuaicheng, et al. "Efficient test-time model adaptation without forgetting." International Conference on Machine Learning. ICML, 2022.
>
> [4] Niu, Shuaicheng, et al. "Towards stable test-time adaptation in dynamic wild world." International Conference on Learning Representations. ICLR, 2023.
>
> [5] Yuan, Longhui, Binhui Xie, and Shuang Li. "Robust test-time adaptation in dynamic scenarios." Proceedings of the IEEE/CVF Conference on Computer Vision and Pattern Recognition. CVPR, 2023.
>
> [6] Alfarra, Motasem, et al. "Evaluation of Test-Time Adaptation Under Computational Time Constraints." International Conference on Machine Learning. ICML, 2024.
>
> [7] Song, Junha, et al. "Ecotta: Memory-efficient continual test-time adaptation via self-distilled regularization." Proceedings of the IEEE/CVF Conference on Computer Vision and Pattern Recognition. CVPR, 2023.
>
> [8] Jia, Hong, et al. "TinyTTA: Efficient Test-time Adaptation via Early-exit Ensembles on Edge Devices." The Thirty-eighth Annual Conference on Neural Information Processing Systems. NeurIPS, 2024.
>
> [9] Hong, Junyuan, et al. "Mecta: Memory-economic continual test-time model adaptation." International Conference on Learning Representations. ICLR, 2023.

---

> ### Author Response · Authors · 2024-11-22
> **Responses to Reviewer sYH2 (Part 2)**
>
> >Q1: I am somewhat confused about the latency differences between, Tent, EATA, SAR and SNAP, all of which are sample selection-based methods. Compared to Tent, EATA does not reduce latency because, this is because in the EATA’s code, even filtered samples are still used in back-propagation, despite halving the number of samples involved in adaptation. However, in SNAP, latency is reduced. If this reduction is due to engineering optimizations, the same should ideally apply to EATA and SAR for a fair comparison. If not, the comparison could be seen as unfair.
>
> Thank you for your thoughtful feedback and for the opportunity to clarify the latency differences between SNAP-TTA, Tent, EATA, and SAR. I’d like to take this opportunity to clarify the distinctions and address your concerns.
>
> First, it is important to note that in all evaluations presented in the paper, SNAP-TTA was not directly compared to EATA or SAR in terms of latency. Instead, the comparisons were made between the original versions of SOTA methods and their enhanced versions, where SNAP-TTA was integrated. This approach ensures that differences in engineering optimizations did not lead to unfair comparisons.
>
> **SNAP-TTA uniquely adopts a strategic batch-skipping mechanism to adapt sparsely, significantly reducing the number of backpropagation steps and thus lowering latency**. This method fundamentally differs from EATA and SAR, which focus on filtering samples for loss computation but still perform backpropagation at the batch level. As only a small number of samples are filtered out in these methods, they end up conducting backpropagation on most batches, leading to considerable latency overhead. Additionally, it is worth emphasizing that neither EATA nor SAR was specifically designed with latency reduction as a primary objective. Even under ideal conditions (e.g., without the PyTorch limitations or based on theoretical analysis in their respective papers), these methods still involve backpropagation on at least 50% of the batches, inherently constraining their ability to reduce latency.
>
> In contrast, SNAP-TTA prioritizes latency efficiency, **achieving robust performance while utilizing only 10% or, in some cases, as little as 1%** of the samples for adaptation. This efficiency underscores SNAP-TTA’s suitability for latency-sensitive applications, particularly in meeting strict Service Level Objectives (SLOs). We hope this clarifies the distinctions between these methods and highlights SNAP-TTA’s complementary contributions to the field.
> ___
> > Q2: Another area of confusion is that, based on my experience, EATA generally outperforms Tent and SAR under standard settings. However, the authors’ results show SAR and Tent performing better than EATA, which contradicts my observations. Could the authors provide further clarification on this?
>
> We sincerely appreciate your insightful observation regarding EATA's performance under standard settings. Indeed, **EATA generally achieves comparable or superior results to Tent and SAR under a standard Adaptation Rate(AR) 1 (Top of *Table 1*)**, aligning with your observations. However, the other experimental results on lower adaptation rates were conducted in sparse adaptation scenarios, where the characteristics of EATA's performance can differ slightly.
>
> In sparse TTA settings, **EATA's reliance on low-entropy samples for adaptation becomes a limitation**. These samples often exhibit **low loss values, providing insufficient gradient updates**, and as a result, they become less informative for domain adaptation. This behavior leads to a **more pronounced performance drop for EATA compared to Tent**, which directly uses the current test batch for adaptation. These challenges in sparse TTA scenarios underline the need for a robust method like SNAP-TTA, which ensures effective adaptation even under such constraints.
>
> Additionally, we would like to acknowledge that EATA's performance, as well as that of other methods, is influenced by various hyperparameters such as batch size, learning rate, and other algorithm-specific settings. While we made our best effort to perform hyperparameter tuning within a reasonable scope, our primary focus in this study was not to compare the absolute performances of SOTA TTA algorithms against each other. Instead, the objective was to **demonstrate the effect of applying or not applying SNAP-TTA across these algorithms**. To this end, we focused on unifying the hyperparameters across for each algorithms and tuned them to a level that achieves reasonable convergence, ensuring a fair baseline for comparison *(Appendix B.1)*.
>
> We humbly request your understanding on this matter and emphasize that our intent was to **highlight the relative benefits introduced by SNAP-TTA rather than to claim definitive rankings among the existing algorithms under all settings.** We hope this explanation clarifies the observed differences and provides a clearer context for interpreting our results.

---

> ### Author Response · Authors · 2024-11-22
> **Responses to Reviewer sYH2 (Part 3)**
>
> >Q3: Does the proposed method reduce latency for a single batch or does it show an average improvement over multiple batches?
>
> Our current implementation focuses on demonstrating the conceptual and practical benefits of SNAP-TTA by evaluating its average latency improvement across multiple batches. This approach stems from limitations in our PyTorch-based implementation, which handles sparse adaptation uniformly across batches. However, conceptually, **SNAP-TTA can also reduce latency for a single batch by proportionally distributing backpropagation updates across adaptation intervals while performing inference in between**. This is a simple implementation detail that does not alter the core concept or its effectiveness. In future work, we plan to explore more dynamic adaptation strategies by incorporating device availability and batch-level overhead monitoring to further optimize latency.
> ___
> >Q4: Lastly, would the proposed method be effective for transformer-based models, such as ViT-base?
>
> Thank you for your question and for bringing up the applicability of our method to transformer-based models such as ViT-base. We are pleased to confirm that **SNAP-TTA is indeed effective for transformer-based architectures**. To provide further evidence of this, we have **additionally evaluated** SNAP-TTA’s performance not only with the Tent algorithm but also with **EATA and SAR on ViT-base models**. These additional results, presented in *Table 5*, demonstrate that SNAP-TTA consistently achieves higher accuracy gains across all algorithms, further validating its effectiveness and versatility when applied to transformer-based models. Detailed explanations of implementations of SNAP-TTA on ViT are in *Appendix F.3*. We sincerely appreciate your interest in this aspect of our work and hope the additional details in these sections comprehensively address your concerns.
> | Methods  |    Gau.   |    Shot   |    Imp.   |    Def.   |    Gla.   |    Mot.   |    Zoom   |    Snow   |    Fro.   |    Fog    |   Brit.   |   Cont.   |   Elas.   |    Pix.   |    JPEG   |    Avg.   |
> |-------------------|:---------:|:---------:|:---------:|:---------:|:---------:|:---------:|:---------:|:---------:|:---------:|:---------:|:---------:|:---------:|:---------:|:---------:|:---------:|:---------:|
> | EATA   |     20.12 |     21.52 |     21.40 |     20.90 |     23.42 |     15.71 |     18.00 |     16.12 |     28.35 |     22.24 |     35.97 |     11.33 |     19.78 |     20.22 |     19.99 |     21.00 |
> | **+ SNAP-TTA** | **40.74** | **43.22** | **43.11** | **40.63** | **44.59** | **51.58** | **50.63** | **54.77** | **58.32** |  **61.5** | **73.91** | **33.85** | **60.19** | **63.35** | **63.01** | **52.23** |
> | SAR |  21.45 |     23.02 |     23.17 |     23.67 |     24.64 |     15.98 |     14.62 |      7.70 |     31.49 | 8.94 |     41.33 |   6.82 |     17.35 |     22.39 | 22.49 |     20.34 |
> | **+ SNAP-TTA** | **37.59** | **38.27** | **36.78** | **38.58** | **39.99** | **49.00** | **45.77** | **43.96** | **56.61** | **59.96** | **73.02** | **19.69** | **54.30** | **61.16** | **61.85** | **47.77** |
> ___
> >Q5: I strongly encourage the authors to move Table 1 to the Appendix and provide additional results on ImageNet-C with various adaptation rates in the main paper, as the CIFAR-10 results are less critical and not sufficiently convincing. Currently, Table 1 occupies nearly an entire page, which I feel could be better utilized for more impactful content.
>
> Thank you for the suggestion. We appreciate the importance of including results for ImageNet-C with various adaptation rates in the main paper to better illustrate the method’s impact. While Table 1 was originally designed to highlight consistent gains across various SOTA algorithms under STTA settings, we recognize that presenting ImageNet-C results in the main text could provide stronger support for our contributions. In response to your feedback, we have revised the manuscript as follows:
> - We replaced *Table 1* in the main paper with results for ImageNet-C across major adaptation rates (0.3, 0.1, and 0.05).
> - We condensed the CIFAR-10-C and CIFAR-100-C results into concise tables (formerly *Tables 2 and 3*), keeping them in the main paper *(Table 2)*.
> - Expanded results, including the detailed breakdown for CIFAR-10-C, CIFAR-100-C, and ImageNet-C, remain in the appendix for readers who seek comprehensive details *(Appendix C)*.
>
> We hope this revision enhances the clarity and impact of the paper.

---

> > ### Comment · Reviewer_sYH2 · 2024-11-25
> > **Follow-up from reviewer**
> >
> > Thanks for the authors’ response. I appreciate that CnDRM operates in parallel with memory-optimization methods like MECTA and can be incorporated with MECTA.
> >
> > ### **Unfair comparison**:
> >
> > Regarding **efficiency comparison**, the authors compare SNAP-TTA (under the sparse setting, i.e., every $k$-th batch) with prior methods like **TENT and EATA (under the fully update setting, i.e., every batch)**. However, for **performance comparisons** across all tables, the authors compare SNAP-TTA with baselines including **TENT and EATA (under the sparse setting ($k$-batch updates)**. This comparison is unfair. While I understand the motivation to adopt the sparse setting to improve average efficiency, this paper risks misleading both the reviewers and readers into believing that SNAP-TTA achieves both higher efficiency and higher performance than baselines. Actually, when baselines are evaluated under the same sparse setting, SNAP-TTA does not demonstrate superior efficiency over them. The efficiency are the same when CnDRM size equals to the batch size.
> >
> > To ensure a fair comparison, if the efficiency is evaluated with baselines in the *fully update setting*, the performance comparisons should also be under the *fully update setting*. In the main tables, the results for all baselines in their original *fully update setting* need to be included.
> >
> > Moreover, how does the memory size of CnDRM affect the performance of SNAP-TTA? Given that saving samples may raise privacy concerns, this aspect should be carefully addressed.

---

> > > ### Comment · Reviewer_sYH2 · 2024-11-25
> > >
> > > One more question: how do you set the learning rate for SNAP-TTA + baseline methods?
> > >
> > > In sparse settings, the number of learning iterations for the baselines is limited, indicating that the updates are very insufficient. In this case, have you considered increasing the learning rate for the baselines? Would this lead to improved performance?

---

> > > > ### Author Response · Authors · 2024-11-25
> > > > **Response to Reviewer sYH2 (Follow-up questions)**
> > > >
> > > > Thank you very much for taking the time to read our rebuttal and your thoughtful follow-up questions. We deeply appreciate your feedback and would like to provide further clarification.
> > > > >To ensure a fair comparison, if the efficiency is evaluated with baselines in the fully update setting, the performance comparisons should also be under the fully update setting. In the main tables, the results for all baselines in their original fully update setting need to be included.
> > > >
> > > > Thank you for pointing this out, and we apologize for the confusion caused. To clarify, the efficiency of SNAP-TTA refers to achieving significantly lower latency than fully adaptive methods while maintaining comparable or superior accuracy.
> > > >
> > > > We also clarify that we did include fair accuracy comparisons with **fully update settings** across all evaluation scenarios (*Table 1, Table 6-Table 11*), and provided dedicated analyses on accuracy comparison with the full-adaptation in *Table 3* and *Figure 5*. We have just revised the captions of the tables to clarify this setup.
> > > >
> > > > The key findings of our work are two-fold: (1) **traditional algorithms experience accuracy drops under sparse adaptation**, and (2) **SNAP-TTA mitigates these drops while retaining latency benefits, thereby boosting efficiency**. The baselines and structure of the main tables were designed to highlight both aspects together. By leveraging class and domain representative samples, SNAP-TTA significantly narrows the trade-off between efficiency and accuracy in sparse adaptation, even outperforming fully adaptive methods in certain cases (e.g., *Table 3*).
> > > > ___
> > > > >Moreover, how does the memory size of CnDRM affect the performance of SNAP-TTA? Given that saving samples may raise privacy concerns, this aspect should be carefully addressed.
> > > >
> > > > As you pointed out, increasing memory size might introduce challenges such as privacy risks. We have conducted additional experiments on the ImageNet-C gaussian noise using the Tent + SNAP-TTA (adaptation rate 0.3) on batch size 16, varying the memory size to evaluate its effect on performance. The results, presented in the table below, show that **increasing memory size does not yield significant performance gains**. This outcome highlights that SNAP-TTA prioritizes storing representative samples, ranked by their proximity to class and domain centroids, which leads to a saturation point in the informativeness of the stored samples. Therefore, to minimize computational overhead while maintaining a plug-and-play structure for easy adaptation, we have designed the memory size to align with the batch size. This ensures efficient adaptation while addressing the potential risks of storing excessive samples. We have added this discussion in the revised paper *Appendix F.4*.
> > > > | Memory Size | Accuracy (%) |
> > > > |:---:|:---:|
> > > > | 16 (Base) | 26.60 |
> > > > | 32 | 28.44 |
> > > > | 64 | 28.89 |
> > > > | 128 | 28.60 |
> > > > ___
> > > > >One more question: how do you set the learning rate for SNAP-TTA + baseline methods?
> > > > In sparse settings, the number of learning iterations for the baselines is limited, indicating that the updates are very insufficient. In this case, have you considered increasing the learning rate for the baselines? Would this lead to improved performance?
> > > >
> > > > We initially set the same learning rate across the adaptation rates to ensure a fair comparison. Agreeing with your comment that the updates might be insufficient for sparse settings, we have conducted additional experiments with varying learning rates, as shown below (evaluated on ImageNet-C gaussian noise via Full/Sparse TTA with adaptation rate 0.3). The results show that higher learning rates improve accuracy for both baselines and SNAP-TTA. Despite this, **the best performance of the naive baseline remains below that of full adaptation** even with a larger learning rate. In contrast, **SNAP-TTA surpasses full adaptation, achieving higher accuracy**. However, applying such a high learning rate to full adaptation causes model collapse. Therefore, we selected a stable learning rate that ensures model convergence across all adaptation rates. We have added this result and discussion in the *Appendix F.5*.
> > > > | Learning Rate | Tent (Full) | Tent (Sparse TTA) | Tent + SNAP-TTA |
> > > > |:---:|:---:|:---:|:---:|
> > > > | $2 \times 10^{-3}\$ | 2.31 | 7.04 | 13.69 |
> > > > | $1 \times 10^{-3}\$ | 4.54 | 16.13 | 27.68 |
> > > > | $5 \times 10^{-4}\$ | 10.22 | **24.96** | **29.95** |
> > > > | $1 \times 10^{-4}\$ | **27.03** | 23.63 | 26.60 |
> > > > | $5 \times 10^{-5}\$ | 26.34 | 20.94 | 24.87 |
> > > >
> > > > Thank you again for your valuable feedback and consideration. We would welcome any additional suggestions or questions you may have.

---

> > > > > ### Comment · Reviewer_sYH2 · 2024-11-28
> > > > >
> > > > > I know that you report the results of baseline methods under the fully updated setting in some tables, as an adaptation rate of 1 corresponds to the fully update, so there is no need for clarification.
> > > > >
> > > > > However, the key question is that efficiency/latency is coupled with the adaptation rate. A low adaptation rate leads to improved latency for all methods. As such, if you compare efficiency/latency with fully update baselines (in Figure 1 of the original manuscript), it is equally important to compare performance with fully update baselines for SNAP-TTA across various adaptation rates in all tables.
> > > > >
> > > > > In some tables, such as Table 3 (ImageNet-C, commonly used in TTA) in the original paper, the results for fully update baselines are missing. Although Table 1 includes results for an adaptation rate of 1.0, the differences among baselines on CIFAR-10 are relatively minor. As a result, the manuscript gives me an impression of an unfair comparison overall.
> > > > >
> > > > > I understand and accept that your method achieves better performance under various sparse adaptation rates. However, to ensure fairness, both latency and performance should be compared under the same adaptation rate and with clear descriptions (as in Table 1 in the revised paper), to avoid the risk of misleading reviewers and readers. For all latency comparison figures, I strongly recommend including efficiency comparisons across different adaptation rates.
> > > > >
> > > > > Lastly, for results with varying learning rates, could you provide additional experiments under different adaptation rates and include more baselines? Addressing these points comprehensively is pretty important, and I would like to raise my score accordingly.

---

> > > > > > ### Author Response · Authors · 2024-11-29
> > > > > > **Response to Reviewer sYH2 (Part 1)**
> > > > > >
> > > > > > Thank you for taking the time to review our response and for providing your thoughtful suggestions. We deeply appreciate your insights and agree with the points you raised.
> > > > > > >However, the key question is that efficiency/latency is coupled with the adaptation rate. A low adaptation rate leads to improved latency for all methods. As such, if you compare efficiency/latency with fully update baselines (in Figure 1 of the original manuscript), it is equally important to compare performance with fully update baselines for SNAP-TTA across various adaptation rates in all tables.
> > > > > > In some tables, such as Table 3 (ImageNet-C, commonly used in TTA) in the original paper, the results for fully update baselines are missing. Although Table 1 includes results for an adaptation rate of 1.0, the differences among baselines on CIFAR-10 are relatively minor. As a result, the manuscript gives me an impression of an unfair comparison overall.
> > > > > >
> > > > > > Following your suggestion, **we have included a latency column in the main tables (*Table 1* and *Table 2*) for each adaptation rate and baseline**, including fully adaptive settings, in our most recent revision. Additionally, **we have incorporated both accuracy and latency comparisons with full adaptation in *Table 2*** to ensure fairness, similar to *Table 3*.
> > > > > >
> > > > > > Moreover, **we have updated Figure 4 to include accuracy alongside latency**, ensuring that no main table or figure in the paper now lacks a fully adaptation baseline for comparison as follow your recommendation.
> > > > > > >For all latency comparison figures, I strongly recommend including efficiency comparisons across different adaptation rates.
> > > > > >
> > > > > > To follow your recommendations about efficiency comparisons across different adaptation rates, **we have extended latency comparison between SNAP-TTA and Original TTA on additional Adaptation Rates(AR) 0.05 and 0.3 for three edge devices (Original Figure 7)**. The results tables are below. They demonstrate that SNAP-TTA consistently reduces latency in rates proportional to adaptation rate, regardless of the adaptation algorithm and edge device. Since the PDF update deadline has passed while running this experiment, we will include these results in the final draft of the paper.
> > > > > >
> > > > > > *Table A. Additional Latency Measurements (AR=1, 0.3, 0.1, 0.05) on Jetson Nano*
> > > > > > | Methods | AR=1 | AR=0.3 | AR=0.1 | AR=0.05 |
> > > > > > |:---:|:---:|:---:|:---:|:---:|
> > > > > > | Tent | 2.57 | 1.97 (-23.51%) | 1.35 (-47.62%) | 1.19 (-53.75%) |
> > > > > > | EATA | 2.52 | 1.90 (-24.70%) | 1.33 (-47.22%) | 1.19 (-52.79%) |
> > > > > > | SAR | 5.15 | 2.87 (-44.29%) | 1.60 (-68.94%) | 1.32 (-74.28%) |
> > > > > > | RoTTA | 5.24 | 2.91 (-44.46%) | 1.62 (-69.13%) | 1.32 (-74.81%) |
> > > > > > | CoTTA | 13.18 | 6.13 (-53.46%) | 2.61 (-80.22%) | 1.82 (-86.19%) |
> > > > > >
> > > > > > *Table B. Additional Latency Measurements (AR=1, 0.3, 0.1, 0.05) on Raspberry Pi 4*
> > > > > > | Methods | AR=1 | AR=0.3 | AR=0.1 | AR=0.05 |
> > > > > > |:---:|:---:|:---:|:---:|:---:|
> > > > > > | Tent | 4.78 | 3.54 (-26.09%) | 3.09 (-35.45%) | 2.35 (-50.87%) |
> > > > > > | EATA | 5.68 | 3.52 (-38.00%) | 2.87 (-49.45%) | 2.31 (-59.28%) |
> > > > > > | SAR | 9.45 | 4.88 (-48.34%) | 2.98 (-68.41%) | 2.54 (-73.16%) |
> > > > > > | RoTTA | 12.07 | 4.95 (-58.97%) | 2.94 (-75.62%) | 2.91 (-75.91%) |
> > > > > > | CoTTA | 41.77 | 11.80 (-71.76%) | 4.93 (-88.19%) | 3.64 (-91.29%) |
> > > > > >
> > > > > > *Table C. Additional Latency Measurements (AR=1, 0.3, 0.1, 0.05) on Raspberry Pi Zero 2W*
> > > > > > | Methods | AR=1 | AR=0.3 | AR=0.1 | AR=0.05 |
> > > > > > |:---:|:---:|:---:|:---:|:---:|
> > > > > > | Tent | 34.96 | 24.67 (-29.42%) | 25.06 (-28.32%) | 17.07 (-51.16%) |
> > > > > > | EATA | 50.72 | 27.01 (-46.75%) | 28.43 (-43.93%) | 17.00 (-66.48%) |
> > > > > > | SAR | 74.64 | 47.79 (-35.96%) | 29.56 (-60.40%) | 18.64 (-75.02%) |
> > > > > > | RoTTA | 154.88 | 86.54 (-44.13%) | 44.08 (-71.54%) | 22.44 (-85.51%) |
> > > > > > | CoTTA | 622.28 | 228.03 (-63.36%) | 92.01 (-85.21%) | 39.22 (-93.70%) |

---

> ### Author Response · Authors · 2024-11-29
> **Response to Reviewer sYH2 (Part 2)**
>
> >Lastly, for results with varying learning rates, could you provide additional experiments under different adaptation rates and include more baselines? Addressing these points comprehensively is pretty important, and I would like to raise my score accordingly.
>
> Following your suggestion, **we have conducted additional experiments not only for an adaptation rate of 0.3 but also for 0. and 0.1**, extending the baselines to **include Tent, CoTTA, and EATA**. The results of these experiments are included in the tables below. **By comparing the best accuracy across various learning rates, SNAP-TTA even outperforms fully adaptive settings in most scenarios.** Since the PDF update deadline has passed while running this experiment, we will include these results in the final draft of the paper.
>
> *Table D. ImageNet-C Gaussian Noise, Adaptation rate 0.5*
> | LR | Tent(Full) | Tent(STTA) | Tent+SNAP | CoTTA(Full) | CoTTA(STTA) | CoTTA+SNAP | EATA(Full) | EATA(STTA) | EATA+SNAP |
> |:---:|:---:|:---:|:---:|:---:|:---:|:---:|:---:|:---:|:---:|
> | 2e-3 | 2.31 | 4.16 | 6.68 | 13.31 | 12.03 | 14.58 | 0.36 | 0.48 | 0.69 |
> | 1e-3 | 4.54 | 10.19 | 16.37 | 13.18 | 11.98 | 14.63 | 1.31 | 1.36 | 22.11 |
> | 5e-4 | 10.22 | 18.43 | **28.36** | 13.15 | 11.95 | **15.17** | 21.96 | 13.97 | 25.42 |
> | 1e-4 | **27.03** | **25.24** | 28.05 | 13.12 | **11.99** | 15.16 | **29.42** | **28.62** | **30** |
> | 5e-5 | 26.34 | 22.62 | 26.32 | **13.34** | 12.1 | 14.93 | 29.37 | 27.3 | 28.76 |
>
> *Table E. ImageNet-C Gaussian Noise, Adaptation rate 0.3*
> | LR | Tent(Full) | Tent(STTA) | Tent+SNAP | CoTTA(Full) | CoTTA(STTA) | CoTTA+SNAP | EATA(Full) | EATA(STTA) | EATA+SNAP |
> |:---:|:---:|:---:|:---:|:---:|:---:|:---:|:---:|:---:|:---:|
> | 2e-3 | 2.31 | 7.04 | 13.69 | 13.31 | 11.88 | 14.67 | 0.36 | 0.59 | 0.75 |
> | 1e-3 | 4.54 | 16.13 | 27.63 | 13.18 | 11.86 | 14.68 | 1.31 | 0.95 | 24.35 |
> | 5e-4 | 10.22 | **24.96** | **29.95** | 13.15 | 11.85 | 15.11 | 21.96 | 20.96 | 27.72 |
> | 1e-4 | **27.03** | 23.63 | 26.60 | 13.12 | 11.74 | **15.26** | **29.42** | **27.35** | **29.48** |
> | 5e-5 | 26.34 | 20.94 | 24.87 | **13.34** | **11.92** | 14.85 | 29.37 | 26.07 | 27.90 |
>
> *Table F. ImageNet-C Gaussian Noise, Adaptation rate 0.1*
> | LR | Tent(Full) | Tent(STTA) | Tent+SNAP | CoTTA(Full) | CoTTA(STTA) | CoTTA+SNAP | EATA(Full) | EATA(STTA) | EATA+SNAP |
> |:---:|:---:|:---:|:---:|:---:|:---:|:---:|:---:|:---:|:---:|
> | 2e-3 | 2.31 | 18.06 | 27.41 | 13.31 | 10.93 | 14.8 | 0.36 | 1.86 | 9.59 |
> | 1e-3 | 4.54 | **25.46** | **31.12** | 13.18 | 10.93 | 14.73 | 1.31 | 2.86 | 24.95 |
> | 5e-4 | 10.22 | 24.71 | 28.01 | 13.15 | 10.92 | **15.18** | 21.96 | 18.76 | **28.09** |
> | 1e-4 | **27.03** | 22 | 26.21 | 13.12 | **11.74** | 15.13 | **29.42** | **22.43** | 26.1 |
> | 5e-5 | 26.34 | 16.72 | 19.31 | **13.34** | 10.92 | 14.76 | 29.37 | 20.32 | 23.28 |
>
> Once again, thank you for your kind and detailed feedback. Your suggestions have been invaluable in improving the quality and rigor of our work. Please let us know if there is anything further we can do to completely address your remaining concerns.

---

> ### Comment · Reviewer_sYH2 · 2024-11-29
>
> Thank you for your response. The paper, particularly the comparisons, is much clearer now.  I will increase my score. Pls also update the tables in Appendix to include latency comparisons in the future.

---

> > ### Author Response · Authors · 2024-11-29
> >
> > We sincerely appreciate the time and effort you have dedicated to reviewing our manuscript. Your insightful feedback and constructive suggestions have greatly enhanced the quality of our work, and we are especially pleased to hear that the revisions have clarified the comparisons, leading to an increased score.
> >
> > Thank you as well for recommending the inclusion of latency comparisons in the Appendix tables. We will ensure this improvement is reflected in the final update.
> >
> > We would be grateful for any additional suggestions or questions you may have.
> >
> > Best regards,
> >
> > Authors.

---

### Official Review · Reviewer_v9DY · 2024-11-04

**Soundness:** 3
**Presentation:** 3
**Contribution:** 2
**Rating:** 6
**Confidence:** 2

**Summary:**

This paper introduces SNAP-TTA, a sparse Test-Time Adaptation (STTA) framework designed for latency-sensitive applications on resource-constrained edge devices. Traditional TTA methods dynamically adjust models using unlabeled test data to handle distribution shifts, but they often incur high computational costs and latency, making them impractical for real-time edge environments. SNAP-TTA addresses these challenges by introducing two key components: (i) Class and Domain Representative Memory (CnDRM), which selects class-representative and domain-representative samples to enable effective adaptation with minimal data, and (ii) Inference-only Batch-aware Memory Normalization (IoBMN), which corrects feature distribution shifts during inference without additional training. By combining SNAP-TTA with five state-of-the-art TTA algorithms, the paper demonstrates that SNAP-TTA achieves significant latency reductions (up to 87.5%) while maintaining competitive accuracy. Experimental results on benchmarks like CIFAR10-C and ImageNet-C show SNAP-TTA’s superior performance in edge settings, making it suitable for real-world, latency-sensitive applications.

**Strengths:**

1. This paper addresses the challenge of achieving high adaptation accuracy while maintaining computational efficiency in Sparse Test-Time Adaptation (STTA), where updates rely on only a small subset of data.
2. SNAP-TTA demonstrates improved classification accuracy across adaptation rates (0.01 to 0.5) compared to baseline TTA methods on CIFAR10-C, CIFAR100-C, and ImageNet-C. At an adaptation rate of 0.1, SNAP-TTA reduces latency by up to 87.5% while mitigating accuracy loss, validating its effectiveness in STTA
3. IoBMN combines memory statistics from domain-representative samples with current inference batch statistics, using a soft shrinkage function to balance them. This dynamic normalization adjustment during inference effectively addresses domain shift, ensuring model adaptability and performance stability.

**Weaknesses:**

1. The reliance on a fixed confidence threshold of CnDRM may limit adaptability across varying data distributions and could lead to suboptimal sampling.
2. In table 5, accuracy differences between methods are small, without statistical analysis, making it unclear if these differences are significant (In Detailed comments 4)

**Questions:**

I have some comments as following:

1. Some results for latency and performance metrics on mobile or embedded systems would be helpful, to further validate the method’s effectiveness and robustness


2. Some in-depth analysis of specific limitations would be helpful, such as how memory overhead might impact performance on resource-constrained devices and how SNAP-TTA handles highly dynamic data distributions in real-world applications. Additionally, there is no discussion on potential trade-offs between latency reduction and accuracy under different conditions


3. The combined CnDRM+IoBMN method performs best, but the contribution of each component is not discussed. A brief explanation of how they work together would improve clarity. The table 5 only shows results at an adaptation rate of 0.1, the authors can mention that the complete data is in appendix.

---

> ### Author Response · Authors · 2024-11-22
> **Responses to Reviewer v9DY (Part 1)**
>
> We sincerely appreciate the time and effort you have devoted to reviewing our work and offering such valuable feedback. Below, we have addressed each of your points in detail.
> ___
> > W1: The reliance on a fixed confidence threshold of CnDRM may limit adaptability across varying data distributions and could lead to suboptimal sampling.
>
> Thank you for your thoughtful feedback regarding the use of a fixed confidence threshold in CnDRM. We understand that a static threshold may limit adaptability across varying data distributions, potentially leading to suboptimal sampling in certain scenarios.
>
> To clarify, the confidence threshold in our framework serves primarily as a safeguard to filter out highly noisy samples, particularly in the unsupervised domain adaptation setting where pseudo-label reliability can vary. It is not the sole sampling criterion but rather an initial step in a multi-stage process designed to prioritize informative samples *(Algorithm 1)*, so the most samples typically surpass this easy threshold.
>
> Additionally, **fixed thresholds are commonly used in SOTA TTA works to establish a baseline for evaluating new approaches**[1-4]. We followed this standard to emphasize the feasibility of our method as an initial investigation into improving sample efficiency. That said, we fully agree that dynamically adapting the threshold based on data characteristics could further enhance the adaptability and performance of CnDRM. We have included this consideration as a promising direction for future research in Section 5. We sincerely appreciate your valuable suggestion and will strive to address this aspect in greater depth in subsequent work.
>
> **_References_**
>
> [1] Niu, Shuaicheng, et al. "Efficient test-time model adaptation without forgetting." International Conference on Machine Learning. ICML, 2022.
>
> [2] Niu, Shuaicheng, et al. "Towards stable test-time adaptation in dynamic wild world." International Conference on Learning Representations. ICLR, 2023.
>
> [3] Gong, Taesik, et al. "Note: Robust continual test-time adaptation against temporal correlation." Advances in Neural Information Processing System. NeurIPS, 2022.
>
> [4] Gong, Taesik, et al. "SoTTA: Robust Test-Time Adaptation on Noisy Data Streams." Advances in Neural Information Processing Systems. NeurIPS, 2024.
> ___
> >W2: In table 5, accuracy differences between methods are small, without statistical analysis, making it unclear if these differences are significant (In Detailed comments 4)
>
> Thank you for highlighting the need for statistical analysis in ablative evaluation table. While the accuracy differences may appear not that great, **the gains of each component of SNAP-TTA are consistent across very diverse experimental setups**, including integration with TTA algorithms, adaptation rates, and datasets (All the variations are in *Appendix D, Table 12-16*). This consistency strongly supports the effectiveness of our proposed components, CnDRM and IoBMN.
>
> Furthermore, while the absolute accuracy improvements might seem modest, SNAP-TTA’s primary focus lies in optimizing latency-sensitive applications. In this context, the crucial performance metric is not just accuracy but the efficiency achieved through sparse adaptation compared to fully adaptive TTA. To directly address the statistical significance of these results, Table 3 provides a detailed analysis comparing SNAP-TTA with naive sparse TTA. Specifically, at an adaptation rate of 0.1, SNAP-TTA achieves an average **latency reduction of 60%** relative to full adaptation while maintaining a **minimal accuracy drop of only 1.1%** *(Table A)*. This **significant efficiency gain** demonstrates the practical value of our approach, even when absolute accuracy differences appear small.
>
> *Table A. Statistical analysis comparing latency and accuracy of SNAP-TTA with fully adaptive TTA methods across various state-of-the-art TTA algorithms.*
> |         | **Latency (s)** |                   | **Accuracy (%)** |                   |
> |---------|:---------------:|:-----------------:|:----------------:|:-----------------:|
> | Methods |   Original TTA  |    **SNAP-TTA**   |   Original TTA   |    **SNAP-TTA**   |
> | Tent    |            3.97 | **2.20 (-44.0%)** |            80.43 | **78.95 (-1.8%)** |
> | CoTTA   |           71.68 | **8.96 (-87.5%)** |            78.00 | **78.83 (+1.1%)** |
> | EATA    |            3.93 | **2.18 (-44.6%)** |            81.56 | **78.61 (-3.6%)** |
> | SAR     |            5.75 | **2.30 (-60.1%)** |            79.05 | **78.06 (+1.2%)** |
> | RoTTA   |            5.93 | **2.25 (-62.0%)** |            77.00 | **77.07 (+0.1%)** |

---

> > ### Author Response · Authors · 2024-11-22
> > **Responses to Reviewer v9DY (Part 2)**
> >
> > >Q1: Some results for latency and performance metrics on mobile or embedded systems would be helpful, to further validate the method’s effectiveness and robustness.
> >
> > Thank you for suggesting the inclusion of latency and performance metrics on diverse mobile and embedded systems.  In our study, the primary benchmarking was conducted using the widely-adopted edge device,  Raspberry Pi 4. To further validate the effectiveness and robustness of SNAP-TTA, we **extended** our performance analysis to include other popular edge devices: **Raspberry Pi Zero 2W** and **NVIDIA Jetson Nano**. The results of these additional evaluations, now detailed in the *Appendix E.3*, consistently demonstrate the significant efficiency gains of SNAP-TTA over the original TTA across various resource-constrained platforms. Specifically, the latency measurements for CoTTA **across the three edge devices** (below table) reveal a **substantial reduction in latency compared to fully adapting** (Original TTA). These findings highlight the remarkable efficiency and robustness of SNAP-TTA on diverse devices.
> > | Methods        | Latency on JetsonNano (s) | Latency on RPi4 (s) | Latency on RPiZero2w (s) |
> > |----------------|---------------------------|---------------------|--------------------------|
> > | Original TTA   | 13.18                     | 41.77               | 622.28                   |
> > | **+ SNAP-TTA** | **2.61 (-80.2%)**         | **4.93 (-88.2%)**   | **92.01 (-85.2%)**       |
> > ___
> > >Q2-1: Some in-depth analysis of specific limitations would be helpful, such as how memory overhead might impact performance on resource-constrained devices.
> >
> > We appreciate your request for a more in-depth analysis of limitations and potential trade-offs. SNAP-TTA introduces minimal memory overhead as it requires only (1) the memory buffer in Class and Domain Representative Memory (CnDRM) for storing representative samples, including both feature statistics (mean and variance) and (2) the statistics required for Inference-only Batch-aware Memory Normalization (IoBMN). Therefore, for a batch size $B$, the total memory overhead can be expressed as: $B \times \left( \text{Image Size} + 2 \times \text{Feature Dimension} \times \text{Bytes per Value} \right) + \text{Feature Dimension} \times \text{Bytes per Value} \times 2. $
> >
> > Then, in the case of ResNet18 on CIFAR, the total memory overhead is calculated as **only 116 KB**. Also on resource-constrained device Raspberry Pi 4, benchmarking results on ResNet50 and ImageNet-C show that SNAP-TTA incurs **negligible peak memory usage overhead (<1.8%)** compared to original TTA algorithms. Additionally, by reducing backpropagation frequency, SNAP-TTA **lowers average memory consumption**, enabling more flexible memory allocation for multitasking on edge devices. We have added both details of these memory overhead tracking results and theoretical analysis in *Appendix E.4*.
> > |         | **Average Mem** |   **(MB)**   | **Peak Mem** | **(MB)** | **Mem Overhead (MB)** |
> > |---------|:---------------:|:------------:|:------------:|:--------:|-----------------------|
> > | Methods |   Original TTA  | **SNAP-TTA** | Original TTA | SNAP-TTA |  **SNAP - Original**  |
> > | Tent    |          764.24 |   **751.35** |       822.93 |   828.46 |      **5.52 (0.67%)** |
> > | CoTTA   |         1133.52 |  **1099.64** |      1211.21 |  1227.99 |     **16.78 (1.13%)** |

---

> ### Author Response · Authors · 2024-11-22
> **Responses to Reviewer v9DY (Part 3)**
>
> >Q2-2: How SNAP-TTA handles highly dynamic data distributions in real-world applications?
>
> To address highly dynamic data distributions in real-world applications, we tested SNAP-TTA in a continuous domain adaptation scenario. Results of the below table illustrate that the **adaptive domain centroid effectively tracks continuous distribution changes**, ensuring sustained performance improvements. Moreover, the application of SNAP-TTA to the CoTTA algorithm yielded notable results: even with minimal adaptation rates (e.g., 0.1 or 0.05), SNAP-TTA performed slightly better than fully adapted models, demonstrating its reliability under challenging conditions. We have added the detailed analysis and result table in *Appendix F.2*.
> | Adaptation Rate| Method |  Gau.|Shot| Imp. | Def.|Gla.| Mot.|Zoom|Snow|Fro.|Fog|Brit. |   Cont.|   Elas.   |    Pix.   |    JPEG   |    Avg.|
> |------|-------------------|:---------:|:---------:|:---------:|:---------:|:---------:|:---------:|:---------:|:---------:|:---------:|:---------:|:---------:|:---------:|:---------:|:---------:|:---------:|:---------:|
> | 1    | CoTTA (Full)      |     13.19 |     13.42 |     13.16 |     11.74 |     11.74 |     22.84 |34.62 |     31.47 |     30.29 |     44.22 |     62.45 |     14.96 |     40.68 |     45.25 |     36.61 |     28.44 |
> | 0.1  | CoTTA      |     10.99 |     12.21 |     11.54 |     11.28 |     11.13 |22.08 |34.80 |     30.69 |     29.45 |     43.87 |     61.92 |     12.76 |     40.03 |     44.99 |     36.43 |     27.61 |
> |      | **+ SNAP-TTA** | **15.19** | **15.97** | **15.91** | **13.94** | **14.18** | **24.76** | **36.50** | **32.61** | **31.76** | **46.14** | **63.60** | **15.60** | **42.17** | **46.77** | **38.08** | **30.21** |
> | 0.05 | CoTTA     |     11.04 |     12.25 |     11.73 |     11.62 |     11.25 |     22.05 |     34.89 |     30.73 |     29.50 |     44.09 |     61.87 |     12.87 |     40.15 |     45.06 |     36.53 |     27.71 |
> |      | **+ SNAP-TTA** | **15.20** | **15.89** | **15.93** | **13.81** | **14.15** | **24.74** | **36.68** | **32.51** | **31.71** | **46.11** | **63.48** | **15.73** | **42.20** | **46.69** | **38.05** | **30.19** |
> ___
> >Q2-3: Additionally, there is no discussion on potential trade-offs between latency reduction and accuracy under different conditions.
>
> Regarding the trade-offs between latency reduction and accuracy, adjusting the adaptation rate to reduce latency directly limits the number of samples used for model updates. This limitation makes it more difficult to respond to both skewed data distributions and rapidly changing domains. Consequently, **as latency is reduced, there is a corresponding trade-off with adaptation accuracy compared to full adaptation**. This trade-off is particularly pronounced in scenarios with frequent domain shifts. However, SNAP-TTA mitigates these through its innovative mechanisms, including the Class and Domain Representative Memory (CnDRM) based on moving domain centroids and Inference-only Batch Memory Normalization (IoBMN). These features enable SNAP-TTA to achieve significant latency reductions while maintaining high domain adaptation performance, as demonstrated in *Table 6-11 and 21*. The results highlight **SNAP-TTA's ability to simultaneously reduce latency and improve accuracy in diverse scenarios**.
> ___
> >Q3: The combined CnDRM+IoBMN method performs best, but the contribution of each component is not discussed. A brief explanation of how they work together would improve clarity. The table 5 only shows results at an adaptation rate of 0.1, the authors can mention that the complete data is in appendix.
>
> The combined use of CnDRM and IoBMN is central to the success of our method in Sparse Test-Time Adaptation (STTA), and we appreciate the opportunity to clarify how these components work together.
> - CnDRM (Class-Domain Representative Memory) ensures that only the most informative and representative samples are selected for adaptation during sparse updates. This targeted selection reduces noise and maximizes the utility of limited adaptation opportunities, addressing the inherent challenges of STTA where updates occur infrequently.
> - IoBMN (Inference-only Batch-aware Memory Normalization) complements CnDRM by mitigating potential mismatches between the stored memory statistics (derived from adaptation batches) and the current inference data. IoBMN dynamically adjusts normalization by blending stable, representative statistics from memory with recent inference batch data. This ensures robust and adaptive normalization, even when adaptation is skipped for several batches.
>
> **The synergy between CnDRM and IoBMN ensures that the model remains both adaptive and robust**, even under sparse adaptation conditions, as demonstrated in our ablation study *(Table 4 and 12-16)*. We have added these detailed explanations in *Section 3.2 and 4*. We hope this explanation provides the clarity you requested, and we are happy to elaborate further if needed.

---

> > ### Comment · Reviewer_v9DY · 2024-11-27
> >
> > I appreciate the authors' detailed response. It addresses some of my concerns, so I will maintain the acceptance rating.

---

> > > ### Author Response · Authors · 2024-11-28
> > >
> > > Thank you for carefully going through our response. We are glad to hear it erased some of your concerns. Please let us know if there is anything we can do to completely resolve your remaining concerns.

---

### Author Response · Authors · 2024-11-22
**Global Response**

Dear Reviewers and Meta-Reviewers,

We sincerely thank you for your thoughtful and constructive feedback. Your suggestions have been invaluable in refining our work, and we deeply appreciate the time and effort you dedicated to reviewing our paper. We have carefully addressed all points and incorporated the necessary improvements in the revised version.

We are pleased that reviewers appreciated:
- The **novelty** and **practicality** of SNAP-TTA for Sparse Test-Time Adaptation (STTA), achieving **high accuracy with significant latency reduction** across challenging benchmarks. [v9DY, sYH2, BUbi, vuQK]
- The **technical soundness** of components of SNAP-TTA for handling domain shifts effectively. [v9DY]
- The plug-and-play design of SNAP-TTA, offering seamless integration with existing TTA methods and improving **efficiency**. [sYH2]
- The **strong empirical evidence**, demonstrating broad adaptability across diverse datasets, adaptation rates, and TTA algorithms. [BUbi, vuQK, sYH2]

We also sincerely thank the reviewers for their valuable suggestions, which helped us identify areas for improvement. In response to this feedback, we have made several major revisions and additions (marked as blue in paper):
- **Clarified the statistical significance** of performance gains in terms of efficiency in sparse adaptation scenarios and provided additional latency-performance trade-off evaluations. *(Section 4)* [v9DY]
- **Extended latency tracking** experiments to include **multiple edge devices** (Raspberry Pi Zero 2W, NVIDIA Jetson Nano) to demonstrate robustness across hardware. *(Appendix E.3)* [v9DY, BUbi, vuQK]
- Added **detailed memory usage analysis** to highlight the negligible overhead of SNAP-TTA and its **compatibility with memory-efficient methods** like MECTA. *(Appendix E.4 and E.5)* [v9DY, sYH2, vuQK, BUbi]
- Provided **additional experiments on transformer-based model** (e.g., ViT-base) to demonstrate SNAP-TTA’s versatility. *(Table 5)* [sYH2, vuQK]
- **Enhanced explanations** of CnDRM’s memory balancing mechanism for better clarity. *(Algorithm 1)* [vuQK]
- Provided additional experiments and analysis of SNAP-TTA under **continual domain shift** scenarios, demonstrating its adaptability and robustness in dynamic real-world scenarios. *(Appendix F.2)* [v9DY]

We believe these revisions address the reviewers' concerns and further strengthen our work. Thank you for your constructive feedback and recognition of our contributions.

Best regards,

The Authors

---

### Meta-Review · Area_Chair_Lo5f · 2024-12-24

**Metareview:**

This paper proposes a sparse test-time adaptation (TTA) framework called SNAP-TTA, which aims to address the latency-accuracy trade-off in existing TTA methods when applied to edge devices. The main benefit of SNAP-TTA is reducing latency. The motivation of the paper is reasonable, and it can be seamlessly integrated into existing TTA methods. However, the main issues lie in some experimental setups and the overall contribution. Some reviewers raised concerns about memory usage during experiments and compatibility across different hardware scenarios, and the authors' experimental additions and analyses did not fully address these concerns. The AC reviewed the paper and all discussions and believes that the study still needs improvement.

**Additional Comments On Reviewer Discussion:**

The common concerns raised by several reviewers include the robustness of the proposed method across different edge devices, memory usage, and the lack of Transformer-based experiments. Although the authors provided analysis and explanations, they did not fully address the reviewers' concerns.

---

### Decision · Program_Chairs · 2025-01-22

Reject